# RADEMACHER COMPLEXITY OVER $\mathcal{H}\Delta\mathcal{H}$ CLASS FOR ADVERSARIALLY ROBUST DOMAIN ADAPTATION

## ABSTRACT

In domain adaptation, a model is trained on a dataset generated from a source domain and its generalization is evaluated on a possibly different target domain. Understanding the generalization capability of the learned model is a longstanding question. Recent studies demonstrated that the adversarial robust learning under $\ell_\infty$ attack is even harder to generalize to different domains. To thoroughly study the fundamental difficulty behind adversarially robust domain adaptation, we propose to analyze a key complexity measure that controls the cross-domain generalization: the adversarial Rademacher complexity over $\mathcal{H}\Delta\mathcal{H}$ class. For linear models, we show that adversarial Rademacher complexity over $\mathcal{H}\Delta\mathcal{H}$ class is always greater than the non-adversarial one, which reveals the intrinsic hardness of adversarially robust domain adaptation. We also establish upper bounds on this complexity measure, and extend them to the ReLU neural network class as well. Finally, based on our adversarially robust domain adaptation theory, we explain *how adversarial training helps transferring the model performance to different domains*. We believe our results initiate the study of the generalization theory of adversarially robust domain adaptation, and could shed lights on distributed adversarially robust learning from heterogeneous sources – a scenario typically encountered in federated learning applications.

## 1 INTRODUCTION

Domain adaptation is a key learning scenario where one tries to generalize the model learnt on a source domain to a target domain. How to predict target accuracy using source accuracy has been a longstanding research topic in both theory Ben-David et al. (2006); Quinonero-Candela et al. (2008); Ben-David et al. (2010); Mansour et al. (2009); Cortes et al. (2015); Zhang et al. (2019; 2020) and application community Long et al. (2015); Saito et al. (2018); You et al. (2019). From a theoretical perspective, this problem can be attacked by establishing bounds on the generalization of the source-domain-learnt model on target domain, using different complexity measures including the VC-dimension Ben-David et al. (2006; 2010); Zhang et al. (2020) and Rademacher complexity Mansour et al. (2009); Zhang et al. (2019). In particular, the latter works Mansour et al. (2009); Zhang et al. (2019) rely on the Rademacher complexity over a so-called $\mathcal{H}\Delta\mathcal{H}$ function class to bound the gap between source and target generalization risks:

**Definition 1** ( Mansour et al. (2009))**.** *Let hypothesis space $\mathcal{H}$ be a set of real (vector)-valued functions defined over input space $\mathcal{X}$ and label space $\mathcal{Y}$: $\mathcal{H} = \{h_{\mathbf{w}} : \mathcal{X} \mapsto \mathcal{Y}\}$ each parameterized by $\mathbf{w} \in \mathcal{W} \subseteq \mathbb{R}^d$, and $\ell : \mathcal{Y} \times \mathcal{Y} \mapsto \mathbb{R}_+$ be the loss function. Given a dataset $\{\mathbf{x}_1, ..., \mathbf{x}_n\}$ sampled i.i.d. from distribution $\mathcal{D}$ defined over $\mathcal{X}$, the empirical Rademacher complexity of $\mathcal{H}\Delta\mathcal{H}$ over this dataset is defined as follows:*

$$\hat{\mathfrak{R}}_{\mathcal{D}}(\ell \circ \mathcal{H}\Delta\mathcal{H}) = \mathbb{E}_\sigma \left[ \sup_{h_{\mathbf{w}}, h_{\mathbf{w}'} \in \mathcal{H}} \frac{1}{n} \sum_{i=1}^n \sigma_i \ell(h_{\mathbf{w}}(\mathbf{x}_i), h_{\mathbf{w}'}(\mathbf{x}_i)) \right] , \tag{1}$$

*where $\sigma_1, \ldots, \sigma_n$ are* i.i.d. *Rademacher random variables with $\mathbb{P}\{\sigma_i = 1\} = \mathbb{P}\{\sigma_i = -1\} = \frac{1}{2}$.*

Intuitively, above quantity measures how well the loss vector realized by two hypotheses within $\mathcal{H}$ correlates with random vectors. The better correlation will imply a richer hypothesis class. However, unlike the classical Rademacher complexity whose loss vector is computed between predictions made by a hypothesis and true labels, Eq. (1) is defined merely over predictions made by two

hypotheses. Authors of Mansour et al. (2009); Zhang et al. (2019) have shown that this complexity measure controls the domain adaptation generalization bound. Unfortunately, none of those works give the precise analysis of $\hat{\Re}_{\mathcal{D}}(\ell \circ \mathcal{H}\Delta\mathcal{H})$. To our best knowledge, Kuroki et al. (2019) is the only prior work to analyze $\hat{\Re}_{\mathcal{D}}(\ell \circ \mathcal{H}\Delta\mathcal{H})$ on linear classifier class, but their analysis is not tight. Due to the importance of such complexity measure, we are interested in characterizing how large this complexity measure can be in terms of model dimension and data diversity, even on some toy model, e.g., linear model. Hence, the first question we investigate in this paper is: *for linear models, what quantities control the Rademacher complexity over $\mathcal{H}\Delta\mathcal{H}$ function class?*

Meanwhile, in modern machine learning, practitioners are not only interested in transferring standard model accuracy to another domain, but also in transferring robustness. Consider adversarially robust risk over domain $\mathcal{D}$: $\mathcal{R}_{\mathcal{D}}^{adv-label}(h_{\mathbf{w}}, y_{\mathcal{D}}) = \mathbb{E}_{\mathbf{x}\sim\mathcal{D}}\left[\max_{\|\boldsymbol{\delta}\|_{\infty}\leq\epsilon}\ell(h_{\mathbf{w}}(\mathbf{x}+\boldsymbol{\delta}), y_{\mathcal{D}}(\mathbf{x}))\right]$, where $y_{\mathcal{D}}(\cdot)$ is the labeling function. In the **adversarially robust domain adaptation** problem, we are curious about the robust risk when the same model $h_{\mathbf{w}}$ is tested on the new domain $\mathcal{D}'$. Unfortunately, as shown empirically Shafahi et al. (2019); Hong et al. (2021); Fan et al. (2021), robust model learnt on source domain will lose its robustness catastrophically on a different domain. That is, the gap between robust risks on the old domain and new domains can be dramatically huge, compared to the standard risk. This observation naturally leads to the question *Why is the robust risk harder to adapt to different domains?*, which we aim to examine in this paper. To answer this question, inspired by the Rademacher complexity over $\mathcal{H}\Delta\mathcal{H}$ function class, we properly extend this complexity measure to the adversarial learning setting, and propose the *adversarial Rademacher complexity over* the $\mathcal{H}\Delta\mathcal{H}$ *class*. We show that, the adversarial version complexity is **always greater than** its non-adversarial counterpart, similar to the results proven in Yin et al. (2019) in the single domain setting. Relying on this new complexity measure, for the first time, we characterize the generalization bound of adversarially robust learning between source and target domain.

Recent studies Salman et al. (2020); Deng et al. (2021) also show that, the model trained adversarially on the source domain, usually entails better standard accuracy on target domain, compared to the normally trained model. In this paper, by further exploring our generalization bound, we show that given large enough adversarial budget, **small source adversarially robust risk will almost guarantee small target domain standard risk**, with the residual error controlled by $\epsilon$. This connection between source robust risk and target standard risk theoretically supports the advantage of performing robust training in domain adaptation tasks.

Our contributions are summarized as follow:

- We study the Rademacher complexity over $\mathcal{H}\Delta\mathcal{H}$ class, and propose the adversarial variant of it, which is a new complexity measure towards better understanding the domain adaptation in adversarial learning. In both linear classification and regression settings, we first show that adversarial Rademacher complexity over $\mathcal{H}\Delta\mathcal{H}$ class is greater than its non-adversarial counterpart. We also show that adversarial complexity is smaller than its non-adversarial counterpart plus residual terms polynomially depending on data dimension, model norm and adversarial budget.
- We generalize our results to ReLU neural networks, where we derive the similar upper bounds of adversarial $\mathcal{H}\Delta\mathcal{H}$ Rademacher complexity of a 2-layer ReLU neural network.
- We also establish the connection between robust learning and standard domain adaptation, which helps explain the widely-observed phenomena that adversarially trained models can have good generalization performance on different domains.
- We support our theoretical analysis by providing experiments illustrating how adversarial training can help domain adaptation, especially with $\ell_1$ regularization. We also highlight numerically the difficulty of transferring adversarial robustness across domains.

## 2 PROBLEM SETUP

We adapt the following notations throughout this paper. We use lower case bold letter to denote vector, e.g., $\mathbf{w}$, and use upper case bold letter to denote matrix, e.g., $\mathbf{M}$. We use $\|\mathbf{w}\|_p$ and $\|\mathbf{M}\|_p$ to denote $\ell_p$-norm of vector $\mathbf{w}$ and matrix $\mathbf{M}$ respectively. We define the $(p,q)$-group norm as the $\|\mathbf{M}\|_{p,q} := \|(\|\mathbf{m}_1\|_p, \ldots, \|\mathbf{m}_n\|_p)^\top\|_q$ where the $\mathbf{m}_i$s are the columns of $\mathbf{M}$.

We use $\mathcal{D} : \mathcal{X} \mapsto \mathbb{R}$ to denote a data distribution (domain) defined over instance space $\mathcal{X}$, and $\hat{\mathcal{D}}$ be the empirical distribution with $n_\mathcal{D}$ samples drawn i.i.d. from $\mathcal{D}$. We let $\mathcal{H} := \{h_\mathbf{w} : \mathcal{X} \mapsto \mathcal{Y}\}$ be the hypothesis space, and vector $\mathbf{w} \in \mathcal{W} \subseteq \mathbb{R}^d$ denotes the model parametrization of $h_\mathbf{w}$. Given a loss function $\ell : \mathcal{Y} \times \mathcal{Y} \mapsto \mathbb{R}$, and a data distribution $\mathcal{D}$, we let $\mathcal{R}_\mathcal{D}(h_\mathbf{w}, h_{\mathbf{w}'}) := \mathbb{E}_{\mathbf{x} \sim \mathcal{D}}[\ell(h_\mathbf{w}(\mathbf{x}), h_{\mathbf{w}'}(\mathbf{x}))]$ be the risk of the *disagreement between models* $h_\mathbf{w}$ and $h_{\mathbf{w}'}$ on domain $\mathcal{D}$. Specially, when the second argument of $\mathcal{R}_\mathcal{D}(\cdot, \cdot)$ is the labeling function over $\mathcal{D}$, it becomes the commonly used risk function. We also define two adversarially robust risks: (1) Model-label robust risk as $\mathcal{R}_\mathcal{D}^{adv-label}(h_\mathbf{w}, y) := \mathbb{E}_{\mathbf{x} \sim \mathcal{D}}\left[\max_{\|\boldsymbol{\delta}\|_\infty \leq \epsilon} \ell(h_\mathbf{w}(\mathbf{x} + \boldsymbol{\delta}), y(\mathbf{x}))\right]$ and (2) Model-model robust risk as $\mathcal{R}_\mathcal{D}^{adv}(h_\mathbf{w}, h_{\mathbf{w}'}) := \mathbb{E}_{\mathbf{x} \sim \mathcal{D}}\left[\max_{\|\boldsymbol{\delta}\|_\infty \leq \epsilon} \ell(h_\mathbf{w}(\mathbf{x} + \boldsymbol{\delta}), h_{\mathbf{w}'}(\mathbf{x} + \boldsymbol{\delta}))\right]$. [1]

In the domain adaptation scenario, we consider source domain $\mathcal{S}$ and target domain $\mathcal{T}$ distributions, and let $\hat{\mathcal{T}}$ and $\hat{\mathcal{S}}$ be the empirical source and target distributions with $n_\mathcal{S}$ and $n_\mathcal{T}$ samples. A key quantity that controls the generalization in domain adaptation is the following discrepancy measure:

**Definition 2** ($\mathcal{H}\Delta\mathcal{H}$ discrepancy Mansour et al. (2009); Ben-David et al. (2010)). *Given a hypothesis class $\mathcal{H}$, risk function $\mathcal{R}_\mathcal{D}(\cdot, \cdot)$, $\mathcal{H}\Delta\mathcal{H}$ discrepancy between distributions $\mathcal{S}$ and $\mathcal{T}$ is defined by:*

$$disc_{\mathcal{H}\Delta\mathcal{H}}(\mathcal{S}, \mathcal{T}) = \max_{h_\mathbf{w}, h_{\mathbf{w}'} \in \mathcal{H}} |\mathcal{R}_\mathcal{S}(h_\mathbf{w}, h_{\mathbf{w}'}) - \mathcal{R}_\mathcal{T}(h_\mathbf{w}, h_{\mathbf{w}'})| \ . \tag{2}$$

The $\mathcal{H}\Delta\mathcal{H}$ discrepancy defines a semi-distance over two distributions, and it does not depend on the labeling function of two distributions hence invariant to potential model shift across domains. Another advantage of it, is that it can be efficiently estimated by finite samples, if the Rademacher complexity over $\mathcal{H}\Delta\mathcal{H}$ is finite. Hence based on Definitions 1 and 2, Mansour et al. (2009) derived the following generalization bound among source and target domains.

**Lemma 1** (Domain adaptation generalization lemma, consequence of Theorem 8 of Mansour et al. (2009)). *Assume that the loss function $\ell$ is symmetric and obeys the triangle inequality. We further assume $\ell$ is bounded by $M$. Then $\forall h_\mathbf{w} \in \mathcal{H}$, the following holds with probability at least $1 - c$:*

$$\mathcal{R}_\mathcal{T}(h_\mathbf{w}, y_\mathcal{T}) \leq \mathcal{R}_\mathcal{S}(h_\mathbf{w}, h_{\mathbf{w}_\mathcal{S}^*}) + disc_{\mathcal{H}\Delta\mathcal{H}}(\hat{\mathcal{S}}, \hat{\mathcal{T}}) + \mathcal{R}_\mathcal{T}(h_{\mathbf{w}_\mathcal{T}^*}, h_{\mathbf{w}_\mathcal{S}^*}) + \mathcal{R}_\mathcal{T}(h_{\mathbf{w}_\mathcal{T}^*}, y_\mathcal{T})$$

$$+ \hat{\mathfrak{R}}_\mathcal{S}(\ell \circ \mathcal{H}\Delta\mathcal{H}) + \hat{\mathfrak{R}}_\mathcal{T}(\ell \circ \mathcal{H}\Delta\mathcal{H}) + \left(3M\sqrt{\frac{\log(2/c)}{n_\mathcal{S}}} + 3M\sqrt{\frac{\log(2/c)}{n_\mathcal{T}}}\right),$$

*where $y_\mathcal{T}$ is the labeling function on target domain, $h_{\mathbf{w}_\mathcal{T}^*}, h_{\mathbf{w}_\mathcal{S}^*}$ are the best target and source models in $\mathcal{H}$, i.e., $h_{\mathbf{w}_\mathcal{S}^*} = \arg\min_{h \in \mathcal{H}} \mathcal{R}_\mathcal{S}(h, y_\mathcal{S})$ and $h_{\mathbf{w}_\mathcal{T}^*} = \arg\min_{h \in \mathcal{H}} \mathcal{R}_\mathcal{T}(h, y_\mathcal{T})$.*

The above bound successfully connects the target risk and source risk, with the help of Rademacher complexity over $\mathcal{H}\Delta\mathcal{H}$ and $disc_{\mathcal{H}\Delta\mathcal{H}}$ distance. It turns out that, $\hat{\mathfrak{R}}_\mathcal{S}(\ell \circ \mathcal{H}\Delta\mathcal{H})$ and $\hat{\mathfrak{R}}_\mathcal{T}(\ell \circ \mathcal{H}\Delta\mathcal{H})$ are the key complexity measures that control the generalization between different domains. Hence, to study the generalization of domain adaptation in the adversarial setting, it naturally motivates us to consider the following adversarial robust variant of this measure as defined below.

**Definition 3** (Adversarial Rademacher complexity over $\mathcal{H}\Delta\mathcal{H}$ class). *Let $\mathcal{H}$ be a set of real-valued hypothesis functions: $\mathcal{H} = \{h_\mathbf{w} : \mathcal{X} \mapsto \mathcal{Y}\}$, and $\ell(\cdot, \cdot) : \mathcal{Y} \times \mathcal{Y} \mapsto \mathbb{R}$ be the loss function. Given a dataset $\{\mathbf{x}_1, ..., \mathbf{x}_n\}$ sampled from distribution $\mathcal{D}$, the empirical adversarial Rademacher complexity of $\mathcal{H}\Delta\mathcal{H}$ over this dataset is defined as follows*

$$\hat{\mathfrak{R}}_\mathcal{D}(\tilde{\ell} \circ \mathcal{H}\Delta\mathcal{H}) = \mathbb{E}_\sigma \left[\sup_{h_\mathbf{w}, h_{\mathbf{w}'} \in \mathcal{H}} \frac{1}{n} \sum_{i=1}^{n} \sigma_i \max_{\|\boldsymbol{\delta}\|_\infty \leq \epsilon} \ell(h_\mathbf{w}(\mathbf{x}_i + \boldsymbol{\delta}), h_{\mathbf{w}'}(\mathbf{x}_i + \boldsymbol{\delta}))\right] \ , \tag{3}$$

*where $\sigma_1, \ldots, \sigma_n$ are i.i.d. Rademacher random variables with $\mathbb{P}\{\sigma_i = 1\} = \mathbb{P}\{\sigma_i = -1\} = \frac{1}{2}$.*

As we can see, (3) is defined over the class:

$$\tilde{\ell} \circ \mathcal{H}\Delta\mathcal{H} := \left\{\mathbf{x} \mapsto \max_{\|\boldsymbol{\delta}\|_\infty \leq \epsilon} \ell(h_\mathbf{w}(\mathbf{x} + \boldsymbol{\delta}), h_{\mathbf{w}'}(\mathbf{x} + \boldsymbol{\delta})) : h_\mathbf{w}, h_{\mathbf{w}'} \in \mathcal{H}\right\} \ .$$

We will see later how this quantity controls the generalization of adversarial domain adaptation. We also generalize $\mathcal{H}\Delta\mathcal{H}$ discrepancy to the adversarial setting:

---

[1] $\mathcal{R}_\mathcal{D}^{adv-label}$ and $\mathcal{R}_\mathcal{D}^{adv}$ are also called constant-in-ball risk and exact-in-ball risk in Gourdeau et al. (2021).

**Definition 4** (Adversarial $\mathcal{H}\Delta\mathcal{H}$ discrepancy). *Given a hypothesis class $\mathcal{H}$, risk function $\mathcal{R}_{\mathcal{D}}^{adv}(\cdot,\cdot)$, the adversarial $\mathcal{H}\Delta\mathcal{H}$ discrepancy distance between two distributions $\mathcal{S}$ and $\mathcal{T}$ is defined by:*

$$disc_{\mathcal{H}\Delta\mathcal{H}}^{adv}(\mathcal{S},\mathcal{T}) = \max_{h_{\mathbf{w}}, h_{\mathbf{w}'} \in \mathcal{H}} |\mathcal{R}_{\mathcal{S}}^{adv}(h_{\mathbf{w}}, h_{\mathbf{w}'}) - \mathcal{R}_{\mathcal{T}}^{adv}(h_{\mathbf{w}}, h_{\mathbf{w}'})| \ . \tag{4}$$

The definition of adversarial $\mathcal{H}\Delta\mathcal{H}$ discrepancy is analogous to standard one, and for linear models can indeed be estimated as a function of the latter. We defer this result to Appendix F, Lemma 19.

**Lemma 2** (Adversarially robust domain adaptation generalization lemma). *Assume that the loss function $\tilde{\ell}$ is symmetric and obeys the triangle inequality. We further assume $\tilde{\ell}$ is bounded by $M$. Then, for any hypothesis $\mathbf{w} \in \mathcal{H}$, the following holds:*

$$\mathcal{R}_{\mathcal{T}}^{adv-label}(h_{\mathbf{w}}, y_{\mathcal{T}}) \leq \mathcal{R}_{\mathcal{S}}^{adv-label}(h_{\mathbf{w}}, y_{\mathcal{S}}) + \mathcal{R}_{\mathcal{S}}^{adv-label}(h_{\mathbf{w}_{\mathcal{S}}^*}, y_{\mathcal{S}})$$
$$+ disc_{\mathcal{H}\Delta\mathcal{H}}^{adv}(\hat{\mathcal{T}}, \hat{\mathcal{S}}) + \mathcal{R}_{\mathcal{T}}^{adv}(h_{\mathbf{w}_{\mathcal{T}}^*}, h_{\mathbf{w}_{\mathcal{S}}^*}) + \mathcal{R}_{\mathcal{T}}^{adv-label}(h_{\mathbf{w}_{\mathcal{T}}^*}, y_{\mathcal{T}})$$
$$+ \hat{\mathfrak{R}}_{\mathcal{S}}(\tilde{\ell} \circ \mathcal{H}\Delta\mathcal{H}) + \hat{\mathfrak{R}}_{\mathcal{T}}(\tilde{\ell} \circ \mathcal{H}\Delta\mathcal{H}) + \left(3M\sqrt{\frac{\log(2/c)}{n_{\mathcal{S}}}} + 3M\sqrt{\frac{\log(2/c)}{n_{\mathcal{T}}}}\right).$$

The proof of Lemma 2 is deferred to Appendix E. Here we establish the relation between source adversarially robust risk and target adversarially robust risk. It shows that the adversarial discrepancy and adversarial complexity measure $\hat{\mathfrak{R}}_{\mathcal{D}}(\tilde{\ell} \circ \mathcal{H}\Delta\mathcal{H})$ on source and target domains are the two key quantities controlling the deviation between the model's performance on the two domains. Hence, to answer our previously proposed question, *why the robust risk is harder to adapt to different domain*, it is essential to study the connection between adversarial and non-adversarial Rademacher complexities over $\mathcal{H}\Delta\mathcal{H}$ class.

## 3 MAIN RESULTS

### 3.1 BINARY CLASSIFICATION SETTING

We start with the binary classification problem where the labels come from $\{-1, +1\}$. Like in Section 4.1 of Yin et al. (2019), we introduce the hypothesis class of linear functions with bounded weights:

$$\mathcal{H} := \{h_{\mathbf{w}} : \mathbf{x} \mapsto \langle \mathbf{w}, \mathbf{x} \rangle, \ \mathbf{w} \in \mathbb{R}^d : \|\mathbf{w}\|_p \leq W\} \ , \tag{5}$$

where $p \geq 1$. Moreover, we consider the following loss $\ell(h_{\mathbf{w}}(\mathbf{x}), y) := \phi(y h_{\mathbf{w}}(\mathbf{x}))$ where $\phi$ is a monotonic non-increasing and $L_\phi$-Lipschitz function. With such a loss $\phi$, the non-adversarial class of loss functions over $\mathcal{H}\Delta\mathcal{H}$ becomes

$$\ell \circ \mathcal{H}\Delta\mathcal{H} := \left\{\mathbf{x} \mapsto \ell(h_{\mathbf{w}}(\mathbf{x}), h_{\mathbf{w}'}(\mathbf{x})) := \phi\big(h_{\mathbf{w}}(\mathbf{x}) h_{\mathbf{w}'}(\mathbf{x})\big) \ : \ h_{\mathbf{w}}, h_{\mathbf{w}'} \in \mathcal{H}\right\} \ .$$

However, directly analyzing $\ell \circ \mathcal{H}\Delta\mathcal{H}$ class will be difficult since we do not assume the formula of $\phi$ explicitly. Hence, following Yin et al. (2019), let us define the following class of functions

$$f \circ \mathcal{H}\Delta\mathcal{H} := \left\{\mathbf{x} \mapsto h_{\mathbf{w}}(\mathbf{x}) h_{\mathbf{w}'}(\mathbf{x}) \ : \ h_{\mathbf{w}}, h_{\mathbf{w}'} \in \mathcal{H}\right\} \ .$$

We switch from the study of the Rademacher complexity defined in (1) over the function class introduced in (5), that is for linear classifiers applied to binary classification, to the following formula

$$\hat{\mathfrak{R}}_{\mathcal{D}}(f \circ \mathcal{H}\Delta\mathcal{H}) = \mathbb{E}_\sigma \left[\sup_{\mathbf{w}, \mathbf{w}' : \|\mathbf{w}\|_p \leq W, \|\mathbf{w}'\|_p \leq W} \frac{1}{n} \sum_{i=1}^n \sigma_i \mathbf{w}^\top \mathbf{x}_i \mathbf{w}'^\top \mathbf{x}_i\right] \ . \tag{6}$$

Indeed, by Ledoux-Talagrand contraction property (Ledoux & Talagrand (2013)) of Rademacher complexity we have that $\hat{\mathfrak{R}}_{\mathcal{D}}(\ell \circ \mathcal{H}\Delta\mathcal{H}) \leq L_\phi \hat{\mathfrak{R}}_{\mathcal{D}}(f \circ \mathcal{H}\Delta\mathcal{H})$. Thus, in the following lemma we aim at estimating $\hat{\mathfrak{R}}_{\mathcal{D}}(f \circ \mathcal{H}\Delta\mathcal{H})$.

**Lemma 3** (Rademacher complexity for binary classification under linear hypothesis). *Consider hypothesis class defined in (5). Assume that a set of data $\{\mathbf{x}_1, ..., \mathbf{x}_n\}$ are draw from $\mathcal{D}$. Let $\hat{\mathfrak{R}}(f \circ \mathcal{H} \Delta \mathcal{H})$ be defined as in (6). Then the following statement holds true for non-adversarial Rademacher complexity :*

$$\hat{\mathfrak{R}}_{\mathcal{D}}(f \circ \mathcal{H} \Delta \mathcal{H}) \leq \frac{W^2}{n} \left( \sqrt{2 \left\| \sum_{i=1}^{n} (\mathbf{x}_i \mathbf{x}_i^\top)^2 \right\|_2 \log(2d)} + \frac{\|\mathbf{X}\|_{2,\infty}^2 \log(2d)}{3} \right) \cdot \begin{cases} 1, & 1 \leq p \leq 2 \\ d^{1-2/p}, & p > 2 \end{cases},$$

*where $\mathbf{X} \in \mathbb{R}^{n \times d}$ is the data matrix and $i$-th row of $\mathbf{X}$ is $\mathbf{x}_i$.*

The proof of Lemma 3 is deferred to the Appendix G.1. Lemma 3 shows that the magnitude of the non-adversarial Rademacher over $\mathcal{H} \Delta \mathcal{H}$ class depends on the spectral norm of data covariance matrix. It implies that a more diverse dataset will result in a larger Rademacher complexity, and hence harder to perform domain adaptation. We notice that Kuroki et al. (2019) also gave an estimation of the upper bound of $\hat{\mathfrak{R}}_{\mathcal{D}}(f \circ \mathcal{H} \Delta \mathcal{H})$ in their Lemma 5, but our bound is superior to theirs in the following two aspectives: (1) Our bound is tighter in terms of the dependency on covariance matrix, since our bound depends on $\left\| \sum_{i=1}^{n} (\mathbf{x}_i \mathbf{x}_i^\top)^2 \right\|_2$ while their bound depends on $\sum_{i=1}^{n} \left\| (\mathbf{x}_i \mathbf{x}_i^\top)^2 \right\|_F^2$. (2) We consider that model capacity is controlled by $p$-norm while they only consider 2-norm.

Then, let us specify the class of functions involved in the definition of the adversarial Rademacher complexity of $\mathcal{H} \Delta \mathcal{H}$ in (3) as follows:

$$\tilde{\ell} \circ \mathcal{H} \Delta \mathcal{H} := \left\{ \max_{\|\boldsymbol{\delta}\|_\infty \leq \epsilon} \phi\big(h_{\mathbf{w}}(\mathbf{x}+\boldsymbol{\delta}) h_{\mathbf{w}'}(\mathbf{x}+\boldsymbol{\delta})\big) \ : \ h_{\mathbf{w}}, h_{\mathbf{w}'} \in \mathcal{H} \right\},$$

and let us define

$$\tilde{f} \circ \mathcal{H} \Delta \mathcal{H} := \left\{ \mathbf{x} \mapsto \min_{\|\boldsymbol{\delta}\|_\infty \leq \epsilon} h_{\mathbf{w}}(\mathbf{x}+\boldsymbol{\delta}) h_{\mathbf{w}'}(\mathbf{x}+\boldsymbol{\delta}) \ : \ h_{\mathbf{w}}, h_{\mathbf{w}'} \in \mathcal{H} \right\} . \tag{7}$$

With the above notations, we can characterize the adversarial counterpart of (6). Again by Ledoux-Talagrand's property, as in Yin et al. (2019); Awasthi et al. (2020), we get that $\hat{\mathfrak{R}}_{\mathcal{D}}(\tilde{\ell} \circ \mathcal{H} \Delta \mathcal{H}) \leq L_\phi \hat{\mathfrak{R}}_{\mathcal{D}}(\tilde{f} \circ \mathcal{H} \Delta \mathcal{H})$, where

$$\hat{\mathfrak{R}}_{\mathcal{D}}(\tilde{f} \circ \mathcal{H} \Delta \mathcal{H}) = \mathbb{E}_{\sigma} \left[ \sup_{\mathbf{w}, \mathbf{w}': \|\mathbf{w}\|_p \leq W, \|\mathbf{w}'\|_p \leq W} \frac{1}{n} \sum_{i=1}^{n} \sigma_i \min_{\|\boldsymbol{\delta}\|_\infty \leq \epsilon} \mathbf{w}^\top(\mathbf{x}_i+\boldsymbol{\delta}) \mathbf{w}'^\top(\mathbf{x}_i+\boldsymbol{\delta}) \right]. \tag{8}$$

**Theorem 1** (Adversarial Rademacher complexity for binary classification under linear hypothesis). *Consider hypothesis class defined in (5). Assume a set of data $\{\mathbf{x}_1, ..., \mathbf{x}_n\}$ are drawn from $\mathcal{D}$. Let $\hat{\mathfrak{R}}_{\mathcal{D}}(f \circ \mathcal{H} \Delta \mathcal{H})$ and $\hat{\mathfrak{R}}_{\mathcal{D}}(\tilde{f} \circ \mathcal{H} \Delta \mathcal{H})$ be defined as in (6) and (8), respectively. The following statement holds true for adversarial Rademacher complexity over $\mathcal{H} \Delta \mathcal{H}$ function class under linear hypothesis (5):*

$$\hat{\mathfrak{R}}_{\mathcal{D}}(\tilde{f} \circ \mathcal{H} \Delta \mathcal{H}) \leq \hat{\mathfrak{R}}_{\mathcal{D}}(f \circ \mathcal{H} \Delta \mathcal{H}) + 2\frac{W^2}{\sqrt{n}} \epsilon d^{1/p^*} \left( 1 + \sqrt{d}\sqrt{\log(3\sqrt{n})} \right) \left( \epsilon d^{1/p^*} + 2 \|\mathbf{X}\|_{p^*,\infty} \right),$$

*where $p^*$ is such that $1/p + 1/p^* = 1$, and $\mathbf{X} \in \mathbb{R}^{n \times d}$ is the data matrix and $i$-th row of $\mathbf{X}$ is $\mathbf{x}_i$. Moreover, the following lower bound also holds:*

$$\hat{\mathfrak{R}}_{\mathcal{D}}(\tilde{f} \circ \mathcal{H} \Delta \mathcal{H}) \geq \hat{\mathfrak{R}}_{\mathcal{D}}(f \circ \mathcal{H} \Delta \mathcal{H}) + \begin{cases} 0, & 1 \leq p \leq 2 \\ \frac{W^2}{n}(1 - d^{1-2/p})\mathbb{E}_\sigma \left\| \sum_{i=1}^{n} \sigma_i \mathbf{x}_i \mathbf{x}_i^\top \right\|_2, & p > 2 \end{cases} . \tag{9}$$

The proofs for Theorem 1 are deferred to Appendices G.2 and G.3. From (1), we notice that the upper bound of $\hat{\mathfrak{R}}_{\mathcal{D}}(\tilde{f} \circ \mathcal{H} \Delta \mathcal{H})$ has the smallest dependence in model dimension $d$ if the weights are constrained by the $\ell_1$-norm ($p = 1$). This is a similar observation as in Yin et al. (2019) where they consider single domain setting. However, in their single domain setting, when $p = 1$, adversarial Rademacher complexity is dimension free while we still have $\sqrt{d}$ dependency. This heavier dependence is likely due to the fact that $\hat{\mathfrak{R}}_{\mathcal{D}}(\tilde{f} \circ \mathcal{H} \Delta \mathcal{H})$ is defined by coupling two models and hence enlarges the complexity. The bound achieves the sublinear convergence $\mathcal{O}(1/\sqrt{n})$ over the number

of samples $n$, and quadratic dependence on maximum model weight $W$. It implies that models with suppressed norm can help adversarially robust domain adaptation since it reduces the Rademacher complexity, as we will see in the experiments.

The lower bound result in (9) shows that, adversarial Rademacher complexity over $\mathcal{H}\Delta\mathcal{H}$ will be always larger than non-adversarial one, which implies that adversarial robust domain adaptation is **at least as hard as non-adversarial domain adaptation**, and that is why, as we will also see in our experiments, given a model, the gap between its source domain robust risk and target domain robust risk is usually larger than that in terms of standard risk. Moreover, the gap between adversarial and non-adversarial complexity is controlled by the spectral norm of Rademacher variable induced covariance matrix. This dependence reveals that a more diverse dataset would be harder to transfer robustness, compared to the standard domain adaptation.

## 3.2 LINEAR REGRESSION SETTING

In this section we consider linear regression problems. The hypothesis class of linear functions with bounded weights remains the same as in (5). However, we consider the following class of quadratic loss functions:

$$\ell \circ \mathcal{H}\Delta\mathcal{H} := \left\{ \mathbf{x} \mapsto \ell(h_{\mathbf{w}}(\mathbf{x}), h_{\mathbf{w}'}(\mathbf{x})) := (h_{\mathbf{w}}(\mathbf{x}) - h_{\mathbf{w}'}(\mathbf{x}))^2,\ h_{\mathbf{w}}, h_{\mathbf{w}'} \in \mathcal{H} \right\} . \tag{10}$$

The following lemma establishes the upper bound of non-adversarial Rademacher complexity over $\mathcal{H}\Delta\mathcal{H}$ in the above setting.

**Lemma 4** (Rademacher complexity for regression under linear hypothesis). *Let $\mathcal{H}$ be the set of linear functions with bounded weights as defined in* (5). *Then the following statement holds true for non-adversarial Rademacher complexity over $\mathcal{H}\Delta\mathcal{H}$ class:*

$$\hat{\mathfrak{R}}_{\mathcal{D}}(\ell \circ \mathcal{H}\Delta\mathcal{H}) \leq \frac{4W^2}{n} \left( \sqrt{2 \left\| \sum_{i=1}^{n} (\mathbf{x}_i \mathbf{x}_i^\top)^2 \right\|_2 \log(2d)} + \frac{1}{3} \|\mathbf{X}\|_{2,\infty}^2 \log(2d) \right) \cdot \begin{cases} 1 & 1 \leq p \leq 2 \\ d^{1-2/p} & p > 2 \end{cases},$$

*where $\mathbf{X} \in \mathbb{R}^{n \times d}$ is the data matrix and $i$-th row of $\mathbf{X}$ is $\mathbf{x}_i$.*

The proof of Lemma 4 is deferred to Appendix H.1. As in the binary classification case presented in Section 3.1, we are able to relate the non-adversarial Rademacher complexity over $\mathcal{H}\Delta\mathcal{H}$ class to the spectral norm of data covariance matrix.

**Theorem 2** (Adversarial Rademacher complexity for regression under linear hypothesis). *Let $\mathcal{H}$ be the set of linear functions with bounded weights as defined in* (5). *Then the following statement holds true for adversarial Rademacher complexity over $\mathcal{H}\Delta\mathcal{H}$ function class:*

$$\hat{\mathfrak{R}}_{\mathcal{D}}(\tilde{\ell} \circ \mathcal{H}\Delta\mathcal{H}) \leq \hat{\mathfrak{R}}_{\mathcal{D}}(\ell \circ \mathcal{H}\Delta\mathcal{H})$$

$$+ 4\frac{W^2}{\sqrt{n}} \left( 2\sqrt{d}\epsilon \|\mathbf{X}\|_{2,\infty} + d\epsilon^2 \right) \left( \sqrt{2d \log(6\sqrt{n})} + 1 \right) \cdot \begin{cases} 1, & 1 \leq p \leq 2 \\ d^{1-2/p}, & p > 2 \end{cases},$$

*where $\mathbf{X} \in \mathbb{R}^{n \times d}$ is the data matrix and $i$-th row of $\mathbf{X}$ is $\mathbf{x}_i$. Meanwhile, the following lower bound holds as well:*

$$\hat{\mathfrak{R}}_{\mathcal{D}}(\tilde{\ell} \circ \mathcal{H}\Delta\mathcal{H}) \geq \hat{\mathfrak{R}}_{\mathcal{D}}(\ell \circ \mathcal{H}\Delta\mathcal{H}) + \begin{cases} 0, & 1 \leq p \leq 2 \\ \frac{4W^2}{n}(1 - d^{1-2/p})\mathbb{E}_{\sigma} \left\| \sum_{i=1}^{n} \sigma_i \mathbf{x}_i \mathbf{x}_i^\top \right\|_2, & p > 2 \end{cases}.$$

The proof of Theorem 2 is deferred to Appendices H.2 and H.3. Few comments can be made concerning the above theorem. First, the upper bound of adversarial Rademacher complexity also depends quadratically on $W$ and adversarial budget $\epsilon$, and super-linearly on model dimension $d$. Second, for the lower bound, we established the similar gap between $\hat{\mathfrak{R}}_{\mathcal{D}}(\tilde{\ell} \circ \mathcal{H}\Delta\mathcal{H})$ and $\hat{\mathfrak{R}}_{\mathcal{D}}(\ell \circ \mathcal{H}\Delta\mathcal{H})$ as in classification setting, which means the data diversity also affects hardness of adversarially robust domain adaptation in regression setting.

## 4 EXTENSION TO NEURAL NETWORKS WITH ReLU ACTIVATION

We next extend our analysis methods to more complicated neural network function class. In this section, we will present our results for two-layer ReLU neural networks. That is, we consider the following hypothesis class

$$\mathcal{H} := \{h_{\mathbf{w}} : \mathbf{x} \mapsto \mathbf{a}^\top \mathrm{ReLU}(\mathbf{W}\mathbf{x}), \, \mathbf{a} \in \mathbb{R}^d, \mathbf{W} \in \mathbb{R}^{d \times d} : \|\mathbf{a}\|_p \leq A, \|\mathbf{W}\|_p \leq W\}. \tag{11}$$

The following theorem establishes the relation between Adversarial Rademacher complexity and non-adversarial version in classification setting. As in Section 3.1, we consider the same loss functions of the form $\ell(h_{\mathbf{w}}(\mathbf{x}), y) := \phi(y h_{\mathbf{w}}(\mathbf{x}))$ in classification setting, and $\ell_2$ loss in regression setting.

**Theorem 3** (Adversarial Rademacher complexity on ReLU neural network class). *Let $\mathcal{H}$ be the set of two-layer ReLU neural networks with bounded weights as defined in* (11). *Then, the following statement holds true for adversarial Rademacher complexity over $\mathcal{H} \Delta \mathcal{H}$ function class, with classification loss*

$$\hat{\mathfrak{R}}_{\mathcal{D}}(\tilde{f} \circ \mathcal{H} \Delta \mathcal{H}) \leq \mathfrak{R}_{\mathcal{D}}(f \circ \mathcal{H} \Delta \mathcal{H})$$

$$+ \frac{A^2 W^2}{n} \epsilon \sqrt{d \log(2d)} \left( \sqrt{2 \sum_{i=1}^n (\sqrt{d}\epsilon + 2\|\mathbf{x}_i\|_2)^2} + \frac{1}{3} \sqrt{\log(2d)}(\sqrt{d}\epsilon + 2\|\mathbf{X}\|_{2,\infty}) \right) \cdot \begin{cases} 1, & 1 \leq p \leq 2 \\ d^{2-4/p}, & p > 2 \end{cases}.$$

*Similarly the following statement holds true for adversarial Rademacher complexity over $\mathcal{H} \Delta \mathcal{H}$ function class with $\ell_2$ loss:*

$$\hat{\mathfrak{R}}_{\mathcal{D}}(\tilde{\ell} \circ \mathcal{H} \Delta \mathcal{H}) \leq \hat{\mathfrak{R}}_{\mathcal{D}}(\ell \circ \mathcal{H} \Delta \mathcal{H})$$

$$+ \frac{A^2}{n} W^2 \epsilon \sqrt{d \log(2d)} \left( 3 \sqrt{2 \sum_{i=1}^n (\sqrt{d}\epsilon + 2\|\mathbf{x}_i\|_2)^2} + \sqrt{\log(2d)}(\sqrt{d}\epsilon + 2\|\mathbf{X}\|_{2,\infty}) \right) \cdot \begin{cases} 1, & 1 \leq p \leq 2 \\ d^{2-4/p}, & p > 2 \end{cases}.$$

The proof of Theorem 3 is deferred to Appendices I.1 and I.2. As we can see from the above theorem, we get the similar upper bound for ReLU neural network class to what we showed in the linear model case. The upper bound of adversarial Rademacher can be bounded by non-adversarial version plus terms depending on the norm of each layer, and the norm of data points. We would like to mention that, unlike the linear case where we show that the gap between adversarial and non-adversarial Rademacher complexity is lower bounded, here we do not establish such lower bound, due to the difficulty of analyzing ReLU unit.

## 5 ADVERSARIAL TRAINING HELPS TRANSFER TO DIFFERENT DOMAIN

Here we discuss the connection between standard ERM learning and adversarially robust learning. As observed by prior works Salman et al. (2020); Deng et al. (2021), if a model is adversarially trained on the source domain, then its standard accuracy on target domain is sometimes better than if it had been fitted via vanilla ERM on source domain. In this section, we try to explain this phenomena from adversarially robust domain adaptation perspective. We found that, when adversarial budget is large enough, *small source adversarial risk almost guarantees the small target domain standard risk*. First, we need to introduce the following optimization problem.

**Definition 5.** *Let $\mathbf{p}$ and $\mathbf{p}'$ be two vectors on the $N$-dimensional simplex and let $\boldsymbol{\ell}$ be a 0-1 vector. Let also $\Lambda$ be an arbitrary subset of $[N] := \{1, \ldots, N\}$. The* Subset Sum Problem with Structural Objective *can be defined as solving the following combinatorial optimization problem:*

$$\begin{cases} \min_{\tilde{\boldsymbol{\ell}} \in \{0,1\}^N} & \left| \mathbf{p}^\top \tilde{\boldsymbol{\ell}} - \mathbf{p}'^\top \boldsymbol{\ell} \right| \\ \quad s.t. & \tilde{\ell}_i = \ell_i, \, \forall \, i \in [N] \setminus \Lambda \end{cases}.$$

*We denote its optimal value as $V^*(\mathbf{p}', \mathbf{p}, \boldsymbol{\ell}, \Lambda)$.*

The above problem is a variant of Subset Sum Problem Hartmanis (1982), which is also *NP-complete*. We look for a subset of coordinates of a simplex vector $\mathbf{p}$, such that their sum is closest

to a given goal. The given goal has special structure: it is defined as sum of a subset of coordinates in another simplex vector $\mathbf{p}'$. If the constraint set $\Lambda_\epsilon$ has more indices, the optimal value will be smaller since we can determine the value on more coordinates of $\tilde{\ell}$. In the following lemma, we explain how this combinatorial measure helps us to connect adversarially robust and standard risks for the binary classification task.

**Lemma 5.** *Consider binary classification task, with sign linear classifier class $\mathcal{H} = \{h_\mathbf{w} : h_\mathbf{w} = \mathrm{sign}(\mathbf{w}^\top \mathbf{x}), \|\mathbf{w}\|_p \leq W\}$ and 0-1 loss function $\phi(x,y) = \frac{1}{2}|x - y|$. Assume all domains share the same labeling function $y(\mathbf{x}) \in \{-1, 1\}$. The following statement holds for any $\mathcal{T}'$:*

$$\mathcal{R}_{\mathcal{T}'}(h_\mathbf{w}, y) \leq \mathcal{R}_{\mathcal{T}}^{adv-label}(h_\mathbf{w}, y) + V^*(\mathbf{p}', \mathbf{p}, \boldsymbol{\ell}, \Lambda_\epsilon),$$

*where $\boldsymbol{\ell}$ is the loss vector such that $\ell_i = \frac{1}{2}|\mathrm{sign}(\mathbf{w}^\top \mathbf{x}_i) - y(\mathbf{x}_i)|$, with $\mathbf{x}_i \in \mathcal{X}$. The vectors $\mathbf{p}$, $\mathbf{p}'$ are probability mass vectors of $\mathcal{T}$ and $\mathcal{T}'$, i.e., $\mathbf{p}(\mathbf{x}) = \mathbb{P}_{X \sim \mathcal{T}}(X = \mathbf{x}), \mathbf{x} \in \mathcal{X}$ and $\mathbf{p}'(\mathbf{x}) = \mathbb{P}_{X \sim \mathcal{T}'}(X = \mathbf{x}), \mathbf{x} \in \mathcal{X}$ Moreover, $\Lambda_\epsilon = \{i : |\mathbf{w}^\top \mathbf{x}_i| \leq \epsilon \|\mathbf{w}\|_1, \mathbf{x}_i \in \mathcal{X}, \forall \mathbf{w}, \|\mathbf{w}\|_p \leq W\}$.*

The corresponding proof is given in Appendix J. Lemma 5 shows that, the standard risk on domain $\mathcal{T}'$ can be bounded by robust risk on domain $\mathcal{T}$, plus the quantity controlled by $\epsilon$. Since the set $\Lambda_\epsilon$ stores all indices $i$ such that we can choose to flip $\tilde{\ell}_i$'s value between 0 and 1, then if we have larger $\epsilon$, there will be more indices in $\Lambda_\epsilon$, which means there are more coordinates in $\tilde{\ell}$ we can play with, and hence smaller value of $V^*$.

## 6 EMPIRICAL RESULTS

In this section, we verify the theoretical implications through empirical studies on a multi-domain dataset, DIGITS Ganin & Lempitsky (2015). DIGITS has $28 \times 28$ images and includes 5 different domains: MNIST (Lecun et al., 1998), SVHN (Netzer et al., 2011), USPS (Hull, 1994), SynthDigits (Ganin & Lempitsky, 2015), and MNIST-M (Ganin & Lempitsky, 2015). All domain datasets are subsampled to contain 7438 images to eliminate the effect of number of samples in generalization.

Given a model $h_\mathbf{w}$ parameterized by $\mathbf{w}$, we consider two training methods:

$$\textit{Adversarial Training}: \ \min_\mathbf{w} \frac{1}{n} \sum_{i=1}^n \max_{\|\boldsymbol{\delta}\|_\infty \leq \epsilon} \ell(h_\mathbf{w}(\mathbf{x}_i + \boldsymbol{\delta}), y(\mathbf{x}_i)), \tag{12}$$

$$\textit{Standard Training}: \ \min_\mathbf{w} \frac{1}{n} \sum_{i=1}^n \ell(h_\mathbf{w}(\mathbf{x}_i), y(\mathbf{x}_i)). \tag{13}$$

To solve the inner maximization in (12), we leverage $k$-step PGD (projected gradient descent) attack (Madry et al., 2018) with a constant noise magnitude $\epsilon$. Following (Madry et al., 2018), we use $\epsilon = 8/255$, $k = 7$, and attack inner-loop step size $2/255$, for training, and adversarial test. Then we use Adam to minimize the losses with 100 epochs and learning rate of $10^{-2}$ decaying in a cosine manner. We evaluate the model performance by: (1) standard accuracy (SA): classification accuracy on the clean test set; and (2) robust accuracy (RA): classification accuracy on adversarial images perturbed from the original test set.

**How does adversarial robustness transfer arcoss domains?** In this experiments, we use a convolutional network whose architecture is elaborated in Appendix K. We report the transfer accuracy in Table 1 where models are trained on source domain (first column in each row) and tested on different target domains (the rest columns), as well as the difference between source SA/RA and target SA/RA. The experiment has the following implications: (1) We observe that transfer difference $\Delta$ is more significant on RA than SA. For example, for the model trained on MNIST dataset, no matter trained standardly or adversarially, their testing RAs on all other domains drop dramatically than SAs. It implies that adversarially robust domain adaptation is harder than standard domain adaptation, as illustrated by Theorem 1. (2) The models trained on complicated dataset may gain higher robust accuracy at simple dataset, e.g., SVHN $\to$ {MNIST, SynthDigits and USPS} and MNIST-M $\to$ MNIST. The increase can be attributed to that the source domain has more complicated features and thus more robust features are learnt. However, exploring the reason behind this interesting phenomena is beyond the scope of this paper.

Table 1: Transferred standard (SA %) and robust (RA %) accuracies tested on different domains from the DIGITS datasets. $\Delta$ indicates the difference between the train and test domain accuracy.

| Source | Target | MNIST SA | RA | MNIST-M SA | RA | SVHN SA | RA | SynthDigits SA | RA | USPS SA | RA |
|---|---|---|---|---|---|---|---|---|---|---|---|
| | | **Standardly-trained models** | | | | | | | | | |
| MNIST | Acc | 98.8 | 95.9 | 34.7 | 15.3 | 16.0 | 5.9 | 25.0 | 7.8 | 49.9 | 27.9 |
| | $\Delta$ | +0.0 | +0.0 | −64.1 | −**80.6** | −82.8 | −**90.1** | −73.7 | −**88.2** | −48.9 | −**68.0** |
| MNIST-M | Acc | 97.2 | 76.7 | 94.1 | 28.5 | 33.9 | 0.0 | 49.1 | 1.7 | 63.6 | 5.3 |
| | $\Delta$ | +3.1 | +**48.2** | +0.0 | +0.0 | −**60.2** | −28.5 | −**45.0** | −26.8 | −**30.5** | −23.2 |
| SVHN | Acc | 59.6 | 32.7 | 47.2 | 5.2 | 87.5 | 6.0 | 84.3 | 28.4 | 64.5 | 22.5 |
| | $\Delta$ | −**27.9** | +26.7 | −**40.3** | −0.8 | +0.0 | +0.0 | −3.2 | +**22.4** | −**22.9** | +16.5 |
| SynthDigits | Acc | 83.6 | 57.0 | 57.9 | 9.1 | 73.0 | 3.8 | 96.1 | 59.8 | 82.7 | 40.4 |
| | $\Delta$ | −**12.5** | −2.8 | −38.2 | −**50.7** | −23.2 | −**56.1** | +0.0 | +0.0 | −13.4 | −**19.4** |
| USPS | Acc | 67.0 | 54.3 | 25.8 | 13.1 | 9.6 | 5.1 | 31.2 | 13.1 | 98.7 | 94.1 |
| | $\Delta$ | −31.7 | −**39.8** | −73.0 | −**80.9** | −89.2 | −89.0 | −67.6 | −**81.0** | +0.0 | +0.0 |
| | | **Adversarially-trained models** | | | | | | | | | |
| MNIST | Acc | 99.0 | 98.3 | 49.5 | 31.9 | 19.4 | 14.6 | 32.2 | 17.3 | 59.7 | 38.8 |
| | $\Delta$ | +0.0 | +0.0 | −49.5 | −**66.4** | −79.6 | −**83.7** | −66.9 | −**81.0** | −39.4 | −**59.5** |
| MNIST-M | Acc | 96.9 | 94.5 | 93.0 | 76.8 | 26.9 | 11.5 | 46.4 | 25.4 | 66.5 | 46.8 |
| | $\Delta$ | +4.0 | +**17.7** | +0.0 | +0.0 | −**66.1** | −65.3 | −46.5 | −**51.4** | −26.5 | −**30.0** |
| SVHN | Acc | 56.2 | 46.6 | 43.3 | 18.0 | 76.2 | 42.6 | 78.9 | 60.2 | 66.8 | 51.3 |
| | $\Delta$ | −**20.0** | +4.0 | −**32.9** | −24.6 | +0.0 | +0.0 | +2.7 | +**17.6** | −**9.3** | +8.7 |
| SynthDigits | Acc | 84.9 | 75.6 | 58.0 | 25.8 | 64.1 | 17.9 | 95.6 | 84.8 | 82.6 | 64.8 |
| | $\Delta$ | −**10.6** | −9.2 | −37.6 | −**59.1** | −31.5 | −**66.9** | +0.0 | +0.0 | −13.0 | −**20.0** |
| USPS | Acc | 72.3 | 65.6 | 24.1 | 15.4 | 9.7 | 5.3 | 30.1 | 16.8 | 98.9 | 97.5 |
| | $\Delta$ | −26.6 | −**31.9** | −74.8 | −**82.2** | −89.1 | −**92.3** | −68.8 | −**80.8** | +0.0 | +0.0 |

**Adversarial training helps domain adaptation.** We can see from Table 1 that, sometimes when models are adversarial trained on simple dataset (e.g., MNIST and SynthDigits dataset), it is noticeable the standard accuracy on other datasets are improved. For example, if we do adversarial training on MNIST dataset, we achieve significantly higher SA on other dataset, than standard trained model on MNIST. The same phenomena happens when we choose SynthDigits or USPS as source domain. Such advantages are consistent with our Corollary 3.

**Does $\ell_1$ regularization help adversarial transfer?** Our Theorem 1 shows that the adversarial Rademacher complexity over $\mathcal{H}\Delta\mathcal{H}$ class is suppressed the most when the $\ell_1$-norm of the model parameters is controlled. To empirically investigate the relation, we consider a linear

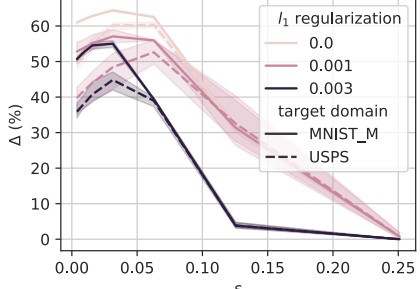

Figure 1: Robust accuracy drops ($\Delta$) by varying the $\ell_1$ regularization intensity ($\mu$) and $\ell_\infty$ perturbation $\epsilon$. A linear classifier is adversarially trained on the MNIST and tested on target domains.

model on vectorized images $\mathbf{x}$ and solve the following optimization problem with $\ell_1$ regularization: $\min_{\mathbf{w}} \frac{1}{n} \sum_{i=1}^{n} \max_{\|\boldsymbol{\delta}\|_\infty \le \epsilon} \ell(h_{\mathbf{w}}(\mathbf{x}_i + \boldsymbol{\delta}), y(\mathbf{x}_i)) + \mu \|\mathbf{w}\|_1$, where $\mu \ge 0$ is the regularization intensity. In Figure 1, we present the drops of robust accuracy from source domain to target domain $\mathrm{RA}_S - \mathrm{RA}_T$, regarding the intensity of $\ell_1$ regularization. Consistent with our theoretical results, increasing $\ell_1$ regularization ($\mu > 0$) can reduce the transfer accuracy drops on different level of adversarial attacks.

## 7 CONCLUSION

In this paper we propose and analyze the adversarial Rademacher complexity over $\mathcal{H}\Delta\mathcal{H}$ class, which is proven to be key factor that controls the generalization of adversarially robust risk to different domains. We theoretically explain that why adversarial domain adaptation is harder than different domain than standard domain adaptation. We also characterize the standard accuracy of a given model on any target domain, using its adversarial accuracy on the source domain, which matches with the recent observation regarding the superiority of adversarially training.

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

## A    MORE RELATED WORK

Here, we briefly discuss some relevant prior works.

**Discrepancy Based Domain Adaptation Theory**    A significant category of the domain adaptation study is discrepancy based generalization analysis. Ben-David et al. (2006) borrowed the $\mathcal{A}$-discrepancy from seminal work Kifer et al. (2004), and gave the target domain generalization in terms of source domain error and this discrepancy measure. Afterwards, Ben-David et al. (2010) proposed $\mathcal{H}\Delta\mathcal{H}$ discrepancy, which is easier to estimate from unlabeld data, and also proved VC dimenson based generalization bound. Mansour et al. (2009) also consider $\mathcal{H}\Delta\mathcal{H}$ discrepancy, while their analysis depends on Rademacher complexity over $\mathcal{H}\Delta\mathcal{H}$ function class. They claim that in some situation, their learning bound is superior to Ben-David et al. (2010)'s bound. Mohri & Muñoz Medina (2012) proposed $\mathcal{Y}$-discrepancy which is a labeling function dependent measure, but hence it cannot be estimated from unlabeled data. Kuroki et al. (2019) advocated a source-guided discrepancy and showed that it is a tighter discrepancy measure than $\mathcal{H}\Delta\mathcal{H}$ discrepancy. Zhang et al. (2020) proposed localized discrepancy measure, where they argued that when defining a discrepancy measure, considering the whole hypothesis class may be too pessimistic, so they chose to incorporate risk level as well into the discrepancy definition.

**Generalization of Adversarially Robust Learning**    To characterize the generalization of adversarially robust learning, a line of researches Khim & Loh (2018); Yin et al. (2019); Awasthi et al. (2020) are conducted via Rademacher complexity point of view. Khim & Loh (2018) is among the first to examine the adversarial Rademacher complexity under $\ell_\infty$ attack, and as a concurrent work, Yin et al. (2019) characterized the upper and lower bound of it, and claim that adversarially robust is at lease as hard as standard ERM learning. Awasthi et al. (2020) further extended Yin et al. (2019)'s results to adversary set under arbitrary norm constraint, and analyze the complexity of neural network as well. Another category of generalization studies of robust learning are based on PAC learning framework. Cullina et al. (2018) proved that empirical robust risk minimization is a successful robust PAC learner. Montasser et al. (2019) show that, the function classes with finite VC dimension are adversarially robustly PAC learnable, with the sample complexity related to dual VC dimension, which could be exponentially larger than vanilla VC dimension. Diochnos et al. (2019) proved the lower sample complexity bound for robust PAC learning under hybrid attack. They show that a sample complexity exponetially in the adversary budget is unavoidable. Gourdeau et al. (2021) also studied the hardness of robust classification under PAC learning framework, and proved some impossibility results regarding the adversary budget. Diochnos et al. (2018) investigated different adversarial risk definitions, and proved negative results on the uniform distribution. Pydi & Jog (2022) also analyzed the existing adversarial risk notions, and discovered the difference and connections among them.

**Robustness Transfer**    Robustness transfer is a newly initiated research area. Shafahi et al. (2019) discovered that by fine-tuning the network on target domain, the robustness can be inherited by the new model. Hong et al. (2021) considered the federated learning scenario, where they wish to transfer robust model from computationally rich users to users that cannot afford adversarial training. They proposed a batch-normalization based method to share robust among different clients. Fan et al. (2021) studied when the robust features learnt in contrastive learning can be transferred to different tasks.

## B    DIFFERENCE BETWEEN ROBUST LEARNING, STANDARD DOMAIN ADAPTATION AND ADVERSARIALLY ROBUST DOMAIN ADAPTATION

Here we briefly discuss the difference between the three relevant learning scenario: adversarially robust learning, standard domain adaptation and adversarially robust domain adaptation. In the goal of adversarially robust learning, we care about the gap between population robust risk and empirical robust risk, on the same domain; While in standard domain adaptation, we consider the gap between the (standard) risk on the target domain and the risk on the source domain. In adversarially robust domain adaptation, we care about the relation between adversarially robust risks on target and source domain, respectively.

## C  TECHNICAL NOVELTY

In this section we explain our technical novelty compared to existing works regard Adversarial Rademacher complexity Yin et al. (2019); Awasthi et al. (2020). Taking classification setting for example, Yin et al. (2019); Awasthi et al. (2020) consider the Rademacher complexity over the loss class between model predictions and labels, i.e., $\mathfrak{R} = \mathbb{E}[\sup_{\mathbf{w}} \frac{1}{n} \sum_{i=1}^{n} \sigma_i \min_{\|\boldsymbol{\delta}\| \leq \infty} \mathbf{w}^{\top}(\mathbf{x} + \boldsymbol{\delta})]$, where the inner minimization problem is **linear** in $\mathbf{w}$ and $\boldsymbol{\delta}$. We consider the loss between predictions among two models, hence the Rademacher complexity is $\mathfrak{R} = \mathbb{E}[\sup_{\mathbf{w},\mathbf{w}'} \frac{1}{n} \sum_{i=1}^{n} \sigma_i \min_{\|\boldsymbol{\delta}\| \leq \infty} \mathbf{w}^{\top}(\mathbf{x} + \boldsymbol{\delta}){\mathbf{w}'}^{\top}(\mathbf{x} + \boldsymbol{\delta})]$, where the inner problem is quadratic in terms of $\mathbf{w}$ and $\boldsymbol{\delta}$. Hence, unlike previous works on single domain Rademacher complexity, where the inner problem has simple closed form solution, we have to use $\varepsilon$-nets and covering number idea to prove upper bound. Our another key technical contribution is in the proof of lower bound results. For lower bound proofs, controlling the magnitude of Rademacher complexity with the inner problem being quadratic objective is significantly harder than linear objective. We derive the (complicated) closed form solution to inner quadratic programming, and leverage the symmetric property of Rademacher random variables to avoid heavy computation.

## D  USEFUL LEMMAS

In this section, we present necessary lemmas that are used further in the proof of our main results.

### D.1  MATRIX CONCENTRATION INEQUALITY

**Theorem 4** (Matrix Bernstein inequality, Thm 6.1.1 of Tropp et al. (2015)). *Let us denote by $\|.\|_2$ the spectral norm of matrix. Consider a finite sequence of $n$ independent, random matrices $\mathbf{Z}_i$ with common dimension $d_1 \times d_2$. Assume that*

$$\mathbb{E}[\mathbf{Z}_i] = 0 \quad and \quad \|\mathbf{Z}_i\|_2 \leq L, \quad \forall i \in [n] \ .$$

*Let $\mathbf{Y} := \sum_{i=1}^{n} \mathbf{Z}_i$. Then,*

$$\mathbb{E}[\|\mathbf{Y}\|_2] = \mathbb{E}\left[\left\|\sum_{i=1}^{n} \mathbf{Z}_i\right\|_2\right] \leq \sqrt{2 \operatorname{Var}(\mathbf{Y}) \log(d_1 + d_2)} + \frac{1}{3} L \log(d_1 + d_2) \ , \qquad (14)$$

*where the matrix variance is given by*

$$\operatorname{Var}(\mathbf{Y}) := \max\left\{\left\|\mathbb{E}[\mathbf{Y}\mathbf{Y}^{\top}]\right\|_2, \left\|\mathbb{E}[\mathbf{Y}^{\top}\mathbf{Y}]\right\|_2\right\}$$

$$= \max\left\{\left\|\sum_{i=1}^{n} \mathbb{E}[\mathbf{Z}_i\mathbf{Z}_i^{\top}]\right\|_2, \left\|\sum_{i=1}^{n} \mathbb{E}[\mathbf{Z}_i^{\top}\mathbf{Z}_i]\right\|_2\right\} \ .$$

### D.2  BASIC LEMMAS

**Lemma 6** (Basic squared norm inequality). *For any vector $a, b$, we have that $\|a - b\|_2^2 \leq 2\|a\|_2^2 + 2\|b\|_2^2$.*

**Lemma 7** (Hölder inequality). *Let $p \in \mathbb{R}$ such that $1 < p < \infty$. Let $p^*$ be its conjugate, that is if $1 < p < \infty$, $p^*$ is such that $\frac{1}{p} + \frac{1}{p^*} = 1$. Let $\mathbf{v}, \mathbf{w} \in \mathbb{R}^d$, then the following inequality holds*

$$|\langle \mathbf{v}, \mathbf{w} \rangle| \leq \|\mathbf{v}\|_p \|\mathbf{w}\|_{p^*} \ .$$

*If $p = 1$, we set $p^* = \infty$.*

**Lemma 8** (Equivalence of $p$-norms). *Let $q > p \geq 1$, then for all $\mathbf{v} \in \mathbb{R}^d$ we have $\|\mathbf{v}\|_q \leq \|\mathbf{v}\|_p \leq d^{1/p-1/q}\|\mathbf{v}\|_q$. It also holds for $q = \infty$, that is $\|\mathbf{v}\|_\infty \leq \|\mathbf{v}\|_p \leq d^{1/p}\|\mathbf{v}\|_\infty$.*

**Lemma 9** (Maximum dot product over $\ell_\infty$ ball). *The $\ell_\infty$ and $\ell_1$ norms are duals of each other. That is:*

$$\max_{\|x\|_\infty \leq \epsilon} z^\top x = \epsilon \|z\|_1 \quad , \tag{15}$$

*and this maximum is attained for $x^* = \epsilon \mathsf{sgn}(z)$, where $\mathsf{sgn}$ denotes the element-wise sign function.*

*Proof.* Hölder inequality implies that for all $z, x \in \mathbb{R}^p$,

$$z^\top x \leq |z^\top x| \leq \|x\|_\infty \|z\|_1 \leq \epsilon \|z\|_1 \quad .$$

Finally, we notice this upper bound is reached for $x^* = \epsilon \mathsf{sgn}(z)$ as

$$z^\top x^* = \epsilon z^\top \mathsf{sgn}(z) = \epsilon \sum_{i=1}^p z_i \mathsf{sgn}(z_i) = \epsilon \|z\|_1 \quad .$$

$\square$

**Lemma 10** (Minimum dot product over $\ell_\infty$ ball). *Let $z \in \mathbb{R}^d$, the solution of*

$$\min_{\|x\|_\infty \leq \epsilon} z^\top x = -\epsilon \|z\|_1 \quad , \tag{16}$$

*is attained at $x^* = -\epsilon \mathsf{sgn}(z)$.*

*Proof.* Same reasoning as in Lemma 9. $\square$

**Lemma 11** (Lower bound of minimum "quadratic" form over $\ell_\infty$ ball). *Let $z \in \mathbb{R}^d$, the solution of*

$$\min_{\|x\|_\infty \leq \epsilon} \langle u, x \rangle \langle v, x \rangle \geq -\epsilon^2 \|u\|_1 \|v\|_1 \quad , \tag{17}$$

*Proof.* Let $x \in B_\infty(0_d, \epsilon)$ which denotes the $\ell_\infty$ centered ball of radius $\epsilon > 0$. Hölder's inequality for dual norms applied twice gives us:

$$|\langle u, x \rangle \langle v, x \rangle| \leq \|x\|_\infty^2 \|u\|_1 \|v\|_1 \leq \epsilon^2 \|u\|_1 \|v\|_1 \quad ,$$

which directly implies that

$$\min_{\|x\|_\infty \leq \epsilon} \langle u, x \rangle \langle v, x \rangle \geq -\epsilon^2 \|u\|_1 \|v\|_1 \quad .$$

$\square$

**Remark 1.** *We make several comments on the above Lemma 11:*

- *Note that if $v = u$, then the objective becomes positive and $0$ is a simpler and sharp lower bound as it is reached for $x = 0_d$.*

- *Else if $v = -u$, then applying Lemma 12 implies that the minimum is reached and equals $-\epsilon^2 \|u\|_1^2$ which means the lower bound in (17) is sharp.*

- *Else if $v \perp u$, then one should be able to prove that the minimum is reached at something like $x^* = \epsilon \frac{u-v}{\|u-v\|_\infty}$, which correspond to an objective equaling: $\langle u, x^* \rangle \langle v, x^* \rangle = -\epsilon^2 \frac{\|u\|_2^2 \|v\|_2^2}{\|u-v\|_\infty^2}$.*

**Lemma 12** (Maximum squared dot product over $\ell_\infty$ ball). *We have that*

$$\max_{\|x\|_\infty \leq \epsilon} (z^\top x)^2 = \epsilon^2 \|z\|_1^2 \quad , \tag{18}$$

*and this maximum is attained for $x \in \{\epsilon \mathsf{sgn}(z), -\epsilon \mathsf{sgn}(z)\}$, where $\mathsf{sgn}$ denotes the element-wise sign function.*

*Proof.* Hölder inequality implies that for all $z, x \in \mathbb{R}^p$,

$$(z^\top x)^2 \leq \|x\|_\infty^2 \|z\|_1^2 \leq \epsilon^2 \|z\|_1^2 \quad .$$

Finally, we notice this upper bound is reached for $x^* = \pm \epsilon \mathsf{sgn}(z)$ as

$$(z^\top x^*)^2 = \epsilon^2 (z^\top \mathsf{sgn}(z))^2 = \epsilon^2 \|z\|_1^2 \quad .$$

$\square$

**Lemma 13.** *Let* $\mathbf{A}$ *be a symmetric matrix, we have that*

$$\sup_{\|\mathbf{w}\|_2 \le W, \|\mathbf{w}'\|_2 \le W} \mathbf{w}^\top \mathbf{A} \mathbf{w}' = \|\mathbf{A}\|_2 \quad .$$

*Proof.* Let $\mathbf{w}, \mathbf{w}'$ with $\ell_2-$norm smaller than $W$. By Cauchy-Schwarz's inequality we directly get that

$$\mathbf{w}^\top \mathbf{A} \mathbf{w}' \le |\langle \mathbf{w},\ \mathbf{A} \mathbf{w}' \rangle| \overset{\text{Cauchy-Schwarz}}{\le} \|\mathbf{A}\|_2 \|\mathbf{w}\|_2 \|\mathbf{w}'\|_2 \le W^2 \|\mathbf{A}\|_2 \quad . \tag{19}$$

We then perform eigendecomposition on $\mathbf{A}$:

$$\mathbf{w}^\top \mathbf{A} \mathbf{w}' = \mathbf{w}^\top \mathbf{U} \boldsymbol{\Sigma} \mathbf{U}^\top \mathbf{w}' = W^2 \mathbf{y}^\top \boldsymbol{\Sigma} \mathbf{y}' \quad ,$$

where $\boldsymbol{\Sigma}$ is a diagonal matrix containing eigenvalues $\lambda_i$'s of $\mathbf{A}$, $\mathbf{U}$ is an orthogonal matrix since $\mathbf{A}$ is symmetric, $\mathbf{y} := \frac{1}{W} \mathbf{U}^\top \mathbf{w}$ and $\mathbf{y}' := \frac{1}{W} \mathbf{U}^\top \mathbf{w}'$. In this orthogonal basis, let $i^*$ be the coordinate of the eigenvalue $\lambda_{i^*}$ with largest magnitude in absolute value. We denote by $(\mathbf{e}_i)_{i \in [d]}$ the canonical basis of $\mathbb{R}^d$. Let $\mathbf{y} = \mathbf{e}_{i^*}$ and $\mathbf{y}' = \mathsf{sgn}(\lambda_{i^*})\mathbf{y}$, we get

$$\mathbf{y}^\top \boldsymbol{\Sigma} \mathbf{y}' = |\lambda_{i^*}| = \sqrt{\max_{i \in [d]} \lambda_i(\mathbf{A})^2} = \sqrt{\lambda_{\max}(\mathbf{A}^2)} = \|\mathbf{A}\|_2 \quad .$$

Thus, the upper bound in (19) is attained by inverting the change of variable from $\mathbf{y}, \mathbf{y}'$ to $\mathbf{w}, \mathbf{w}'$.

$\square$

**Lemma 14.** *Let* $\mathbf{A} \in \mathbb{R}^{d \times d}$. *Then the following statements hold:*

$$\sup_{\|\mathbf{w}\|_p \le W, \|\mathbf{w}'\|_p \le W} \mathbf{w}^\top \mathbf{A} \mathbf{w}' \le \sup_{\|\mathbf{w}\|_2 \le W, \|\mathbf{w}'\|_2 \le W} \mathbf{w}^\top \mathbf{A} \mathbf{w}' \cdot \begin{cases} 1, & \text{if } 1 \le p \le 2 \\ d^{1-2/p}, & \text{else if } p > 2 \end{cases}, \tag{20}$$

$$\sup_{\|\mathbf{w}\|_p \le W, \|\mathbf{w}'\|_p \le W} \mathbf{w}^\top \mathbf{A} \mathbf{w}' \ge \sup_{\|\mathbf{w}\|_2 \le W, \|\mathbf{w}'\|_2 \le W} \mathbf{w}^\top \mathbf{A} \mathbf{w}' \cdot \begin{cases} d^{1-2/p}, & \text{if } 1 \le p \le 2 \\ 1, & \text{else if } p > 2 \end{cases}. \tag{21}$$

*Proof.* We begin with proving the first inequality. If $1 \le p \le 2$, we know that:

$$\mathcal{B}_p(W) \subseteq \mathcal{B}_2(W).$$

Hence $\sup_{\|\mathbf{w}\|_p \le W, \|\mathbf{w}'\|_p \le W} \mathbf{w}^\top \mathbf{A} \mathbf{w}' \le \sup_{\|\mathbf{w}\|_2 \le W, \|\mathbf{w}'\|_2 \le W} \mathbf{w}^\top \mathbf{A} \mathbf{w}'$.

If $p > 2$, since $\frac{1}{d^{1/2-1/p}} \|\mathbf{w}\|_2 \le \|\mathbf{w}\|_p$, we know that $\|\mathbf{w}\|_p \le W$ implies $\frac{1}{d^{1/2-1/p}} \|\mathbf{w}\|_2 \le W$. So we have:

$$\mathcal{B}_p(W) := \{\mathbf{w} : \|\mathbf{w}\|_p \le W\} \subseteq \{\mathbf{w} : \frac{1}{d^{1/2-1/p}} \|\mathbf{w}\|_2 \le W\} \subseteq \mathcal{B}_2(W d^{1/2-1/p}).$$

Hence:

$$\sup_{\|\mathbf{w}\|_p \le W, \|\mathbf{w}'\|_p \le W} \mathbf{w}^\top \mathbf{A} \mathbf{w}' \le \sup_{\|\mathbf{w}\|_2 \le W d^{1/2-1/p}, \|\mathbf{w}'\|_2 \le W d^{1/2-1/p}} \mathbf{w}^\top \mathbf{A} \mathbf{w}'$$

$$\le \sup_{\|\mathbf{w}\|_2 \le W, \|\mathbf{w}'\|_2 \le W} \mathbf{w}^\top \mathbf{A} \mathbf{w}' \cdot d^{1-2/p} \quad .$$

Now we switch to prove the second inequality. If $1 \le p \le 2$, then $\|\mathbf{w}\|_2 \ge \frac{1}{d^{1/p-1/2}} \|\mathbf{w}\|_p$, so we know

$$\mathcal{B}_2(W) := \{\mathbf{w} : \|\mathbf{w}\|_2 \le W\} \subseteq \{\mathbf{w} : \frac{1}{d^{1/p-1/2}} \|\mathbf{w}\|_p \le W\} = \mathcal{B}_p(d^{1/p-1/2}W).$$

Hence:

$$\sup_{\|\mathbf{w}\|_2 \le W, \|\mathbf{w}'\|_2 \le W} \mathbf{w}^\top \mathbf{A} \mathbf{w}' \le \sup_{\|\mathbf{w}\|_p \le d^{1/p-1/2}W, \|\mathbf{w}'\|_p \le d^{1/p-1/2}W} \mathbf{w}^\top \mathbf{A} \mathbf{w}'$$

$$\le d^{2/p-1} \sup_{\|\mathbf{w}\|_p \le W, \|\mathbf{w}'\|_p \le W} \mathbf{w}^\top \mathbf{A} \mathbf{w}' \quad .$$

If $p > 2$, then we have

$$\mathcal{B}_2(W) \subseteq \mathcal{B}_p(W),$$

so we can conclude the relation:

$$\sup_{\|\mathbf{w}\|_2 \leq W, \|\mathbf{w}'\|_2 \leq W} \mathbf{w}^\top \mathbf{A} \mathbf{w}' \leq \sup_{\|\mathbf{w}\|_p \leq W, \|\mathbf{w}'\|_p \leq W} \mathbf{w}^\top \mathbf{A} \mathbf{w}'.$$

$\square$

**Lemma 15** (Partition)**.** *Let us define* $\mathcal{A} := \{-1, +1\}^N$*. Then, there must be an equal partition of* $\mathcal{A} = \mathcal{A}^+ + \mathcal{A}^-$*, such that* $\mathcal{A}^-$ *is obtained by multiplying* $-1$ *on each vector in* $\mathcal{A}^+$*. That is,* $|\mathcal{A}^+| = |\mathcal{A}^-|$ *and* $\mathcal{A}^- = \{-\mathbf{a}, \mathbf{a} \in \mathcal{A}^+\}$*.*

*Proof.* We prove by induction. When $N = 1$, we have $\mathcal{A}_1 = \{-1, 1\}$, and we can partition it as $\mathcal{A}^+ = \{1\}$, and $\mathcal{A}^- = \{-1\}$;

The we assume the hypothesis holds for $N = k$, that is, $\mathcal{A}_k := \{-1, +1\}^k$ can be partition as $\mathcal{A}_k = \mathcal{A}_k^+ + \mathcal{A}_k^-$ such that $|\mathcal{A}_k^+| = |\mathcal{A}_k^-|$ and $\mathcal{A}_k^- = \{-\mathbf{a}, \mathbf{a} \in \mathcal{A}_k^+\}$. Now, for $N = k + 1$, we append all vectors $\mathbf{a} \in \mathcal{A}_k^+$ by 1, and put $[\mathbf{a}^\top, 1]$ into $\mathcal{A}_{k+1}^+$ and append all vectors $\mathbf{a} \in \mathcal{A}_k^-$ by $-1$, and put $[\mathbf{a}^\top, -1]$ into $\mathcal{A}_{k+1}^-$. It can be verify that, $|\mathcal{A}_{k+1}^+| = |\mathcal{A}_{k+1}^-|$ and $\mathcal{A}_{k+1}^- = \{-\mathbf{a}, \mathbf{a} \in \mathcal{A}_{k+1}^+\}$.

$\square$

### D.3 QUADRATIC OBJECTIVE SUBJECT TO INFINITE NORM CONSTRAINT

**Lemma 16.** *Let* $\mathbf{w} \in \mathbb{R}^p$*,* $a \in \mathbb{R}$ *and* $\epsilon \geq 0$*. Let us consider the problem*

$$\boldsymbol{\delta}^* = \underset{\boldsymbol{\delta} \in \mathbb{R}^p : \|\boldsymbol{\delta}\|_\infty \leq \epsilon}{\arg\max} (\mathbf{w}^\top \boldsymbol{\delta} + a)^2 .$$

*The solution is given by*

$$\boldsymbol{\delta}^* = \epsilon \, \mathsf{sgn}(a) \mathsf{sgn}(\mathbf{w}) \in \mathbb{R}^p , \tag{22}$$

*where we overload the notation* $\mathsf{sgn}$ *denotes in the mean time a single element and a coordinate wise sign operator, i.e.,* $\mathsf{sgn}(a) \in \mathbb{R}$ *but* $\mathsf{sgn}(\mathbf{w}) \in \mathbb{R}^p$*. Moreover, the maximum reached is*

$$(\mathbf{w}^T \boldsymbol{\delta}^* + a)^2 = (\epsilon \, \mathsf{sgn}(a) \mathbf{w}^T \mathsf{sgn}(\mathbf{w}) + a)^2 = (\epsilon \|\mathbf{w}\|_1 + |a|)^2 . \tag{23}$$

*Proof.* Let us give a first intuition and proof in dimension one and then extend this to larger dimensions.

- Case $p = 1$ ($\mathbf{w}$ becomes $w$). In this setting the problem intuition is clear: one should select $\boldsymbol{\delta}$, with maximal amplitude, that makes $\boldsymbol{\delta} w$ having the same sign as $a$. If $a$ and $w$ have the same sign, then $\boldsymbol{\delta} = \epsilon$. Else, $\boldsymbol{\delta} = -\epsilon$.

- Case $p \in \mathbb{N}^*$. For all $\mathbf{w}, \delta \in \mathbb{R}^p$, Hölder inequality gives that

$$|\mathbf{w}^\top \delta| \leq \|\delta\|_\infty \|\mathbf{w}\|_1 .$$

Let $\boldsymbol{\delta} \in \mathbb{R}^p$ such that $\|\boldsymbol{\delta}\|_\infty \leq \epsilon$. Then, this implies that on the feasible set

$$|\mathbf{w}^\top \delta| \leq \epsilon \|\mathbf{w}\|_1 . \tag{24}$$

Thus,

$$\begin{aligned}
(\mathbf{w}^T \boldsymbol{\delta} + a)^2 &= (\mathbf{w}^T \boldsymbol{\delta})^2 + 2a\mathbf{w}^T \boldsymbol{\delta} + a^2 \\
&\overset{(24)}{\leq} \epsilon^2 \|\mathbf{w}\|_1^2 + 2a\mathbf{w}^T \boldsymbol{\delta} + a^2 \\
&\leq \epsilon^2 \|\mathbf{w}\|_1^2 + 2|a||\mathbf{w}^T \boldsymbol{\delta}| + a^2 \\
&\overset{(24)}{\leq} \epsilon^2 \|\mathbf{w}\|_1^2 + 2|a|\epsilon \|\mathbf{w}\|_1 + a^2 \\
&= (\epsilon \|\mathbf{w}\|_1 + |a|)^2 .
\end{aligned}$$

Finally, one can check that upper bound of the objective is attained for $\boldsymbol{\delta}^*$ given in (22).

$\square$

**Lemma 17.** *Let* $\mathbf{w} \in \mathbb{R}^p$, $a \in \mathbb{R}$ *and* $\epsilon \geq 0$. *Let us consider the problem*

$$\boldsymbol{\delta}^* = \underset{\boldsymbol{\delta} \in \mathbb{R}^p : \|\boldsymbol{\delta}\|_\infty \leq \epsilon}{\arg\min} (\mathbf{w}^T \boldsymbol{\delta} + a)^2 \ . \tag{25}$$

*Let* $I := \{i \in [p] : \mathbf{w}_i \neq 0\}$.

- *If* $\epsilon \|\mathbf{w}\|_1 \geq |a|$, *then a solution is given by*

$$\begin{cases} \boldsymbol{\delta}_i^* = -\frac{a}{\|\mathbf{w}\|_1} \frac{\mathbf{w}_i}{|\mathbf{w}_i|} & \forall i \in I \\ \boldsymbol{\delta}_i^* = 0 & \forall i \in [p] \backslash I \end{cases} \ .$$

- *Else* $\epsilon \|\mathbf{w}\|_1 < |a|$, *and a the solution is given by*

$$\begin{cases} \boldsymbol{\delta}_i^* = -\epsilon \frac{a}{|a|} \frac{\mathbf{w}_i}{|\mathbf{w}_i|} & \forall i \in I \\ \boldsymbol{\delta}_i^* = 0 & \forall i \in [p] \backslash I \end{cases} \ .$$

*This solution can be condensed in the following formulation:*

$$\begin{cases} \boldsymbol{\delta}_i^* = -a \frac{w_i}{|w_i|} \min \left\{ \frac{1}{\|\mathbf{w}\|_1}, \frac{\epsilon}{|a|} \right\} & \forall i \in I \\ \boldsymbol{\delta}_i^* = 0 & \forall i \in [p] \backslash I \end{cases} \ . \tag{26}$$

*The minimal value is given by:*

$$\underset{\boldsymbol{\delta} \in \mathbb{R}^p : \|\boldsymbol{\delta}\|_\infty \leq \epsilon}{\min} (\mathbf{w}^T \boldsymbol{\delta} + a)^2 = a^2 \left( 1 - \min \left\{ 1, \frac{\epsilon \|\mathbf{w}\|_1}{|a|} \right\} \right)^2 \ .$$

**Remark 2.** *Note that in general there are an infinite number of solutions to* (25) *as in* (26) *one can choose arbitrarily the value of* $\boldsymbol{\delta}_i^*$ *for all* $i \in [p] \backslash I$ *(as soon as it is kept smaller than* $\epsilon$ *in absolute value).*

*Proof.* Let $[p] := \{1, \ldots, p\}$. Let us try to build a solution which drives the dot product $\mathbf{w}^\top \boldsymbol{\delta}$ towards $-a$. Let $I := \{i \in [p] : \mathbf{w}_i \neq 0\}$.

**Case 1:** If $\epsilon \|\mathbf{w}\|_1 \geq |a|$. Let us $\boldsymbol{\delta}^* \in \mathbb{R}^p$ such that

$$\begin{cases} \boldsymbol{\delta}_i^* = -\frac{a}{\|\mathbf{w}\|_1} \frac{\mathbf{w}_i}{|\mathbf{w}_i|} & \forall i \in I \\ \boldsymbol{\delta}_i^* = 0 & \forall i \in [p] \backslash I \end{cases} \ .$$

This vector is in the feasible set as $\|\boldsymbol{\delta}^*\|_\infty = \max_{i \in I} \frac{|a|}{\|\mathbf{w}\|_1} \frac{|\mathbf{w}_i|}{|\mathbf{w}_i|} = \frac{|a|}{\|\mathbf{w}\|_1} \leq \epsilon$, as assumed. Then,

$$\mathbf{w}^T \boldsymbol{\delta}^* + a = -\frac{a}{\|\mathbf{w}\|_1} \sum_{i \in I} \frac{\mathbf{w}_i^2}{|\mathbf{w}_i|} + a = 0 \ .$$

This means that if the entries of vector $\mathbf{w}$ are large enough (in absolute value), we can build a feasible vector $\boldsymbol{\delta}^*$ such that the objective in (25) is zero.

**Case 2:** If $\epsilon \|\mathbf{w}\|_1 < |a|$. Let us $\boldsymbol{\delta}^* \in \mathbb{R}^p$ such that

$$\begin{cases} \boldsymbol{\delta}_i^* = -\epsilon \frac{a}{|a|} \frac{\mathbf{w}_i}{|\mathbf{w}_i|} & \forall i \in I \\ \boldsymbol{\delta}_i^* = 0 & \forall i \in [p] \backslash I \end{cases} \ .$$

This vector is in the feasible set as $\|\boldsymbol{\delta}^*\|_\infty = \max_{i \in I} \epsilon \frac{|a|}{|a|} \frac{|\mathbf{w}_i|}{|\mathbf{w}_i|} = \epsilon$, as assumed. Then,

$$\mathbf{w}^T \boldsymbol{\delta}^* + a = -\epsilon \frac{a}{|a|} \sum_{i \in I} \frac{\mathbf{w}_i^2}{|\mathbf{w}_i|} + a = a \underbrace{\left( 1 - \epsilon \frac{\|\mathbf{w}\|_1}{|a|} \right)}_{\in [0,1]} \ .$$

This means that if the entries of vector $\mathbf{w}$ are too small (in absolute value), we can only build a feasible vector $\boldsymbol{\delta}^*$ such that $\mathbf{w}^\top \boldsymbol{\delta}^*$ close too $-a$. And the corresponding objective in (25) becomes $a^2 \left(1 - \epsilon \frac{\|\mathbf{w}\|_1}{|a|}\right)^2$.

Finally, one just can show with Hölder inequality that $(\mathbf{w}^T \boldsymbol{\delta} + a)^2 \geq a^2 \left(1 - \epsilon \frac{\|\mathbf{w}\|_1}{|a|}\right)^2$ for all $\boldsymbol{\delta}$ in the feasible set, which concludes the proof. If $a \geq 0$, the computation follows easily, else we can just replace $\boldsymbol{\delta} \leftarrow -\boldsymbol{\delta}$ to get back to the former case.

$\square$

## E  PROOF OF GENERALIZATION LEMMA ( LEMMA 2)

In this section we provide the proof of Lemma 2. First let us introduce the following helper lemma.

**Lemma 18.** *Assume $\hat{\mathcal{S}}$ and $\hat{\mathcal{T}}$ are the sets of data drawn from $\mathcal{S}$ and $\mathcal{T}$, with size $n_{\mathcal{S}}$ and $n_{\mathcal{T}}$ respectively, and the value of $\tilde{\ell}(\cdot)$ is bounded by $M$. Then we have:*

$$|disc^{adv}_{\mathcal{H}\Delta\mathcal{H}}(\mathcal{S}, \mathcal{T}) - disc^{adv}_{\mathcal{H}\Delta\mathcal{H}}(\hat{\mathcal{S}}, \hat{\mathcal{T}})| \le \hat{\mathfrak{R}}_{\mathcal{S}}(\tilde{\ell} \circ \mathcal{H}\Delta\mathcal{H}) + \hat{\mathfrak{R}}_{\mathcal{T}}(\tilde{\ell} \circ \mathcal{H}\Delta\mathcal{H})$$
$$+ \left( 3M\sqrt{\frac{\log(2/c)}{n_{\mathcal{S}}}} + 3M\sqrt{\frac{\log(2/c)}{n_{\mathcal{T}}}} \right) \ .$$

*Proof.* Since absolute value satisfies triangle inequality, we have:

$$disc^{adv}_{\mathcal{H}\Delta\mathcal{H}}(\mathcal{S}, \mathcal{T}) \overset{(4)}{\le} disc^{adv}_{\mathcal{H}\Delta\mathcal{H}}(\mathcal{S}, \hat{\mathcal{S}}) + disc^{adv}_{\mathcal{H}\Delta\mathcal{H}}(\mathcal{T}, \hat{\mathcal{T}}) + disc^{adv}_{\mathcal{H}\Delta\mathcal{H}}(\hat{\mathcal{S}}, \hat{\mathcal{T}}) \ .$$

According to Rademacher-based generalization bound of Mohri et al. (2018), we know that

$$disc^{adv}_{\mathcal{H}\Delta\mathcal{H}}(\mathcal{S}, \hat{\mathcal{S}}) = \max_{h,h'\in\mathcal{H}} |\mathcal{R}^{adv}_{\mathcal{S}}(h, h') - \mathcal{R}^{adv}_{\hat{\mathcal{S}}}(h, h')| \le \mathfrak{R}_{\hat{\mathcal{S}}}(\tilde{\ell} \circ \mathcal{H}\Delta\mathcal{H}) + 3M\sqrt{\frac{\log(2/c)}{n_{\mathcal{S}}}} \ ,$$

and so is for $disc^{adv}_{\mathcal{H}\Delta\mathcal{H}}(\mathcal{T}, \hat{\mathcal{T}})$. $\qquad\square$

**Proof of Lemma 2.**

*Proof.* Since the loss function $\tilde{\ell}$ satisfies triangle inequality, we can split $\mathcal{R}^{adv}_{\mathcal{T}}(\mathbf{w}, \mathbf{v}^*_{\mathcal{T}})$ into the following terms:

$$\mathcal{R}^{adv-label}_{\mathcal{T}}(h_{\mathbf{w}}, y_{\mathcal{T}}) \le \mathcal{R}^{adv}_{\mathcal{T}}(h_{\mathbf{w}}, h_{\mathbf{w}^*_{\mathcal{S}}}) + \mathcal{R}^{adv}_{\mathcal{T}}(h_{\mathbf{w}^*_{\mathcal{S}}}, h_{\mathbf{w}^*_{\mathcal{T}}}) + \mathcal{R}^{adv}_{\mathcal{T}}(h_{\mathbf{w}^*_{\mathcal{T}}}, y_{\mathcal{T}})$$
$$\le \mathcal{R}^{adv}_{\mathcal{S}}(h_{\mathbf{w}}, h_{\mathbf{w}^*_{\mathcal{S}}}) + disc^{adv}_{\mathcal{H}\Delta\mathcal{H}}(\mathcal{T}, \mathcal{S}) + \mathcal{R}^{adv}_{\mathcal{T}}(h_{\mathbf{w}^*_{\mathcal{S}}}, h_{\mathbf{w}^*_{\mathcal{T}}}) + \mathcal{R}^{adv}_{\mathcal{T}}(h_{\mathbf{w}^*_{\mathcal{T}}}, y_{\mathcal{T}})$$
$$\le \mathcal{R}^{adv}_{\mathcal{S}}(h_{\mathbf{w}}, h_{\mathbf{w}^*_{\mathcal{S}}}) + disc^{adv}_{\mathcal{H}\Delta\mathcal{H}}(\hat{\mathcal{T}}, \hat{\mathcal{S}}) + \mathcal{R}^{adv}_{\mathcal{T}}(h_{\mathbf{w}^*_{\mathcal{S}}}, h_{\mathbf{w}^*_{\mathcal{T}}}) + \mathcal{R}^{adv}_{\mathcal{T}}(h_{\mathbf{w}^*_{\mathcal{T}}}, y_{\mathcal{T}})$$
$$+ \hat{\mathfrak{R}}_{\mathcal{S}}(\tilde{\ell} \circ \mathcal{H}\Delta\mathcal{H}) + \hat{\mathfrak{R}}_{\mathcal{T}}(\tilde{\ell} \circ \mathcal{H}\Delta\mathcal{H}) + \left( 3M\sqrt{\frac{\log(2/c)}{n_{\mathcal{S}}}} + 3M\sqrt{\frac{\log(2/c)}{n_{\mathcal{T}}}} \right)$$
$$\le \mathcal{R}^{adv-label}_{\mathcal{S}}(h_{\mathbf{w}}, y_{\mathcal{S}}) + \mathcal{R}^{adv-label}_{\mathcal{S}}(h_{\mathbf{w}^*_{\mathcal{S}}}, y_{\mathcal{S}})$$
$$+ disc^{adv}_{\mathcal{H}\Delta\mathcal{H}}(\hat{\mathcal{T}}, \hat{\mathcal{S}}) + \mathcal{R}^{adv}_{\mathcal{T}}(h_{\mathbf{w}^*_{\mathcal{S}}}, h_{\mathbf{w}^*_{\mathcal{T}}}) + \mathcal{R}^{adv}_{\mathcal{T}}(h_{\mathbf{w}^*_{\mathcal{T}}}, y_{\mathcal{T}})$$
$$+ \hat{\mathfrak{R}}_{\mathcal{S}}(\tilde{\ell} \circ \mathcal{H}\Delta\mathcal{H}) + \hat{\mathfrak{R}}_{\mathcal{T}}(\tilde{\ell} \circ \mathcal{H}\Delta\mathcal{H}) + \left( 3M\sqrt{\frac{\log(2/c)}{n_{\mathcal{S}}}} + 3M\sqrt{\frac{\log(2/c)}{n_{\mathcal{T}}}} \right),$$

where we plug in Lemma 18 at last step. $\qquad\square$

## F  EXTENSIONS

### F.1  ESTIMATION OF ADVERSARIAL DISCREPANCY FROM STANDARD DISCREPANCY

The following Lemma gives the bound if we estimate adversarial $\mathcal{H}\Delta\mathcal{H}$ discrepancy from standard $\mathcal{H}\Delta\mathcal{H}$ discrepancy.

**Lemma 19.** *The following relations between adversarial discrepancy from standard discrepancy holds for linear model class with bounded norm: $\mathcal{H} = \{h_{\mathbf{w}} : \mathbf{x} \mapsto \langle \mathbf{w}, \mathbf{x} \rangle, \|\mathbf{w}\|_p \le W\}$. For $L_\phi$-Lipschitz binary classification loss, we have:*

$$disc^{adv}_{\mathcal{H}\Delta\mathcal{H}}(\hat{\mathcal{S}}, \hat{\mathcal{T}}) \le disc_{\mathcal{H}\Delta\mathcal{H}}(\hat{\mathcal{S}}, \hat{\mathcal{T}})$$
$$+ 2W^2 L_\phi \sqrt{d}\epsilon \left( \frac{1}{n_{\mathcal{T}}} \sum_{\mathbf{x}_i \in \hat{\mathcal{T}}} \|\mathbf{x}_i\|_2 + \frac{1}{n_{\mathcal{S}}} \sum_{\mathbf{x}_i \in \hat{\mathcal{S}}} \|\mathbf{x}_i\|_2 \right) \cdot \begin{cases} 1 & 1 \le p \le 2 \\ d^{1-2/p} & p > 2 \end{cases} \ .$$

*For $\ell_2$ regression loss, we have:*

$$disc_{\mathcal{H}\Delta\mathcal{H}}^{adv}(\hat{\mathcal{S}}, \hat{\mathcal{T}}) \leq disc_{\mathcal{H}\Delta\mathcal{H}}(\hat{\mathcal{S}}, \hat{\mathcal{T}})$$

$$+ 8\sqrt{d}\epsilon W^2 \left(\frac{1}{n_{\mathcal{T}}} \sum_{\mathbf{x}_i \in \hat{\mathcal{T}}} \|\mathbf{x}_i\|_2 + \frac{1}{n_{\mathcal{T}}} \sum_{\mathbf{x}_i \in \hat{\mathcal{S}}} \|\mathbf{x}_i\|_2\right) \cdot \begin{cases} 1 & 1 \leq p \leq 2 \\ d^{1-2/p} & p > 2 \end{cases} .$$

*Proof.* Let $\mathcal{B}_p(W) := \{\mathbf{w} \in \mathbb{R}^d : \|\mathbf{w}\|_p \leq W\}$ be the $\ell_p$-norm centered ball with radius $W$. By the definition of $disc_{\mathcal{H}\Delta\mathcal{H}}^{adv}(\hat{\mathcal{S}}, \hat{\mathcal{T}})$, we have:

$$disc_{\mathcal{H}\Delta\mathcal{H}}^{adv}(\hat{\mathcal{S}}, \hat{\mathcal{T}}) \overset{(4)}{=} \max_{\mathbf{w},\mathbf{w}' \in \{\mathbf{w}:\|\mathbf{w}\|_p \leq W\}} |\mathcal{R}_{\hat{\mathcal{T}}}^{adv}(\mathbf{w}, \mathbf{w}') - \mathcal{R}_{\hat{\mathcal{S}}}^{adv}(\mathbf{w}, \mathbf{w}')|$$

$$\leq \max_{\mathbf{w},\mathbf{w}' \in \mathcal{B}_p(W)^2} |\mathcal{R}_{\hat{\mathcal{T}}}(\mathbf{w}, \mathbf{w}') - \mathcal{R}_{\hat{\mathcal{S}}}(\mathbf{w}, \mathbf{w}') + \mathcal{R}_{\hat{\mathcal{T}}}^{adv}(\mathbf{w}, \mathbf{w}') - \mathcal{R}_{\hat{\mathcal{T}}}(\mathbf{w}, \mathbf{w}')$$

$$- (\mathcal{R}_{\hat{\mathcal{S}}}^{adv}(\mathbf{w}, \mathbf{w}') - \mathcal{R}_{\hat{\mathcal{S}}}(\mathbf{w}, \mathbf{w}'))|$$

$$\overset{(2)}{\leq} disc_{\mathcal{H}\Delta\mathcal{H}}(\hat{\mathcal{S}}, \hat{\mathcal{T}}) + \max_{\mathbf{w},\mathbf{w}' \in \mathcal{B}_p(W)^2} |\mathcal{R}_{\hat{\mathcal{T}}}^{adv}(\mathbf{w}, \mathbf{w}') - \mathcal{R}_{\hat{\mathcal{T}}}(\mathbf{w}, \mathbf{w}')|$$

$$+ \max_{\mathbf{w},\mathbf{w}' \in \mathcal{B}_p(W)^2} |\mathcal{R}_{\hat{\mathcal{S}}}^{adv}(\mathbf{w}, \mathbf{w}') - \mathcal{R}_{\hat{\mathcal{S}}}(\mathbf{w}, \mathbf{w}')| .$$

Now we study the gap $\max_{\mathbf{w},\mathbf{w}' \in \mathcal{B}_p(W)^2} |\mathcal{R}_{\hat{\mathcal{T}}}^{adv}(\mathbf{w}, \mathbf{w}') - \mathcal{R}_{\hat{\mathcal{T}}}(\mathbf{w}, \mathbf{w}')|$ and $\max_{\mathbf{w},\mathbf{w}' \in \mathcal{B}_p(W)^2} |\mathcal{R}_{\hat{\mathcal{S}}}^{adv}(\mathbf{w}, \mathbf{w}') - \mathcal{R}_{\hat{\mathcal{S}}}(\mathbf{w}, \mathbf{w}')|$. For linear classification, we have:

$$\max_{\mathbf{w},\mathbf{w}' \in \mathcal{B}_p(W)^2} |\mathcal{R}_{\hat{\mathcal{T}}}^{adv}(\mathbf{w}, \mathbf{w}') - \mathcal{R}_{\hat{\mathcal{T}}}(\mathbf{w}, \mathbf{w}')|$$

$$= \max_{\mathbf{w},\mathbf{w}' \in \mathcal{B}_p(W)^2} \left| \frac{1}{n_{\mathcal{T}}} \sum_{\mathbf{x}_i \in \hat{\mathcal{T}}} \max_{\boldsymbol{\delta}:\|\boldsymbol{\delta}\|_\infty \leq \epsilon} (\phi(\langle\mathbf{w}, \mathbf{x}_i + \boldsymbol{\delta}\rangle \cdot \langle\mathbf{w}', \mathbf{x}_i + \boldsymbol{\delta}\rangle) - \phi(\langle\mathbf{w}, \mathbf{x}_i\rangle \cdot \langle\mathbf{w}', \mathbf{x}_i\rangle)) \right|$$

$$\leq \max_{\mathbf{w},\mathbf{w}' \in \mathcal{B}_p(W)^2} \left| \frac{1}{n_{\mathcal{T}}} \sum_{\mathbf{x}_i \in \hat{\mathcal{T}}} \max_{\boldsymbol{\delta}:\|\boldsymbol{\delta}\|_\infty \leq \epsilon} L_\phi |(\langle\mathbf{w}, \mathbf{x}_i + \boldsymbol{\delta}\rangle \cdot \langle\mathbf{w}', \mathbf{x}_i + \boldsymbol{\delta}\rangle) - (\langle\mathbf{w}, \mathbf{x}_i\rangle \cdot \langle\mathbf{w}', \mathbf{x}_i\rangle)| \right|$$

$$\leq \max_{\mathbf{w},\mathbf{w}' \in \mathcal{B}_p(W)^2} \left| \frac{1}{n_{\mathcal{T}}} \sum_{\mathbf{x}_i \in \hat{\mathcal{T}}} \max_{\boldsymbol{\delta}:\|\boldsymbol{\delta}\|_\infty \leq \epsilon} L_\phi |\mathbf{w}^\top \left((\mathbf{x}_i + \boldsymbol{\delta})(\mathbf{x}_i + \boldsymbol{\delta})^\top - \mathbf{x}_i\mathbf{x}_i^\top\right) \mathbf{w}'| \right|$$

$$\leq L_\phi \frac{1}{n_{\mathcal{T}}} \sum_{\mathbf{x}_i \in \hat{\mathcal{T}}} \max_{\mathbf{w},\mathbf{w}' \in \mathcal{B}_p(W)^2} |\mathbf{w}^\top \left((\mathbf{x}_i + \boldsymbol{\delta}_i^*)(\mathbf{x}_i + \boldsymbol{\delta}_i^*)^\top - \mathbf{x}_i\mathbf{x}_i^\top\right) \mathbf{w}'|$$

$$\overset{\text{Cauchy-Schwarz}}{\leq} L_\phi \frac{1}{n_{\mathcal{T}}} \sum_{\mathbf{x}_i \in \hat{\mathcal{T}}} \max_{\mathbf{w},\mathbf{w}' \in \mathcal{B}_p(W)^2} \|\mathbf{w}\|_2 \|(\mathbf{x}_i + \boldsymbol{\delta}_i^*)(\mathbf{x}_i + \boldsymbol{\delta}_i^*)^\top - \mathbf{x}_i\mathbf{x}_i^\top\|_2 \|\mathbf{w}'\|_2$$

$$\leq L_\phi \frac{1}{n_{\mathcal{T}}} \sum_{\mathbf{x}_i \in \hat{\mathcal{T}}} \max_{\mathbf{w},\mathbf{w}' \in \mathcal{B}_p(W)^2} \|\mathbf{w}\|_2 \|\boldsymbol{\delta}_i^* \mathbf{x}_i^\top + \mathbf{x}_i \boldsymbol{\delta}_i^{*\top}\|_2 \|\mathbf{w}'\|_2 ,$$

where $\boldsymbol{\delta}_i^*$ is a maximizer of the $i$-th optimization problem in $\boldsymbol{\delta}$ over the $\ell_\infty$-ball of radius $\epsilon$. We know that $\|\mathbf{w}\|_2 \leq W \cdot \begin{cases} 1 & 1 \leq p \leq 2 \\ d^{1/2-1/p} & p > 2 \end{cases}$, and $\|\boldsymbol{\delta}_i^* \mathbf{x}_i^\top + \mathbf{x}_i \boldsymbol{\delta}_i^{*\top}\|_2 \leq 2\sqrt{d}\epsilon\|\mathbf{x}_i\|_2$, so we have:

$$\max_{\mathbf{w},\mathbf{w}' \in \mathcal{B}_p(W)^2} |\mathcal{R}_{\hat{\mathcal{T}}}^{adv}(\mathbf{w}, \mathbf{w}') - \mathcal{R}_{\hat{\mathcal{T}}}(\mathbf{w}, \mathbf{w}')| \leq W^2 L_\phi \frac{1}{n_{\mathcal{T}}} \sum_{\mathbf{x}_i \in \hat{\mathcal{T}}} 2\sqrt{d}\epsilon\|\mathbf{x}_i\|_2 \cdot \begin{cases} 1 & 1 \leq p \leq 2 \\ d^{1-2/p} & p > 2 \end{cases} .$$

which concludes the proof for linear classification setting. Now we switch to regression setting:

$$\max_{\mathbf{w},\mathbf{w}'\in\mathcal{B}_p(W)^2}|\mathcal{R}^{adv}_{\hat{\mathcal{T}}}(\mathbf{w},\mathbf{w}')-\mathcal{R}_{\hat{\mathcal{T}}}(\mathbf{w},\mathbf{w}')|$$

$$=\max_{\mathbf{w},\mathbf{w}'\in\mathcal{B}_p(W)^2}\left|\frac{1}{n_{\mathcal{T}}}\sum_{\mathbf{x}_i\in\hat{\mathcal{T}}}\max_{\boldsymbol{\delta}:\|\boldsymbol{\delta}\|_\infty\leq\epsilon}\|\langle\mathbf{w},\mathbf{x}_i+\boldsymbol{\delta}\rangle-\langle\mathbf{w}',\mathbf{x}_i+\boldsymbol{\delta}\rangle\|_2^2-\|\langle\mathbf{w},\mathbf{x}_i\rangle-\langle\mathbf{w}',\mathbf{x}_i\rangle\|_2^2\right|$$

$$=\max_{\mathbf{w},\mathbf{w}'\in\mathcal{B}_p(W)^2}\left|\frac{1}{n_{\mathcal{T}}}\sum_{\mathbf{x}_i\in\hat{\mathcal{T}}}\max_{\boldsymbol{\delta}:\|\boldsymbol{\delta}\|_\infty\leq\epsilon}\|\langle\mathbf{w}-\mathbf{w}',\mathbf{x}_i+\boldsymbol{\delta}\rangle\|_2^2-\|\langle\mathbf{w}-\mathbf{w}',\mathbf{x}_i\rangle\|_2^2\right|$$

$$=\max_{\mathbf{w},\mathbf{w}'\in\mathcal{B}_p(W)^2}\left|\frac{1}{n_{\mathcal{T}}}\sum_{\mathbf{x}_i\in\hat{\mathcal{T}}}(\mathbf{w}-\mathbf{w}')^\top\left[(\mathbf{x}_i+\boldsymbol{\delta}_i^*)(\mathbf{x}_i+\boldsymbol{\delta}_i^*)^\top-\mathbf{x}_i\mathbf{x}_i^\top\right](\mathbf{w}-\mathbf{w}')\right|$$

$$=\max_{\mathbf{w},\mathbf{w}'\in\mathcal{B}_p(W)^2}\left|(\mathbf{w}-\mathbf{w}')^\top\frac{1}{n_{\mathcal{T}}}\sum_{\mathbf{x}_i\in\hat{\mathcal{T}}}\left[\boldsymbol{\delta}_i^*\mathbf{x}_i^\top+\mathbf{x}_i\boldsymbol{\delta}_i^{*\top}\right](\mathbf{w}-\mathbf{w}')\right|$$

Now, we let $\mathbf{v}:=\mathbf{w}-\mathbf{w}'$, and re-write the above inequality as:

$$\max_{\mathbf{w},\mathbf{w}'\in\mathcal{B}_p(W)^2}|\mathcal{R}^{adv}_{\hat{\mathcal{T}}}(\mathbf{w},\mathbf{w}')-\mathcal{R}_{\hat{\mathcal{T}}}(\mathbf{w},\mathbf{w}')|$$

$$\overset{\text{Cauchy-Schwarz}}{\leq}\max_{\mathbf{v}:\|\mathbf{v}\|_p\leq 2W}\left\|\mathbf{v}^\top\right\|_2\left\|\frac{1}{n_{\mathcal{T}}}\sum_{\mathbf{x}_i\in\hat{\mathcal{T}}}\left[\boldsymbol{\delta}_i^*\mathbf{x}_i^\top+\mathbf{x}_i\boldsymbol{\delta}_i^{*\top}\right]\right\|_2\|\mathbf{v}\|_2$$

$$\leq\left\|\frac{1}{n_{\mathcal{T}}}\sum_{\mathbf{x}_i\in\hat{\mathcal{T}}}\left[\boldsymbol{\delta}_i^*\mathbf{x}_i^\top+\mathbf{x}_i\boldsymbol{\delta}_i^{*\top}\right]\right\|_2 4W\cdot\begin{cases}1 & p\leq 2\\ d^{1-2/p} & p>2\end{cases}$$

$$\leq 8\sqrt{d}W\frac{1}{n_{\mathcal{T}}}\sum_{\mathbf{x}_i\in\hat{\mathcal{T}}}\|\mathbf{x}_i\|_2\cdot\begin{cases}1 & p\leq 2\\ d^{1-2/p} & p>2\end{cases}.$$

where we use norm equivalence (Lemma 8) to bound $\|\mathbf{v}\|_2$. $\qquad\square$

### F.2 ADVERSARIALLY ROBUST DOMAIN ADAPTATION GENERALIZATION BOUND

In this section, we will present the generalizatin bound of adversarially robust domain adaptation, using our upper bound for adversarial Rademacher complexity over $\mathcal{H}\Delta\mathcal{H}$ class. An immediate implication of Theorem 1 is the following bound:

**Corollary 1** (Adversarially Robust Domain Adapation Learning Bound, Linear Classification)**.** *Assume that the loss function $\tilde{\ell}$ is symmetric and obeys the triangle inequality. We further assume $\tilde{\ell}$ is bounded by $M$. Then, for any hypothesis $h_\mathbf{w}\in\mathcal{H}$, the following holds:*

$$\mathcal{R}^{adv-label}_{\mathcal{T}}(h_\mathbf{w},y_{\mathcal{T}})\leq\mathcal{R}^{adv-label}_{\mathcal{S}}(h_\mathbf{w},y_{\mathcal{S}})+\mathcal{R}^{adv-label}_{\mathcal{S}}(h_{\mathbf{w}_{\mathcal{S}}^*},y_{\mathcal{S}})$$

$$+disc^{adv}_{\mathcal{H}\Delta\mathcal{H}}(\hat{\mathcal{T}},\hat{\mathcal{S}})+\mathcal{R}^{adv}_{\mathcal{T}}(h_{\mathbf{w}_{\mathcal{T}}^*},h_{\mathbf{w}_{\mathcal{S}}^*})+\mathcal{R}^{adv}_{\mathcal{T}}(h_{\mathbf{w}_{\mathcal{T}}^*},y_{\mathcal{T}})$$

$$+\hat{\mathfrak{R}}_{\mathcal{S}}(\ell\circ\mathcal{H}\Delta\mathcal{H})+\hat{\mathfrak{R}}_{\mathcal{T}}(\ell\circ\mathcal{H}\Delta\mathcal{H})+3M\left(\sqrt{\frac{\log(2/c)}{n_{\mathcal{S}}}}+\sqrt{\frac{\log(2/c)}{n_{\mathcal{T}}}}\right)$$

$$+\tilde{\mathcal{O}}\left(L_\phi\epsilon d^{1/p^*}W^2\sqrt{d}\left(\frac{\epsilon d^{1/p^*}+2\|\mathbf{X}_{\mathcal{T}}\|_{p^*,\infty}}{\sqrt{n_{\mathcal{T}}}}+\frac{\epsilon d^{1/p^*}+2\|\mathbf{X}_{\mathcal{S}}\|_{p^*,\infty}}{\sqrt{n_{\mathcal{S}}}}\right)\right),$$

*where $p^*$ is such that $1/p+1/p^*=1$, and $\mathbf{X}_{\mathcal{S}}$ and $\mathbf{X}_{\mathcal{T}}$ are the data matrix concatenated by data points from $\hat{\mathcal{S}}$ and $\hat{\mathcal{T}}$, respectively.*

*Proof.* Combining Lemma 2 and Theorem 1 will conclude the proof. $\qquad\square$

An immediate implication of Theorem 2 is the following result.

**Corollary 2** (Adversarially Robust Domain Adapation Learning Bound, Linear Regression). *Assume that the loss function $\tilde{\ell}$ is symmetric and convex. Also let $\hat{S}$ and $\hat{T}$ have $n_S$ and $n_T$ data points, respectively. We further assume $\tilde{\ell}$ is bounded by $M$. Then, for any hypothesis $h_{\mathbf{w}} \in \mathcal{H}$, the following holds with probability at least $1 - c$:*

$$
\mathcal{R}_{\mathcal{T}}^{adv-label}(h_{\mathbf{w}}, y_{\mathcal{T}}) \leq 6\mathcal{R}_{\mathcal{S}}^{adv-label}(h_{\mathbf{w}}, y_{\mathcal{S}}) + 6\mathcal{R}_{\mathcal{S}}^{adv-label}(h_{\mathbf{w}_{\mathcal{S}}^*}, y_{\mathcal{S}})
$$

$$
+ 4disc_{\mathcal{H}\Delta\mathcal{H}}^{adv}(\hat{\mathcal{T}}, \hat{\mathcal{S}}) + 3\mathcal{R}_{\mathcal{T}}^{adv}(h_{\mathbf{w}_{\mathcal{T}}^*}, h_{\mathbf{w}_{\mathcal{S}}^*}) + 3\mathcal{R}_{\mathcal{T}}^{adv}(h_{\mathbf{w}_{\mathcal{S}}^*}, y_{\mathcal{T}})
$$

$$
+ M\mathcal{O}\left(\sqrt{\frac{\log(2/c)}{n_S}} + \sqrt{\frac{\log(2/c)}{n_T}}\right) + 3\hat{\mathfrak{R}}_{\mathcal{S}}(\ell \circ \mathcal{H}\Delta\mathcal{H}) + 3\hat{\mathfrak{R}}_{\mathcal{T}}(\ell \circ \mathcal{H}\Delta\mathcal{H})
$$

$$
+ \tilde{\mathcal{O}}\left(\frac{W^2}{\sqrt{n_S}}\left(d\epsilon \|\mathbf{X}_{\mathcal{S}}\|_{2,\infty} + d^{3/2}\epsilon^2\right) + \frac{W^2}{\sqrt{n_T}}\left(d\epsilon \|\mathbf{X}_{\mathcal{T}}\|_{2,\infty} + d^{3/2}\epsilon^2\right)\right) \times \begin{cases} 1, & 1 \leq p \leq 2 \\ d^{1-2/p}, & p > 2 \end{cases},
$$

*where $\mathbf{X}_{\mathcal{S}}$ and $\mathbf{X}_{\mathcal{T}}$ are the data matrix concatenated by data points from $\hat{\mathcal{S}}$ and $\hat{\mathcal{T}}$, respectively.*

*Proof.* The following proof is almost identical to that of Lemma 2, and the only change is that we apply Jensen's inequality instead of triangle inequality here:

$$
\mathcal{R}_{\mathcal{T}}^{adv}(h_{\mathbf{w}}, y_{\mathcal{T}}) \leq 3\mathcal{R}_{\mathcal{T}}^{adv}(h_{\mathbf{w}}, h_{\mathbf{w}_{\mathcal{S}}^*}) + 3\mathcal{R}_{\mathcal{T}}^{adv}(h_{\mathbf{w}_{\mathcal{S}}^*}, h_{\mathbf{w}_{\mathcal{T}}^*}) + 3\mathcal{R}_{\mathcal{T}}^{adv}(h_{\mathbf{w}_{\mathcal{T}}^*}, y_{\mathcal{T}})
$$

$$
\leq 3\mathcal{R}_{\mathcal{S}}^{adv}(h_{\mathbf{w}}, h_{\mathbf{w}_{\mathcal{S}}^*}) + 3disc_{\mathcal{H}\Delta\mathcal{H}}^{adv}(\mathcal{T}, \mathcal{S}) + 3\mathcal{R}_{\mathcal{T}}^{adv}(h_{\mathbf{w}_{\mathcal{S}}^*}, h_{\mathbf{w}_{\mathcal{T}}^*}) + 3\mathcal{R}_{\mathcal{T}}^{adv}(h_{\mathbf{w}_{\mathcal{T}}^*}, y_{\mathcal{T}})
$$

$$
\leq 3\mathcal{R}_{\mathcal{S}}^{adv}(h_{\mathbf{w}}, h_{\mathbf{w}_{\mathcal{S}}^*}) + 3disc_{\mathcal{H}\Delta\mathcal{H}}^{adv}(\hat{\mathcal{T}}, \hat{\mathcal{S}}) + 3\mathcal{R}_{\mathcal{T}}^{adv}(h_{\mathbf{w}_{\mathcal{S}}^*}, h_{\mathbf{w}_{\mathcal{T}}^*}) + 3\mathcal{R}_{\mathcal{T}}^{adv}(h_{\mathbf{w}_{\mathcal{T}}^*}, y_{\mathcal{T}})
$$

$$
+ 3\hat{\mathfrak{R}}_{\mathcal{S}}(\tilde{\ell} \circ \mathcal{H}\Delta\mathcal{H}) + 3\hat{\mathfrak{R}}_{\mathcal{T}}(\tilde{\ell} \circ \mathcal{H}\Delta\mathcal{H}) + 3\left(3M\sqrt{\frac{\log(2/c)}{n_S}} + 3M\sqrt{\frac{\log(2/c)}{n_T}}\right)
$$

$$
\leq 6\mathcal{R}_{\mathcal{S}}^{adv-label}(h_{\mathbf{w}}, y_{\mathcal{S}}) + 6\mathcal{R}_{\mathcal{S}}^{adv-label}(h_{\mathbf{w}_{\mathcal{S}}^*}, y_{\mathcal{S}})
$$

$$
+ 3disc_{\mathcal{H}\Delta\mathcal{H}}^{adv}(\hat{\mathcal{T}}, \hat{\mathcal{S}}) + 3\mathcal{R}_{\mathcal{T}}^{adv}(h_{\mathbf{w}_{\mathcal{S}}^*}, h_{\mathbf{w}_{\mathcal{T}}^*}) + 3\mathcal{R}_{\mathcal{T}}^{adv}(h_{\mathbf{w}_{\mathcal{T}}^*}, y_{\mathcal{T}})
$$

$$
+ 3\hat{\mathfrak{R}}_{\mathcal{S}}(\tilde{\ell} \circ \mathcal{H}\Delta\mathcal{H}) + 3\hat{\mathfrak{R}}_{\mathcal{T}}(\tilde{\ell} \circ \mathcal{H}\Delta\mathcal{H}) + 3\left(3M\sqrt{\frac{\log(2/c)}{n_S}} + 3M\sqrt{\frac{\log(2/c)}{n_T}}\right)
$$

where we plug in Lemma 18 at last step. Finally plugging in Theorem 2 will conclude the proof. $\square$

## G   PROOFS FOR BINARY CLASSIFICATION

### G.1   PROOF OF LEMMA 3

*Proof.* To simplify notations, we omit to specify the fact that the model parameters $\mathbf{w}$ and $\mathbf{w}'$ belong to $\mathbb{R}^d$. We first prove the upper bound results. By definition, we have

$$
\hat{\mathfrak{R}}_{\mathcal{D}}(f \circ \mathcal{H}\Delta\mathcal{H}) = \mathbb{E}_{\sigma}\left[\sup_{\|\mathbf{w}\|_p \leq W, \|\mathbf{w}'\|_p \leq W} \frac{1}{n}\sum_{i=1}^n \sigma_i \mathbf{w}^T \mathbf{x}_i \mathbf{w}'^T \mathbf{x}_i\right]
$$

$$
\overset{(20)}{\leq} \mathbb{E}_{\sigma}\left[\sup_{\|\mathbf{w}\|_2 \leq W, \|\mathbf{w}'\|_2 \leq W} \mathbf{w}^\top \left(\frac{1}{n}\sum_{i=1}^n \sigma_i \mathbf{x}_i \mathbf{x}_i^\top\right) \mathbf{w}'\right] \cdot \begin{cases} 1, & \text{if } 1 \leq p \leq 2 \\ d^{1-2/p}, & \text{else if } p > 2 \end{cases}
$$

$$
\overset{Lemma\ 13}{=} \frac{W^2}{n}\mathbb{E}_{\sigma}\left[\left\|\sum_{i=1}^n \sigma_i \mathbf{x}_i \mathbf{x}_i^\top\right\|_2\right] \cdot \begin{cases} 1, & \text{if } 1 \leq p \leq 2 \\ d^{1-2/p}, & \text{else if } p > 2 \end{cases}. \tag{27}
$$

We now look for a more explicit upper bound of the above Rademacher complexity depending on the dimension $d$ and on a norm of covariance of data points $\mathbf{x}_1, \ldots, \mathbf{x}_n$. To do so, we introduce some notations before applying a matrix Bernstein inequality (Theorem 6.1.1 of Tropp et al. (2015))

recalled in Theorem 4. Let $\mathbf{Z}_i := \sigma_i \mathbf{x}_i \mathbf{x}_i^\top \in \mathbb{R}^{d \times d}$ for all $i \in [n]$. These random matrices are symmetric, independent, have zero mean and are such that for all $i \in [n]$. Moreover, let $\mathbf{Y} := \sum_{i=1}^n \mathbf{Z}_i$. For each $\mathbf{Z}_i$, we notice that it has bounded spectral norm:

$$\|\mathbf{Z}_i\|_2 = \sqrt{\lambda_{\max}(\mathbf{Z}_i^2)} = \sqrt{\lambda_{\max}(\mathbf{x}_i \mathbf{x}_i^\top \mathbf{x}_i \mathbf{x}_i^\top)} = \|\mathbf{x}_i\|_2^2 \leq \max_{j \in [n]} \|\mathbf{x}_j\|_2^2 = \|\mathbf{X}\|_{2,\infty}^2 \quad,$$

so that according to matrix Bernstein inequality, we get the desired bound

$$\hat{\mathfrak{R}}(f \circ \mathcal{H}\Delta\mathcal{H}) \overset{(14)}{\leq} \frac{W^2}{n} \left( \sqrt{2 \left\| \sum_{i=1}^n (\mathbf{x}_i \mathbf{x}_i^\top)^2 \right\|_2 \log(2d)} + \frac{1}{3} \|\mathbf{X}\|_{2,\infty}^2 \log(2d) \right)$$
$$\times \begin{cases} 1, & \text{if } 1 \leq p \leq 2 \\ d^{1-2/p}, & \text{else if } p > 2 \end{cases} .$$

$\square$

## G.2  Proof of the upper bound of Theorem 1

Alike the analysis of Theorem 7 from Awasthi et al. (2020), the below study uses the notion of coverings. For completeness sake, we recall its definition.

**Definition 6** ($\rho$-covering). *Let $\rho > 0$ and let $(V, \|.\|)$ be a normed space. A set $\mathcal{C} \subseteq V$ is an $\epsilon$-covering of $V$ if for any $v \in V$, there exists $v' \in \mathcal{C}$ such that $\|v - v'\| \leq \rho$.*

We also copy Lemma 6 from Awasthi et al. (2020) dealing with the size of coverings of balls.

**Lemma 20.** *Let $\rho > 0$. Let $\mathcal{B} \subseteq \mathbb{R}^d$ be a the ball of radius $R \geq 0$ in a norm $\|.\|$ and let $\mathcal{C}$ be one of the smallest $\rho$-covering of $\mathcal{B}$ w.r.t. $\|.\|$. Then,*

$$|\mathcal{C}| \leq \left( \frac{3R}{\rho} \right)^d .$$

Now we are ready to present the proof of upper bound of the adversarial Rademacher complexity for linear binary classification.

*Proof of the upper bound of Theorem 1.* In this proof, we consider the linear hypothesis class were the norm of the models is controlled by a general $\ell_p$-norm for $p > 0$. Let $\mathcal{B}_p(W) := \{\mathbf{w} \in \mathbb{R}^d : \|\mathbf{w}\|_p \leq W\}$, the hypothesis class defined in (5) then writes

$$\mathcal{H} := \{h_\mathbf{w} : \mathbf{x} \mapsto \langle \mathbf{w}, \mathbf{x} \rangle : \mathbf{w} \in \mathcal{B}_p(W)\} .$$

Similarly, let $\mathcal{B}_\infty(\epsilon) := \{\boldsymbol{\delta} \in \mathbb{R}^d : \|\boldsymbol{\delta}\|_\infty \leq \epsilon\}$.

Recall that we define in (8)

$$\hat{\mathfrak{R}}_S(\tilde{f} \circ \mathcal{H}\Delta\mathcal{H}) = \mathbb{E}_\sigma \left[ \sup_{\mathbf{w},\mathbf{w}' \in \mathcal{B}_p(W)^2} \frac{1}{n} \sum_{i=1}^n \sigma_i \min_{\boldsymbol{\delta} \in \mathcal{B}_\infty(\epsilon)} \mathbf{w}^T(\mathbf{x}_i + \boldsymbol{\delta})\mathbf{w}'^T(\mathbf{x}_i + \boldsymbol{\delta}) \right]$$

$$= \mathbb{E}_\sigma \left[ \sup_{\mathbf{w},\mathbf{w}' \in \mathcal{B}_p(W)^2} \frac{1}{n} \sum_{i=1}^n \sigma_i (\mathbf{x}_i^\top \mathbf{w}\mathbf{w}'^\top \mathbf{x}_i + \min_{\boldsymbol{\delta} \in \mathcal{B}_\infty(\epsilon)} \boldsymbol{\delta}^\top \mathbf{w}\mathbf{w}'^\top \boldsymbol{\delta} + \mathbf{x}_i^\top (\mathbf{w}\mathbf{w}'^\top + \mathbf{w}'\mathbf{w}^\top)\boldsymbol{\delta}) \right]$$

$$\leq \hat{\mathfrak{R}}_S(f \circ \mathcal{H}\Delta\mathcal{H}) + \underbrace{\mathbb{E}_\sigma \left[ \sup_{\mathbf{w},\mathbf{w}' \in \mathcal{B}_p(W)^2} \frac{1}{n} \sum_{i=1}^n \sigma_i \min_{\boldsymbol{\delta} \in \mathcal{B}_\infty(\epsilon)} (\boldsymbol{\delta}^\top \mathbf{w}\mathbf{w}'^\top \boldsymbol{\delta} + \mathbf{x}_i^\top (\mathbf{w}\mathbf{w}'^\top + \mathbf{w}'\mathbf{w}^\top)\boldsymbol{\delta}) \right]}_{A} .$$

Now we examine the upper bound of the second term $A$ using the notion of covering recalled in Definition 6. Let $\mathcal{C}$ be a $\rho$-covering of the $\ell_p$ ball $\mathcal{B}_p(W)$ w.r.t. the $\ell_p$-norm, with $\rho > 0$. Let us define

$$\psi_i(\mathbf{w}, \mathbf{w}') := \min_{\boldsymbol{\delta} \in \mathcal{B}_\infty(\epsilon)} \boldsymbol{\delta}^\top \mathbf{w}\mathbf{w}'^\top \boldsymbol{\delta} + \mathbf{x}_i^\top (\mathbf{w}\mathbf{w}'^\top + \mathbf{w}'\mathbf{w}^\top)\boldsymbol{\delta} . \tag{28}$$

Thus we can rewrite $A$ as

$$A = \mathbb{E}_\sigma \left[ \sup_{\mathbf{w}, \mathbf{w}' \in \mathcal{B}_p(W)^2} \frac{1}{n} \sum_{i=1}^n \sigma_i \psi_i(\mathbf{w}, \mathbf{w}') \right]$$

$$= \mathbb{E}_\sigma \left[ \sup_{\substack{\mathbf{w}, \mathbf{w}' \in \mathcal{B}_p(W)^2 \\ \mathbf{w}_c, \mathbf{w}'_c \in \mathcal{C}^2 : \|\mathbf{w} - \mathbf{w}_c\|_p, \|\mathbf{w}' - \mathbf{w}'_c\|_p \leq \rho}} \frac{1}{n} \sum_{i=1}^n \sigma_i \left( \psi_i(\mathbf{w}_c, \mathbf{w}'_c) + \psi_i(\mathbf{w}, \mathbf{w}') - \psi_i(\mathbf{w}_c, \mathbf{w}'_c) \right) \right],$$

where $\mathbf{w}_c$, respectively $\mathbf{w}'_c$, is the closest element to $\mathbf{w}$, resp. $\mathbf{w}'$, in $\mathcal{C}$. Using the subadditivity of the supremum, we get

$$A \leq \mathbb{E}_\sigma \left[ \sup_{\tilde{\mathbf{w}}, \tilde{\mathbf{w}}' \in \mathcal{C}^2} \frac{1}{n} \sum_{i=1}^n \sigma_i \psi_i(\tilde{\mathbf{w}}, \tilde{\mathbf{w}}') \right] + \mathbb{E}_\sigma \left[ \sup_{\mathbf{w}, \mathbf{w}' \in \mathcal{B}_p(W)^2} \frac{1}{n} \sum_{i=1}^n \sigma_i \left( \psi_i(\mathbf{w}, \mathbf{w}') - \psi_i(\mathbf{w}_c, \mathbf{w}'_c) \right) \right]$$

$$\leq \mathbb{E}_\sigma \left[ \sup_{\tilde{\mathbf{w}}, \tilde{\mathbf{w}}' \in \mathcal{C}^2} \frac{1}{n} \sum_{i=1}^n \sigma_i \psi_i(\tilde{\mathbf{w}}, \tilde{\mathbf{w}}') \right] + \sup_{\mathbf{w}, \mathbf{w}' \in \mathcal{B}_p(W)^2} \frac{1}{n} \sum_{i=1}^n |\psi_i(\mathbf{w}, \mathbf{w}') - \psi_i(\mathbf{w}_c, \mathbf{w}'_c)|$$

$$\leq \underbrace{\mathbb{E}_\sigma \left[ \sup_{\tilde{\mathbf{w}}, \tilde{\mathbf{w}}' \in \mathcal{C}^2} \frac{1}{n} \sum_{i=1}^n \sigma_i \psi_i(\tilde{\mathbf{w}}, \tilde{\mathbf{w}}') \right]}_{(I)} + \underbrace{\max_{i \in [n]} \sup_{\mathbf{w}, \mathbf{w}' \in \mathcal{B}_p(W)^2} |\psi_i(\mathbf{w}, \mathbf{w}') - \psi_i(\mathbf{w}_c, \mathbf{w}'_c)|}_{(II)} . \quad (29)$$

where we recall that $\mathbf{w}_c$, resp. $\mathbf{w}'_c$, is the closest vector to $\mathbf{w}$, resp. $\mathbf{w}'$, in $\mathcal{C}$.

**Bounding $(I)$:** We first need to bound the left-hand side term $(I)$. We introduce the vector

$$\boldsymbol{\psi}(\tilde{\mathbf{w}}, \tilde{\mathbf{w}}') := [\psi_1(\tilde{\mathbf{w}}, \tilde{\mathbf{w}}'), \ldots, \psi_n(\tilde{\mathbf{w}}, \tilde{\mathbf{w}}')]^\top \in \mathbb{R}^n .$$

By Massart's lemma (Lemma 5.2 of Massart (2000)), we are able to control the first term $(I)$ in (29):

$$(I) = \mathbb{E}_\sigma \left[ \sup_{\tilde{\mathbf{w}}, \tilde{\mathbf{w}}' \in \mathcal{C}^2} \frac{1}{n} \sum_{i=1}^n \sigma_i \psi_i(\tilde{\mathbf{w}}, \tilde{\mathbf{w}}') \right] \leq \frac{K \sqrt{2 \log(|\mathcal{C}|^2)}}{n} , \quad (30)$$

with $K$ given by the largest $\ell_2$-norm of $\boldsymbol{\psi}$ over the covering $\mathcal{C}^2$, that is

$$K^2 = \max_{\tilde{\mathbf{w}} \tilde{\mathbf{w}}' \in \mathcal{C}^2} \|\boldsymbol{\psi}\|_2^2 = \max_{\tilde{\mathbf{w}} \tilde{\mathbf{w}}' \in \mathcal{C}^2} \sum_{i=1}^n \psi_i(\tilde{\mathbf{w}}, \tilde{\mathbf{w}}')^2 . \quad (31)$$

Now we examine the upper and lower bound of $\psi_i(\tilde{\mathbf{w}}, \tilde{\mathbf{w}}')$. For upper bound, by taking $\boldsymbol{\delta} = 0$ we know that $\psi_i(\tilde{\mathbf{w}}, \tilde{\mathbf{w}}')$ is non-positive. Thus, we only have to control how negative this term can be. Let $\tilde{\mathbf{w}}, \tilde{\mathbf{w}}' \in \mathcal{C}^2$, we have

$$\psi_i(\tilde{\mathbf{w}}, \tilde{\mathbf{w}}') = \min_{\boldsymbol{\delta} \in \mathcal{B}_\infty(\epsilon)} \boldsymbol{\delta}^\top \tilde{\mathbf{w}} \tilde{\mathbf{w}}'^\top \boldsymbol{\delta} + \mathbf{x}_i^\top (\tilde{\mathbf{w}} \tilde{\mathbf{w}}'^\top + \tilde{\mathbf{w}}' \tilde{\mathbf{w}}^\top) \boldsymbol{\delta}$$

$$\geq \min_{\boldsymbol{\delta} \in \mathcal{B}_\infty(\epsilon)} \boldsymbol{\delta}^\top \tilde{\mathbf{w}} \tilde{\mathbf{w}}'^\top \boldsymbol{\delta} + \min_{\boldsymbol{\delta} \in \mathcal{B}_\infty(\epsilon)} \mathbf{x}_i^\top (\tilde{\mathbf{w}} \tilde{\mathbf{w}}'^\top + \tilde{\mathbf{w}}' \tilde{\mathbf{w}}^\top) \boldsymbol{\delta}$$

$$\overset{Lemma\ 10}{=} \min_{\boldsymbol{\delta} \in \mathcal{B}_\infty(\epsilon)} \boldsymbol{\delta}^\top \tilde{\mathbf{w}} \tilde{\mathbf{w}}'^\top \boldsymbol{\delta} - \epsilon \left\| (\tilde{\mathbf{w}} \tilde{\mathbf{w}}'^\top + \tilde{\mathbf{w}}' \tilde{\mathbf{w}}^\top) \mathbf{x}_i \right\|_1 . \quad (32)$$

We focus on the first term which is a quadratic optimization problem under infinite norm constraints. For all $\boldsymbol{\delta}, \mathbf{w}, \mathbf{w}' \in \mathcal{B}_\infty(\epsilon) \times \mathcal{B}_p(W)^2$, this quadratic form can be lower bounded by calling Hölder's inequality twice and norm equivalence, that is if $p^* \geq 1$ we have $\|\mathbf{v}\|_{p^*} \leq d^{1/p^*} \|\mathbf{v}\|_\infty$:

$$\boldsymbol{\delta}^\top \tilde{\mathbf{w}} \tilde{\mathbf{w}}'^\top \boldsymbol{\delta} \geq -|\boldsymbol{\delta}^\top \tilde{\mathbf{w}} \tilde{\mathbf{w}}'^\top \boldsymbol{\delta}|$$

$$\overset{Lemma\ 7}{\geq} -\|\boldsymbol{\delta}\|_{p^*}^2 \|\tilde{\mathbf{w}}\|_p \|\tilde{\mathbf{w}}'\|_p$$

$$\overset{Lemma\ 8}{\geq} \begin{cases} -d^{2/p^*} \|\boldsymbol{\delta}\|_\infty^2 \|\tilde{\mathbf{w}}\|_p \|\tilde{\mathbf{w}}'\|_p & \text{if } p > 1 \\ -\|\boldsymbol{\delta}\|_\infty^2 \|\tilde{\mathbf{w}}\|_1 \|\tilde{\mathbf{w}}'\|_1 & \text{else if } p = 1 \end{cases}$$

$$\overset{\boldsymbol{\delta} \in \mathcal{B}_\infty(\epsilon),\ \tilde{\mathbf{w}}, \tilde{\mathbf{w}}' \in \mathcal{B}_p(W)^2}{\geq} \begin{cases} -d^{2/p^*} \epsilon^2 W^2 & \text{if } p > 1 \\ -\epsilon^2 W^2 & \text{else if } p = 1 \end{cases} .$$

Using the same tools, we now study the second term

$$-\left\|\tilde{\mathbf{w}}\tilde{\mathbf{w}}'^{\top}\mathbf{x}_i\right\|_1 = -|\langle \tilde{\mathbf{w}}', \mathbf{x}_i\rangle|\,\|\tilde{\mathbf{w}}\|_1$$

$$\overset{Lemma\ 7}{\geq} -\|\mathbf{x}_i\|_{p^*}\,\|\tilde{\mathbf{w}}'\|_p\,\|\tilde{\mathbf{w}}\|_1$$

$$\overset{Lemma\ 8}{\geq} \begin{cases} -d^{1/p^*}\,\|\mathbf{x}_i\|_{p^*}\,\|\tilde{\mathbf{w}}'\|_p\,\|\tilde{\mathbf{w}}\|_p & \text{if } p > 1 \\ -\,\|\mathbf{x}_i\|_\infty\,\|\tilde{\mathbf{w}}'\|_1\,\|\tilde{\mathbf{w}}\|_1 & \text{else if } p = 1 \end{cases}$$

$$\overset{\tilde{\mathbf{w}},\tilde{\mathbf{w}}'\in\mathcal{B}_p(W)^2}{\geq} \begin{cases} -d^{1/p^*}W^2\,\|\mathbf{x}_i\|_{p^*} & \text{if } p > 1 \\ -W^2\,\|\mathbf{x}_i\|_\infty & \text{else if } p = 1 \end{cases},$$

and symmetrically we get the same bound for $-\left\|\tilde{\mathbf{w}}'\tilde{\mathbf{w}}^{\top}\mathbf{x}_i\right\|_1$. By taking the convention that $d^{1/p^*} = 1$ if $p = 1$ (*i.e.* $p^* = \infty$), we drop the disjunction between $p = 1$ and $p > 1$ in what follows. Combining (32) and the above two inequalities, the auxiliary function $\psi_i$ (28) can be lower bounded after applying the triangle inequality:

$$\psi_i(\tilde{\mathbf{w}}, \tilde{\mathbf{w}}') \geq -d^{2/p^*}\epsilon^2 W^2 - 2\epsilon d^{1/p^*} W^2 \,\|\mathbf{x}_i\|_{p^*}\ .$$

So that we get

$$\psi_i(\tilde{\mathbf{w}}, \tilde{\mathbf{w}}')^2 \leq (d^{2/p^*}\epsilon^2 W^2 + 2\epsilon d^{1/p^*} W^2 \,\|\mathbf{x}_i\|_{p^*})^2$$

$$\leq \epsilon^2 d^{2/p^*} W^4 (\epsilon d^{1/p^*} + 2\max_{j\in[n]} \|\mathbf{x}_j\|_{p^*})^2$$

$$= \epsilon^2 d^{2/p^*} W^4 (\epsilon d^{1/p^*} + 2\,\|\mathbf{X}\|_{p^*,\infty})^2\ .$$

Finally we get the following upper bound for $K$ defined in (31):

$$K \leq \sqrt{n}\epsilon d^{1/p^*} W^2 (\epsilon d^{1/p^*} + 2\,\|\mathbf{X}\|_{p^*,\infty})\ ,$$

which, jointly with the application of Lemma 20 implies the upper bound for $(I)$:

$$(I) \overset{(30)}{\leq} \frac{K\sqrt{2\log|\mathcal{C}|^2}}{n} \leq \frac{\epsilon d^{1/p^*} W^2 (\epsilon d^{1/p^*} + 2\,\|\mathbf{X}\|_{p^*,\infty})}{\sqrt{n}}\sqrt{4d\log(3W/\rho)}\ . \qquad (33)$$

**Bounding** $(II)$. Now we turn to bounding the second term of (29). Let $\mathbf{w}, \mathbf{w}' \in \mathcal{B}_p(W)^2$ and let $\mathbf{w}_c$, resp. $\mathbf{w}'_c$, be the closest element to $\mathbf{w}$, resp. $\mathbf{w}'$, in $\mathcal{C}$. Let us define an "implicit" minimizer w.r.t. $\boldsymbol{\delta}$ (the objective being continuous over a closed ball it is attained) for $\psi_i(\mathbf{w}_c, \mathbf{w}'_c)$:

$$\boldsymbol{\delta}_c^* := \underset{\|\boldsymbol{\delta}_c\|_\infty \leq \epsilon}{\arg\min}\ \boldsymbol{\delta}_c^{\top} \mathbf{w}_c\mathbf{w}'^{\top}_c \boldsymbol{\delta}_c + \mathbf{x}_i^{\top}(\mathbf{w}_c\mathbf{w}'^{\top}_c + \mathbf{w}'_c\mathbf{w}^{\top}_c)\boldsymbol{\delta}_c\ . \qquad (34)$$

Thus, we have

$$\psi_i(\mathbf{w}, \mathbf{w}') - \psi_i(\mathbf{w}_c, \mathbf{w}'_c)$$

$$\overset{(34)}{=} \min_{\boldsymbol{\delta}\in\mathcal{B}_\infty(\epsilon)} \boldsymbol{\delta}^{\top}\mathbf{w}\mathbf{w}'^{\top}\boldsymbol{\delta} + \mathbf{x}_i^{\top}(\mathbf{w}\mathbf{w}'^{\top} + \mathbf{w}'\mathbf{w}^{\top})\boldsymbol{\delta} - (\boldsymbol{\delta}_c^*)^{\top}\mathbf{w}_c\mathbf{w}'^{\top}_c \boldsymbol{\delta}_c^* - \mathbf{x}_i^{\top}(\mathbf{w}_c\mathbf{w}'^{\top}_c + \mathbf{w}'_c\mathbf{w}^{\top}_c)\boldsymbol{\delta}_c^*$$

$$\leq (\boldsymbol{\delta}_c^*)^{\top}\mathbf{w}\mathbf{w}'^{\top}\boldsymbol{\delta}_c^* + \mathbf{x}_i^{\top}(\mathbf{w}\mathbf{w}'^{\top} + \mathbf{w}'\mathbf{w}^{\top})\boldsymbol{\delta}_c^* - (\boldsymbol{\delta}_c^*)^{\top}\mathbf{w}_c\mathbf{w}'^{\top}_c \boldsymbol{\delta}_c^* - \mathbf{x}_i^{\top}(\mathbf{w}_c\mathbf{w}'^{\top}_c + \mathbf{w}'_c\mathbf{w}^{\top}_c)\boldsymbol{\delta}_c^*$$

$$= (\boldsymbol{\delta}_c^*)^{\top}(\mathbf{w}\mathbf{w}'^{\top} - \mathbf{w}_c\mathbf{w}'^{\top}_c)\boldsymbol{\delta}_c^* + \mathbf{x}_i^{\top}(\mathbf{w}\mathbf{w}'^{\top} - \mathbf{w}_c\mathbf{w}'^{\top}_c + \mathbf{w}'\mathbf{w}^{\top} - \mathbf{w}'_c\mathbf{w}^{\top}_c)\boldsymbol{\delta}_c^*$$

$$= (\boldsymbol{\delta}_c^*)^{\top}\left(\mathbf{w}(\mathbf{w}' - \mathbf{w}'_c)^{\top} - (\mathbf{w}_c - \mathbf{w})\mathbf{w}'^{\top}_c\right)\boldsymbol{\delta}_c^*$$

$$\quad + \mathbf{x}_i^{\top}\left(\mathbf{w}(\mathbf{w}' - \mathbf{w}'_c)^{\top} - (\mathbf{w}_c - \mathbf{w})\mathbf{w}'^{\top}_c + \mathbf{w}'(\mathbf{w} - \mathbf{w}_c)^{\top} - (\mathbf{w}'_c - \mathbf{w}')\mathbf{w}^{\top}_c\right)\boldsymbol{\delta}_c^*\ .$$

We focus on upper bounding a single term of the ones appearing above. By applying Hölder's inequality twice and norm equivalence we get:

$$|(\boldsymbol{\delta}_c^*)^{\top}\mathbf{w}(\mathbf{w}' - \mathbf{w}'_c)^{\top}\boldsymbol{\delta}_c^*| \overset{Lemma\ 7}{\leq} \|\mathbf{w}\|_p\,\|\boldsymbol{\delta}_c^*\|_{p^*}^2\,\|\mathbf{w}' - \mathbf{w}'_c\|_p \overset{Lemma\ 8}{\leq} \rho\epsilon^2 d^{2/p^*} W\ ,$$

where in the last line we used that $\|\mathbf{w} - \mathbf{w}_c\|_p \leq \rho$, by the definition of the $\rho$-covering of the ball $\mathcal{B}_p(W)$ w.r.t. the $\ell_p$-norm. Proceeding identically with other terms involving $\mathbf{x}_i$ we get

$$|\mathbf{x}_i^\top \mathbf{w}(\mathbf{w}' - \mathbf{w}_c')^\top \boldsymbol{\delta}_c^*| \overset{Lemma\ 7}{\leq} \|\mathbf{x}_i\|_{p^*} \|\mathbf{w}\|_p \|\mathbf{w}' - \mathbf{w}_c'\|_p \|\boldsymbol{\delta}_c^*\|_{p^*} \overset{Lemma\ 8}{\leq} \rho \epsilon d^{1/p^*} W \|\mathbf{x}_i\|_{p^*} \quad,$$

finally get that

$$\psi_i(\mathbf{w}, \mathbf{w}') - \psi_i(\mathbf{w}_c, \mathbf{w}_c') \leq 2\rho\epsilon^2 d^{2/p^*} W + 4\rho\epsilon d^{1/p^*} W \|\mathbf{x}_i\|_{p^*} \quad.$$

Similarly we can prove the same bound holds for other side of the difference (by using an "implicit" minimizer of $\psi_i(\mathbf{w}, \mathbf{w}')$). Thus we are able to control $(II)$:

$$(II) \overset{(29)}{=} \max_{i \in [n]} \sup_{\mathbf{w}, \mathbf{w}' \in \mathcal{B}_p(W)^2} |\psi_i(\mathbf{w}, \mathbf{w}') - \psi_i(\mathbf{w}_c, \mathbf{w}_c')| \leq 2\rho\epsilon d^{1/p^*} W (\epsilon d^{1/p^*} + 2\|\mathbf{X}\|_{p^*, \infty}) \quad. \quad (35)$$

And finally, we proved that

$$A \overset{(33)+(35)}{\leq} 2\epsilon d^{1/p^*} W (\epsilon d^{1/p^*} + 2\|\mathbf{X}\|_{p^*, \infty}) \left( \rho + \sqrt{\frac{d}{n}} W \sqrt{\log(3W/\rho)} \right) \quad,$$

which concludes the first part of the proof if we choose $\rho = W/\sqrt{n}$:

$$A \leq 2\epsilon \frac{d^{1/p^*}}{\sqrt{n}} W^2 \left( 1 + \sqrt{d}\sqrt{\log(3\sqrt{n})} \right) (\epsilon d^{1/p^*} + 2\|\mathbf{X}\|_{p^*, \infty}) \quad.$$

$\square$

### G.3 PROOF OF THE LOWER BOUND OF THEOREM 1

In this subsection we present the proof of lower bound of the adversarial Rademacher complexity for binary classification under linear hypothesis.

*Proof of the lower bound of Theorem 1.* Now we are going to prove the lower bound result of adversarial Rademacher complexity. Recall the definition of non-adversarial Rademacher complexity

$$\hat{\mathfrak{R}}_\mathcal{D}(f \circ \mathcal{H}\Delta\mathcal{H}) = \mathbb{E}\left[ \sup_{\|\mathbf{w}\|_p \leq W, \|\mathbf{w}'\|_p \leq W} \frac{1}{n} \sum_{i=1}^n \sigma_i \mathbf{w}^\top \mathbf{x}_i \mathbf{x}_i^\top \mathbf{w}' \right]$$

$$= \frac{1}{n} \mathbb{E}\left[ \sup_{\|\mathbf{w}\|_p \leq W, \|\mathbf{w}'\|_p \leq W} \mathbf{w}^\top \left( \sum_{i=1}^n \sigma_i \mathbf{x}_i \mathbf{x}_i^\top \right) \mathbf{w}' \right]$$

$$\overset{(20)}{\leq} \frac{1}{n} \mathbb{E}\left[ \sup_{\|\mathbf{w}\|_2 \leq W, \|\mathbf{w}'\|_2 \leq 2W} \mathbf{w}^\top \left( \sum_{i=1}^n \sigma_i \mathbf{x}_i \mathbf{x}_i^\top \right) \mathbf{w}' \right] \times \begin{cases} 1, & \text{if } 1 \leq p \leq 2 \\ d^{1-2/p}, & \text{else if } p > 2 \end{cases}$$

$$\overset{Lemma\ 13}{=} \frac{W^2}{n} \mathbb{E}\left[ \left\| \sum_{i=1}^n \sigma_i \mathbf{x}_i \mathbf{x}_i^\top \right\|_2 \right] \times \begin{cases} 1, & \text{if } 1 \leq p \leq 2 \\ d^{1-2/p}, & \text{else if } p > 2 \end{cases} \quad,$$

due to equivalence of norms. Now, we denote $\mathbf{v}^*$ such that $\mathbf{v}^* = \arg\max_{\|\mathbf{w}\|_p \leq W, \|\mathbf{w}'\|_p \leq W} \frac{1}{n} \sum_{i=1}^n \sigma_i \mathbf{w}^\top \mathbf{x}_i \mathbf{x}_i^\top \mathbf{w}'$. According to Lemma 13 the maximum value of $\frac{1}{n} \sum_{i=1}^n \sigma_i \mathbf{w}^\top \mathbf{x}_i \mathbf{x}_i^\top \mathbf{w}'$ is:

$$\sup_{\|\mathbf{w}\|_2 \leq W, \|\mathbf{w}'\|_2 \leq W} \frac{1}{n} \sum_{i=1}^n \sigma_i \mathbf{w}^\top \mathbf{x}_i \mathbf{x}_i^\top \mathbf{w}' = \frac{W^2}{n} \left\| \sum_{i=1}^n \sigma_i \mathbf{x}_i \mathbf{x}_i^\top \right\|_2$$

and if we define $\mathbf{S}(\boldsymbol{\sigma}) := \sum_{i=1}^n \sigma_i \mathbf{x}_i \mathbf{x}_i^\top$ the maxima is attained when $\mathbf{v}^* = W\mathbf{v}_{\max}(\mathbf{S}(\boldsymbol{\sigma})^2)$ is an eigenvector of $\ell_2$-norm $W$ associated to the largest eigenvalue of $\mathbf{S}(\boldsymbol{\sigma})^2$.

Now, we switch to adversarial Rademacher:

$$\hat{\mathfrak{R}}_{\mathcal{D}}(\tilde{f} \circ \mathcal{H}\Delta\mathcal{H})$$

$$= \mathbb{E}\left[\sup_{\|\mathbf{w}\|_p \leq W, \|\mathbf{w}'\|_p \leq W} \frac{1}{n}\sum_{i=1}^{n} \sigma_i \min_{\|\boldsymbol{\delta}\|_\infty \leq \epsilon} \mathbf{w}^\top(\mathbf{x}_i + \boldsymbol{\delta})(\mathbf{x}_i + \boldsymbol{\delta})^\top \mathbf{w}'\right]$$

$$\geq \mathbb{E}\left[\sup_{\|\mathbf{w}\|_2 \leq W, \|\mathbf{w}'\|_2 \leq W} \frac{1}{n}\sum_{i=1}^{n} \sigma_i \min_{\|\boldsymbol{\delta}\|_\infty \leq \epsilon} \mathbf{w}^\top(\mathbf{x}_i + \boldsymbol{\delta})(\mathbf{x}_i + \boldsymbol{\delta})^\top \mathbf{w}'\right] \times \begin{cases} d^{1-2/p}, & \text{if } 1 \leq p \leq 2 \\ 1, & \text{else if } p > 2 \end{cases}$$

$$\geq \mathbb{E}\left[\sup_{\|\mathbf{w}\|_2 \leq W} \frac{1}{n}\sum_{i=1}^{n} \sigma_i \min_{\|\boldsymbol{\delta}\|_\infty \leq \epsilon} \mathbf{w}^\top(\mathbf{x}_i + \boldsymbol{\delta})(\mathbf{x}_i + \boldsymbol{\delta})^\top \mathbf{w}\right] \times \begin{cases} d^{1-2/p}, & \text{if } 1 \leq p \leq 2 \\ 1, & \text{else if } p > 2 \end{cases}$$

where in the first inequality we used that, if $1 \leq p \leq 2$, then $\mathcal{B}_2(W) \subseteq \mathcal{B}_p(d^{1/p-1/2}W)$ and else when $p > 2$, we simply have that $\mathcal{B}_2(W) \subseteq \mathcal{B}_p(W)$. According to Lemma 17, we have:

$$\hat{\mathfrak{R}}_{\mathcal{D}}(\tilde{f} \circ \mathcal{H}\Delta\mathcal{H}) \geq \mathbb{E}\left[\sup_{\|\mathbf{w}\|_2 \leq 2W} \frac{1}{n}\sum_{i=1}^{n} \sigma_i \left(\mathbf{w}^\top \mathbf{x}_i - \mathbf{w}^\top \mathbf{x}_i \min\left\{1, \frac{\epsilon\|\mathbf{w}\|_1}{|\mathbf{w}^\top \mathbf{x}_i|}\right\}\right)^2\right]$$

**Case I:** $1 \leq p \leq 2$.

First, to avoid confusion in different Rademacher variables, let us use $\boldsymbol{\sigma}'$ and $\boldsymbol{\sigma}$ to denote the Rademacher variables in $\hat{\mathfrak{R}}_{\mathcal{D}}(\tilde{f} \circ \mathcal{H}\Delta\mathcal{H})$ and $\hat{\mathfrak{R}}_{\mathcal{D}}(f \circ \mathcal{H}\Delta\mathcal{H})$. Then, let us define $\mathbf{v}'^* := W\mathbf{v}_{\max}(\mathbf{S}(\boldsymbol{\sigma}')^2)$. Then we consider the gap:

$$\hat{\mathfrak{R}}_{\mathcal{D}}(\tilde{f} \circ \mathcal{H}\Delta\mathcal{H}) - \hat{\mathfrak{R}}_{\mathcal{D}}(f \circ \mathcal{H}\Delta\mathcal{H})$$

$$= \mathbb{E}_{\boldsymbol{\sigma}'}\left[\sup_{\|\mathbf{w}\|_p \leq 2W} \frac{1}{n}\sum_{i=1}^{n} \sigma_i' \left(\mathbf{w}^\top \mathbf{x}_i - \mathbf{w}^\top \mathbf{x}_i \min\left\{1, \frac{\epsilon\|\mathbf{w}\|_1}{|\mathbf{w}^\top \mathbf{x}_i|}\right\}\right)^2\right]$$

$$- \mathbb{E}_{\boldsymbol{\sigma}}\left[\sup_{\|\mathbf{w}\|_2 \leq W, \|\mathbf{w}'\|_2 \leq W} \frac{1}{n}\sum_{i=1}^{n} \sigma_i \mathbf{w}^\top \mathbf{x}_i \mathbf{x}_i^\top \mathbf{w}'\right]$$

$$\geq \mathbb{E}_{\boldsymbol{\sigma}'}\left[\frac{1}{n}\sum_{i=1}^{n} \sigma_i' \left(\mathbf{v}'^{*\top}\mathbf{x}_i - \mathbf{v}'^{*\top}\mathbf{x}_i \min\left\{1, \frac{\epsilon\|\mathbf{v}'^*\|_1}{|\mathbf{v}'^{*\top}\mathbf{x}_i|}\right\}\right)^2\right] - \mathbb{E}_{\boldsymbol{\sigma}}\left[\frac{1}{n}\sum_{i=1}^{n} \sigma_i \mathbf{v}^{*\top}\mathbf{x}_i \mathbf{x}_i^\top \mathbf{v}^*\right]$$

$$= \frac{W^2}{n}\mathbb{E}_{\boldsymbol{\sigma}'}\left[\sum_{i=1}^{n} \sigma_i' \left(-2(\mathbf{v}_{\max}(\mathbf{S}(\boldsymbol{\sigma}')^2)^\top \mathbf{x}_i)^2 \min\left\{1, \frac{\epsilon\|\mathbf{v}_{\max}(\mathbf{S}(\boldsymbol{\sigma}')^2)\|_1}{|\mathbf{v}_{\max}(\mathbf{S}(\boldsymbol{\sigma}')^2)^\top \mathbf{x}_i|}\right\}\right)\right]$$

$$+ \frac{W^2}{n}\mathbb{E}_{\boldsymbol{\sigma}'}\left[\sum_{i=1}^{n} \sigma_i' \left((\mathbf{v}_{\max}(\mathbf{S}(\boldsymbol{\sigma}')^2)^\top \mathbf{x}_i)^2 \min\left\{1, \frac{\epsilon\|\mathbf{v}_{\max}(\mathbf{S}(\boldsymbol{\sigma}')^2))\|_1}{|\mathbf{v}_{\max}(\mathbf{S}(\boldsymbol{\sigma}')^2)^\top \mathbf{x}_i|}\right\}^2\right)\right]$$

Let us define

$$I(\boldsymbol{\sigma}') := \sum_{i=1}^{n} \sigma_i' \left(-2(\mathbf{v}_{\max}(\mathbf{S}(\boldsymbol{\sigma}')^2)^\top \mathbf{x}_i)^2 \min\left\{1, \frac{\epsilon\|\mathbf{v}_{\max}(\mathbf{S}(\boldsymbol{\sigma}')^2)\|_1}{|\mathbf{v}_{\max}(\mathbf{S}(\boldsymbol{\sigma}')^2)^\top \mathbf{x}_i|}\right\}\right)$$

and

$$J(\boldsymbol{\sigma}') := \sum_{i=1}^{n} \sigma_i' \left((\mathbf{v}_{\max}(\mathbf{S}(\boldsymbol{\sigma}')^2)^\top \mathbf{x}_i)^2 \min\left\{1, \frac{\epsilon\|\mathbf{v}_{\max}(\mathbf{S}(\boldsymbol{\sigma}')^2)\|_1}{|\mathbf{v}_{\max}(\mathbf{S}(\boldsymbol{\sigma}')^2)^\top \mathbf{x}_i|}\right\}^2\right) ,$$

so that

$$\hat{\mathfrak{R}}_{\mathcal{D}}(\tilde{f} \circ \mathcal{H}\Delta\mathcal{H}) - \hat{\mathfrak{R}}_{\mathcal{D}}(f \circ \mathcal{H}\Delta\mathcal{H}) \geq \frac{W^2}{n}\left(\mathbb{E}_{\boldsymbol{\sigma}'}\left[I(\boldsymbol{\sigma}')\right] + \mathbb{E}_{\boldsymbol{\sigma}'}\left[J(\boldsymbol{\sigma}')\right]\right) .$$

We are now going to prove that $I(\boldsymbol{\sigma}) + I(-\boldsymbol{\sigma}) = 0$ and $J(\boldsymbol{\sigma}) + J(-\boldsymbol{\sigma}) = 0$.

First we know that $\mathbf{v}_{\max}(\mathbf{S}(\boldsymbol{\sigma})^2) = \mathbf{v}_{\max}(\mathbf{S}(-\boldsymbol{\sigma})^2)$, since $\mathbf{S}(-\boldsymbol{\sigma})^2 = (-\sum_{i=1}^n \sigma_i \mathbf{x}_i \mathbf{x}_i^\top)^2 = \mathbf{S}(\boldsymbol{\sigma})^2$. So

$$I(\boldsymbol{\sigma}) + I(-\boldsymbol{\sigma}) = \sum_{i=1}^n \sigma_i' \left( -2(\mathbf{v}_{\max}(\mathbf{S}(\boldsymbol{\sigma}')^2)^\top \mathbf{x}_i)^2 \min\left\{ 1, \frac{\epsilon \|\mathbf{v}_{\max}(\mathbf{S}(\boldsymbol{\sigma}')^2)\|_1}{|\mathbf{v}_{\max}^\top(\mathbf{S}(\boldsymbol{\sigma})^2)\mathbf{x}_i|} \right\} \right)$$
$$+ \sum_{i=1}^n -\sigma_i \left( -2(\mathbf{v}_{\max}(\mathbf{S}(\boldsymbol{\sigma}')^2)^\top \mathbf{x}_i)^2 \min\left\{ 1, \frac{\epsilon \|\mathbf{v}_{\max}(\mathbf{S}(\boldsymbol{\sigma}')^2)\|_1}{|\mathbf{v}_{\max}(\mathbf{S}(\boldsymbol{\sigma}')^2)^\top \mathbf{x}_i|} \right\} \right)$$
$$= 0.$$

Similarly $J(\boldsymbol{\sigma}) + J(-\boldsymbol{\sigma}) = 0$.

According to Lemma 15, we can split $\{-1, +1\}^n$ into $\mathcal{A}^+$ and $\mathcal{A}^-$, such that $|\mathcal{A}^+| = |\mathcal{A}^-|$ and $\mathcal{A}^- = -\mathcal{A}^+$ where $-$ is element-wised negative sign. So we know:

$$\mathbb{E}_{\boldsymbol{\sigma}}[I(\boldsymbol{\sigma})] = \sum_{\boldsymbol{\sigma} \in \mathcal{A}^+} \frac{1}{2^n} I(\boldsymbol{\sigma}) + \sum_{\boldsymbol{\sigma} \in \mathcal{A}^-} \frac{1}{2^n} I(\boldsymbol{\sigma}) = \sum_{\boldsymbol{\sigma} \in \mathcal{A}^+} \frac{1}{2^n} I(\boldsymbol{\sigma}) + \sum_{\boldsymbol{\sigma} \in \mathcal{A}^+} \frac{1}{2^n} I(-\boldsymbol{\sigma}) = 0$$

$$\mathbb{E}_{\boldsymbol{\sigma}}[J(\boldsymbol{\sigma})] = \sum_{\boldsymbol{\sigma} \in \mathcal{A}^+} \frac{1}{2^n} J(\boldsymbol{\sigma}) + \sum_{\boldsymbol{\sigma} \in \mathcal{A}^-} \frac{1}{2^n} J(\boldsymbol{\sigma}) = \sum_{\boldsymbol{\sigma} \in \mathcal{A}^+} \frac{1}{2^n} J(\boldsymbol{\sigma}) + \sum_{\boldsymbol{\sigma} \in \mathcal{A}^+} \frac{1}{2^n} J(-\boldsymbol{\sigma}) = 0$$

Hence we conclude that $\hat{\mathfrak{R}}_{\mathcal{D}}(\tilde{f} \circ \mathcal{H} \Delta \mathcal{H}) \geq \hat{\mathfrak{R}}_{\mathcal{D}}(f \circ \mathcal{H} \Delta \mathcal{H})$.

**Case II:** $p > 2$.

Similarly we have that

$$\hat{\mathfrak{R}}_{\mathcal{D}}(\tilde{f} \circ \mathcal{H} \Delta \mathcal{H}) - \hat{\mathfrak{R}}_{\mathcal{D}}(f \circ \mathcal{H} \Delta \mathcal{H})$$
$$= \mathbb{E}_{\boldsymbol{\sigma}'} \left[ \sup_{\|\mathbf{w}\|_p \leq 2W} \frac{1}{n} \sum_{i=1}^n \sigma_i' \left( \mathbf{w}^\top \mathbf{x}_i - \mathbf{w}^\top \mathbf{x}_i \min\left\{ 1, \frac{\epsilon \|\mathbf{w}\|_1}{|\mathbf{w}^\top \mathbf{x}_i|} \right\} \right)^2 \right]$$
$$- d^{1-2/p} \mathbb{E}_{\boldsymbol{\sigma}} \left[ \sup_{\|\mathbf{w}\|_2 \leq W, \|\mathbf{w}'\|_2 \leq W} \frac{1}{n} \sum_{i=1}^n \sigma_i \mathbf{w}^\top \mathbf{x}_i \mathbf{x}_i^\top \mathbf{w}' \right]$$
$$\geq \mathbb{E}_{\boldsymbol{\sigma}'} \left[ \frac{1}{n} \sum_{i=1}^n \sigma_i' \left( \mathbf{v}'^{*\top} \mathbf{x}_i - \mathbf{v}'^{*\top} \mathbf{x}_i \min\left\{ 1, \frac{\epsilon \|\mathbf{v}'^*\|_1}{|\mathbf{v}'^{*\top} \mathbf{x}_i|} \right\} \right)^2 \right]$$
$$- d^{1-2/p} \mathbb{E}_{\boldsymbol{\sigma}} \left[ \frac{1}{n} \sum_{i=1}^n \sigma_i \mathbf{v}^{*\top} \mathbf{x}_i \mathbf{x}_i^\top \mathbf{v}^* \right]$$
$$= \frac{W^2}{n} (1 - d^{1-2/p}) \mathbb{E}_{\boldsymbol{\sigma}} \left\| \sum_{i=1}^n \sigma_i \mathbf{x}_i \mathbf{x}_i^\top \right\|$$
$$+ \frac{W^2}{n} \mathbb{E}_{\boldsymbol{\sigma}'} \left[ \sum_{i=1}^n \sigma_i' \left( -2(\mathbf{v}_{\max}(\mathbf{S}(\boldsymbol{\sigma}')^2)^\top \mathbf{x}_i)^2 \min\left\{ 1, \frac{\epsilon \|\mathbf{v}_{\max}(\mathbf{S}(\boldsymbol{\sigma}')^2)\|_1}{|\mathbf{v}_{\max}(\mathbf{S}(\boldsymbol{\sigma}')^2)^\top \mathbf{x}_i|} \right\} \right) \right]$$
$$+ \frac{W^2}{n} \mathbb{E}_{\boldsymbol{\sigma}'} \left[ \sum_{i=1}^n \sigma_i' \left( (\mathbf{v}_{\max}(\mathbf{S}(\boldsymbol{\sigma}')^2)^\top \mathbf{x}_i)^2 \min\left\{ 1, \frac{\epsilon \|\mathbf{v}_{\max}(\mathbf{S}(\boldsymbol{\sigma}')^2)\|_1}{|\mathbf{v}_{\max}(\mathbf{S}(\boldsymbol{\sigma}')^2)^\top \mathbf{x}_i|} \right\}^2 \right) \right]$$
$$\geq \frac{W^2}{n} (1 - d^{1-2/p}) \mathbb{E}_{\boldsymbol{\sigma}} \left\| \sum_{i=1}^n \sigma_i \mathbf{x}_i \mathbf{x}_i^\top \right\|_2.$$

where in the last step we also use the same reasoning as in **Case I**.

$\square$

# H PROOFS FOR LINEAR REGRESSION

## H.1 PROOF OF LEMMA 4

In this subsection we are going to present the proof of upper bound of the Rademacher complexity for regression under linear hypothesis.

*Proof.* We first aim at controlling the non-adversarial Rademacher complexity over the $\mathcal{H}\Delta\mathcal{H}$ class. We specify its definition given in (1) for the linear regression setting below

$$\hat{\mathfrak{R}}_{\mathcal{D}}(\ell \circ \mathcal{H}\Delta\mathcal{H}) = \mathbb{E}_\sigma \left[ \sup_{\mathbf{w},\mathbf{w}':\|\mathbf{w}\|_p \leq W, \|\mathbf{w}'\|_p \leq W} \frac{1}{n} \sum_{i=1}^n \sigma_i (\mathbf{w}^T \mathbf{x}_i - \mathbf{w}'^T \mathbf{x}_i)^2 \right] \ .$$

We introduce the variable change $\mathbf{v} := \mathbf{w} - \mathbf{w}'$, which yields to

$$\hat{\mathfrak{R}}_{\mathcal{D}}(\ell \circ \mathcal{H}\Delta\mathcal{H}) = \mathbb{E}_\sigma \left[ \sup_{\mathbf{v}:\|\mathbf{v}\|_p \leq 2W} \frac{1}{n} \sum_{i=1}^n \sigma_i (\mathbf{v}^\top \mathbf{x}_i)^2 \right] \ . \tag{36}$$

We first derive the upper bound of $\hat{\mathfrak{R}}_S(\ell \circ \mathcal{H}\Delta\mathcal{H})$. We follow similar steps than in Appendix G.1: we rewrite the supremum as a spectral norm and then apply a matrix Bernstein inequality. We have that

$$\begin{aligned}
\hat{\mathfrak{R}}_{\mathcal{D}}(\ell \circ \mathcal{H}\Delta\mathcal{H}) &= \mathbb{E}_\sigma \left[ \sup_{\mathbf{v}:\|\mathbf{v}\|_p \leq 2W} \frac{1}{n} \sum_{i=1}^n \sigma_i (\mathbf{v}^\top \mathbf{x}_i)^2 \right] \\
&= \mathbb{E}_\sigma \left[ \sup_{\mathbf{v}:\|\mathbf{v}\|_p \leq 2W} \mathbf{v}^\top \left( \frac{1}{n} \sum_{i=1}^n \sigma_i \mathbf{x}_i \mathbf{x}_i^\top \right) \mathbf{v} \right] \\
&\overset{(20)}{\leq} \mathbb{E}_\sigma \left[ \sup_{\mathbf{v}:\|\mathbf{v}\|_2 \leq 2W} \mathbf{v}^\top \left( \frac{1}{n} \sum_{i=1}^n \sigma_i \mathbf{x}_i \mathbf{x}_i^\top \right) \mathbf{v}' \right] \cdot \begin{cases} 1, & \text{if } 1 \leq p \leq 2 \\ d^{1-2/p}, & \text{else if } p > 2 \end{cases} \\
&= \frac{4W^2}{n} \mathbb{E}_\sigma \left[ \left\| \sum_{i=1}^n \sigma_i \mathbf{x}_i \mathbf{x}_i^\top \right\|_2 \right] \times \begin{cases} 1, & \text{if } 1 \leq p \leq 2 \\ d^{1-2/p}, & \text{else if } p > 2 \end{cases} \ .
\end{aligned}$$

Following exactly the same steps as in Appendix G.1, we apply the matrix Bernstein inequality. Let us denote by $\mathbf{Z}_i := \sigma_i \mathbf{x}_i \mathbf{x}_i^\top \in \mathbb{R}^{d \times d}$ the random matrices we want to apply Theorem 4 to. Then, $\mathbf{Z}_i^2 = \sigma_i^2 (\mathbf{x}_i \mathbf{x}_i^\top)^2 = (\mathbf{x}_i \mathbf{x}_i^\top)^2$ is a deterministic matrix. These random matrices $\mathbf{Z}_i$ are symmetric, independent, have zero mean and are such that for all $i \in [n]$

$$\|\mathbf{Z}_i\|_2 = \sqrt{\lambda_{\max}(\mathbf{Z}_i^2)} = \sqrt{\lambda_{\max}(\mathbf{x}_i \mathbf{x}_i^\top \mathbf{x}_i \mathbf{x}_i^\top)} = \|\mathbf{x}_i\|_2^2 \leq \max_{j \in [n]} \|\mathbf{x}_j\|_2^2 = \|\mathbf{X}\|_{2,\infty}^2 \ .$$

Moreover, let $\mathbf{Y} := \sum_{i=1}^n \mathbf{Z}_i$. According to matrix Bernstein inequality, we get the desired bound

$$\begin{aligned}
&\hat{\mathfrak{R}}_{\mathcal{D}}(\ell \circ \mathcal{H}\Delta\mathcal{H}) \\
&\overset{(14)}{\leq} \frac{4W^2}{n} \left( \sqrt{2 \left\| \sum_{i=1}^n (\mathbf{x}_i \mathbf{x}_i^\top)^2 \right\|_2 \log(2d)} + \frac{1}{3} \|\mathbf{X}\|_{2,\infty}^2 \log(2d) \right) \times \begin{cases} 1 & p \leq 2 \\ d^{1-2/p} & p > 2 \end{cases} \ .
\end{aligned}$$

$\square$

## H.2 PROOF OF THE UPPER BOUND OF THEOREM 2

In this subsection we will present the proof of upper bound of the adversarial Rademacher complexity for regression under linear hypothesis.

*Proof.* We then examine the adversarial Rademacher complexity of linear regression models defined in (3) as

$$\hat{\mathfrak{R}}_S(\tilde{\ell} \circ \mathcal{H}\Delta\mathcal{H}) = \mathbb{E}_\sigma \left[ \sup_{\substack{\mathbf{w}:\|\mathbf{w}\|_p \leq W \\ \mathbf{w}':\|\mathbf{w}'\|_p \leq W}} \frac{1}{n} \sum_{i=1}^n \sigma_i \max_{\boldsymbol{\delta}:\|\boldsymbol{\delta}\|_\infty \leq \epsilon} (\mathbf{w}^T(\mathbf{x}_i + \boldsymbol{\delta}) - \mathbf{w}'^T(\mathbf{x}_i + \boldsymbol{\delta}))^2 \right] \quad . \quad (37)$$

We start by expressing $\hat{\mathfrak{R}}_{\mathcal{D}}(\tilde{\ell} \circ \mathcal{H}\Delta\mathcal{H})$ as a function of $\hat{\mathfrak{R}}_{\mathcal{D}}(\ell \circ \mathcal{H}\Delta\mathcal{H})$, its non-adversarial counterpart studied in Lemma 4. Let $\mathbf{v} := \mathbf{w} - \mathbf{w}'$. We expend this quantity as follows

$$\hat{\mathfrak{R}}_S(\tilde{\ell} \circ \mathcal{H}\Delta\mathcal{H}) \stackrel{(37)}{=} \mathbb{E}_\sigma \left[ \sup_{\mathbf{v}:\|\mathbf{v}\|_p \leq 2W} \frac{1}{n} \sum_{i=1}^n \sigma_i \max_{\boldsymbol{\delta}:\|\boldsymbol{\delta}\|_\infty \leq \epsilon} (\mathbf{v}^T\mathbf{x}_i + \mathbf{v}^T\boldsymbol{\delta})^2 \right]$$

$$= \mathbb{E}_\sigma \left[ \sup_{\mathbf{v}:\|\mathbf{v}\|_p \leq 2W} \frac{1}{n} \sum_{i=1}^n \sigma_i \max_{\boldsymbol{\delta}:\|\boldsymbol{\delta}\|_\infty \leq \epsilon} (\mathbf{v}^T\mathbf{x}_i)^2 + 2\mathbf{v}^T\mathbf{x}_i\mathbf{v}^T\boldsymbol{\delta} + (\mathbf{v}^T\boldsymbol{\delta})^2 \right]$$

$$\leq \hat{\mathfrak{R}}_S(\ell \circ \mathcal{H}\Delta\mathcal{H}) + \mathbb{E}_\sigma \left[ \sup_{\mathbf{v}:\|\mathbf{v}\|_p \leq 2W} \frac{1}{n} \sum_{i=1}^n \sigma_i \max_{\boldsymbol{\delta}:\|\boldsymbol{\delta}\|_\infty \leq \epsilon} 2\mathbf{v}^T\mathbf{x}_i\boldsymbol{\delta}^T\mathbf{v} + (\mathbf{v}^T\boldsymbol{\delta})^2 \right]$$

$$= \hat{\mathfrak{R}}_S(\ell \circ \mathcal{H}\Delta\mathcal{H}) + \underbrace{\mathbb{E}_\sigma \left[ \sup_{\mathbf{v}:\|\mathbf{v}\|_p \leq 2W} \frac{1}{n} \sum_{i=1}^n \sigma_i \max_{\boldsymbol{\delta}:\|\boldsymbol{\delta}\|_\infty \leq \epsilon} \mathbf{v}^\top(2\mathbf{x}_i\boldsymbol{\delta}^\top + \boldsymbol{\delta}\boldsymbol{\delta}^\top)\mathbf{v} \right]}_{A} ,$$

$$(38)$$

where we used the subadditivity of the supremum in to make appear the non-adversarial Rademacher complexity over $\mathcal{H}\Delta\mathcal{H}$ class.

Now we examine the upper bound of the second term $A$ using the notion of covering recalled in Definition 6. Let $\mathcal{C}$ be a covering of the centered $\ell_p$ ball of radius $2W$, that we denote by $\mathcal{B}_p(2W)$, with $\ell_p$ balls of radius $\rho > 0$. Let us define

$$\zeta_i(\mathbf{v}) := \max_{\boldsymbol{\delta}:\|\boldsymbol{\delta}\|_\infty \leq \epsilon} \mathbf{v}^\top(2\mathbf{x}_i\boldsymbol{\delta}^\top + \boldsymbol{\delta}\boldsymbol{\delta}^\top)\mathbf{v} \quad . \quad (39)$$

Thus we can rewrite $A$ as

$$A \stackrel{(39)}{=} \mathbb{E}_\sigma \left[ \sup_{\mathbf{v}\in\mathcal{B}_p(2W)} \frac{1}{n} \sum_{i=1}^n \sigma_i\zeta_i(\mathbf{v}) \right]$$

$$= \mathbb{E}_\sigma \left[ \sup_{\substack{\mathbf{v}\in\mathcal{B}_p(2W) \\ \mathbf{v}_c\in\mathcal{C}:\|\mathbf{v}-\mathbf{v}_c\|_p\leq\rho}} \frac{1}{n} \sum_{i=1}^n \sigma_i\big(\zeta_i(\mathbf{v}_c) + \zeta_i(\mathbf{v}) - \zeta_i(\mathbf{v}_c)\big) \right]$$

$$\leq \mathbb{E}_\sigma \left[ \sup_{\tilde{\mathbf{v}}\in\mathcal{C}} \frac{1}{n} \sum_{i=1}^n \sigma_i\zeta_i(\tilde{\mathbf{v}}) \right] + \mathbb{E}_\sigma \left[ \sup_{\mathbf{v}\in\mathcal{B}_p(2W)} \frac{1}{n} \sum_{i=1}^n \sigma_i\big(\zeta_i(\mathbf{v}) - \zeta_i(\mathbf{v}_c)\big) \right]$$

$$\leq \underbrace{\mathbb{E}_\sigma \left[ \sup_{\tilde{\mathbf{v}}\in\mathcal{C}} \frac{1}{n} \sum_{i=1}^n \sigma_i\zeta_i(\tilde{\mathbf{v}}) \right]}_{(I)} + \underbrace{\sup_{\mathbf{v}\in\mathcal{B}_p(2W)} \frac{1}{n} \sum_{i=1}^n |\zeta_i(\mathbf{v}) - \zeta_i(\mathbf{v}_c)|}_{(II)} \quad . \quad (40)$$

where $\mathbf{v}_c$ is the closest element to $\mathbf{v}$ in $\mathcal{C}$ and where we used the subadditivity of the supremum.

**Bounding** $(I)$**:** We first need to bound the left-hand side term $(I)$. We introduce the vector

$$\boldsymbol{\zeta}(\tilde{\mathbf{v}}) := [\zeta_1(\tilde{\mathbf{v}}), \ldots, \zeta_n(\tilde{\mathbf{v}})]^\top \in \mathbb{R}^n \quad .$$

By Massart's lemma (Lemma 5.2 of Massart (2000)), we are able to control the first term $(I)$ in (29):

$$(I) = \mathbb{E}_\sigma \left[ \sup_{\tilde{\mathbf{v}}\in\mathcal{C}} \frac{1}{n} \sum_{i=1}^n \sigma_i\zeta_i(\tilde{\mathbf{v}}) \right] \leq \frac{K\sqrt{2\log|\mathcal{C}|}}{n} \quad , \quad (41)$$

with $K$ given by the largest $\ell_2$-norm of $\boldsymbol{\zeta}$ over the covering $\mathcal{C}$, that is

$$K^2 = \max_{\tilde{\mathbf{v}} \in \mathcal{C}} \|\boldsymbol{\zeta}\|_2^2 = \max_{\tilde{\mathbf{v}} \in \mathcal{C}} \sum_{i=1}^n \zeta_i(\tilde{\mathbf{v}})^2 \le \max_{\mathbf{v} \in \mathcal{B}_p(2W)} \sum_{i=1}^n \zeta_i(\mathbf{v})^2 \ . \tag{42}$$

Now we examine the upper bound of $K$. Note that $\zeta_i(\mathbf{v}) \ge 0$, then we can upper bound as follows

$$
\begin{aligned}
\zeta_i(\mathbf{v}) &= \max_{\boldsymbol{\delta}: \|\boldsymbol{\delta}\|_\infty \le \epsilon} \mathbf{v}^\top (2\mathbf{x}_i \boldsymbol{\delta}^\top + \boldsymbol{\delta}\boldsymbol{\delta}^\top) \mathbf{v} \\
&\le \|\mathbf{v}\|_2^2 \max_{\boldsymbol{\delta}: \|\boldsymbol{\delta}\|_\infty \le \epsilon} \left\| 2\mathbf{x}_i \boldsymbol{\delta}^\top + \boldsymbol{\delta}\boldsymbol{\delta}^\top \right\|_2 \\
&\le \|\mathbf{v}\|_2^2 \left( \max_{\boldsymbol{\delta}: \|\boldsymbol{\delta}\|_\infty \le \epsilon} \left\| 2\mathbf{x}_i \boldsymbol{\delta}^\top \right\|_2 + \max_{\boldsymbol{\delta}: \|\boldsymbol{\delta}\|_\infty \le \epsilon} \left\| \boldsymbol{\delta}\boldsymbol{\delta}^\top \right\|_2 \right) \\
&= \|\mathbf{v}\|_2^2 \max_{\boldsymbol{\delta}: \|\boldsymbol{\delta}\|_\infty \le \epsilon} \|\boldsymbol{\delta}\|_2 \left( 2\|\mathbf{x}_i\|_2 + \max_{\boldsymbol{\delta}: \|\boldsymbol{\delta}\|_\infty \le \epsilon} \|\boldsymbol{\delta}\|_2 \right) \\
&\stackrel{\|\boldsymbol{\delta}\|_2 \le \sqrt{d}\|\boldsymbol{\delta}\|_\infty}{\le} \|\mathbf{v}\|_2^2 \sqrt{d}\epsilon \left( \sqrt{d}\epsilon + 2\|\mathbf{x}_i\|_2 \right) \ ,
\end{aligned}
\tag{43}
$$

where we applied Cauchy-Schwarz inequality and the definition of operator norm and then the sub-additivity of the maximum. Recalling the case disjunction

$$
\begin{cases}
\|\mathbf{v}\|_2 \le \|\mathbf{v}\|_p, & \text{if } 1 \le p \le 2 \\
\|\mathbf{v}\|_2 \le \|\mathbf{v}\|_p \, d^{1/2-1/p}, & \text{else if } p > 2
\end{cases} \ , \tag{44}
$$

we are able to upper bound $K^2$, by using (42) and (43) which leads to

$$
\begin{aligned}
K^2 &\stackrel{(42)}{\le} \max_{\mathbf{v} \in \mathcal{B}_p(2W)} \sum_{i=1}^n \zeta_i(\mathbf{v})^2 \\
&\stackrel{(43)}{\le} \max_{\mathbf{v} \in \mathcal{B}_p(2W)} \|\mathbf{v}\|_2^4 (\sqrt{d}\epsilon)^2 \sum_{i=1}^n \left( \sqrt{d}\epsilon + 2\|\mathbf{x}_i\|_2 \right)^2 \\
&\stackrel{(44)}{\le} \max_{\mathbf{v} \in \mathcal{B}_p(2W)} \|\mathbf{v}\|_p^4 (\sqrt{d}\epsilon)^2 \sum_{i=1}^n \left( \sqrt{d}\epsilon + 2\|\mathbf{x}_i\|_2 \right)^2 \cdot \begin{cases} 1, & \text{if } 1 \le p \le 2 \\ (d^{1-2/p})^2, & \text{else if } p > 2 \end{cases} \\
&= (4W^2)^2 (\sqrt{d}\epsilon)^2 \sum_{i=1}^n \left( \sqrt{d}\epsilon + 2\|\mathbf{x}_i\|_2 \right)^2 \cdot \begin{cases} 1, & \text{if } 1 \le p \le 2 \\ (d^{1-2/p})^2, & \text{else if } p > 2 \end{cases} \ .
\end{aligned}
$$

Then, by taking the square root in the above we get

$$K \le 4W^2 \sqrt{d}\epsilon \sqrt{\sum_{i=1}^n \left( \sqrt{d}\epsilon + 2\|\mathbf{x}_i\|_2 \right)^2} \cdot \begin{cases} 1, & \text{if } 1 \le p \le 2 \\ d^{1-2/p}, & \text{else if } p > 2 \end{cases} \ .$$

Jointly with the application of Lemma 20, we can conclude that

$$(I) \le 4\frac{W^2}{n} \sqrt{d}\epsilon \sqrt{\sum_{i=1}^n \left( \sqrt{d}\epsilon + 2\|\mathbf{x}_i\|_2 \right)^2} \sqrt{2d\log(6W/\rho)} \times \begin{cases} 1, & \text{if } 1 \le p \le 2 \\ d^{1-2/p}, & \text{else if } p > 2 \end{cases} \ . \tag{45}$$

**Bounding** $(II)$**.** Now we turn to bounding the second term of (40)

$$(II) := \sup_{\mathbf{v} \in \mathcal{B}_p(2W)} \frac{1}{n} \sum_{i=1}^n |\zeta_i(\mathbf{v}) - \zeta_i(\mathbf{v}_c)| \ ,$$

recalling that $\zeta_i(\mathbf{v}) := \max_{\boldsymbol{\delta}: \|\boldsymbol{\delta}\|_\infty \le \epsilon} \mathbf{v}^\top (2\mathbf{x}_i \boldsymbol{\delta}^\top + \boldsymbol{\delta}\boldsymbol{\delta}^\top)\mathbf{v}$. Let $i \in [n]$, $\mathbf{v} \in \mathcal{B}_p(2W)$ and its corresponding closest point in the covering $\mathbf{v}_c \in \mathcal{C}$. Let us define

$$
\begin{cases}
\boldsymbol{\delta}^* := \arg\max_{\|\boldsymbol{\delta}\|_\infty \le \epsilon} \mathbf{v}^\top (2\mathbf{x}_i \boldsymbol{\delta}^\top + \boldsymbol{\delta}\boldsymbol{\delta}^\top)\mathbf{v} \\
\boldsymbol{\delta}_c^* := \arg\max_{\|\boldsymbol{\delta}\|_\infty \le \epsilon} \mathbf{v}_c^\top (2\mathbf{x}_i \boldsymbol{\delta}^\top + \boldsymbol{\delta}\boldsymbol{\delta}^\top)\mathbf{v}_c
\end{cases} \ .
$$

Also, let $\mathbf{M}^* := 2\mathbf{x}_i\boldsymbol{\delta}^{*\top} + \boldsymbol{\delta}^*\boldsymbol{\delta}^{*\top}$. Then, we can make the difference explicit and upper bound it

$$
\begin{aligned}
\zeta_i(\mathbf{v}) - \zeta_i(\mathbf{v}_c) &= \mathbf{v}^\top(2\mathbf{x}_i\boldsymbol{\delta}^{*\top} + \boldsymbol{\delta}^*\boldsymbol{\delta}^{*\top})\mathbf{v} - \mathbf{v}_c^\top(2\mathbf{x}_i\boldsymbol{\delta}_c^{*\top} + \boldsymbol{\delta}_c^*\boldsymbol{\delta}_c^{*\top})\mathbf{v}_c \\
&\leq \mathbf{v}^\top(2\mathbf{x}_i\boldsymbol{\delta}^{*\top} + \boldsymbol{\delta}^*\boldsymbol{\delta}^{*\top})\mathbf{v} - \mathbf{v}_c^\top(2\mathbf{x}_i\boldsymbol{\delta}^{*\top} + \boldsymbol{\delta}^*\boldsymbol{\delta}^{*\top})\mathbf{v}_c \\
&= \mathbf{v}^\top\mathbf{M}^*\mathbf{v} - \mathbf{v}_c^\top\mathbf{M}^*\mathbf{v}_c \\
&= (\mathbf{v} - \mathbf{v}_c)^\top\mathbf{M}^*\mathbf{v} + \mathbf{v}_c^\top\mathbf{M}^*(\mathbf{v} - \mathbf{v}_c) \\
&\overset{\text{Cauchy-Schwarz}}{\leq} \|\mathbf{M}^*\|_2\|\mathbf{v} - \mathbf{v}_c\|_2(\|\mathbf{v}\|_2 + \|\mathbf{v}_c\|_2) \\
&\leq 4\rho W\sqrt{d}\epsilon\left(\sqrt{d}\epsilon + 2\|\mathbf{x}_i\|_2\right) \times \begin{cases} 1, & \text{if } 1 \leq p \leq 2 \\ d^{1-2/p}, & \text{else if } p > 2 \end{cases},
\end{aligned}
$$

where lastly we used the same arguments leading to (43), the norm transfer in (44) and the inequality $\|.\|_2 \leq \sqrt{d}\|.\|_\infty$. Symmetrically, we can show that the above upper bound holds for $\zeta_i(\mathbf{v}_c) - \zeta_i(\mathbf{v})$. Thus, $(II)$ is upper bounded by

$$
(II) \leq 4\rho W\sqrt{d}\epsilon\left(\sqrt{d}\epsilon + \frac{2}{n}\sum_{i=1}^n\|\mathbf{x}_i\|_2\right) \times \begin{cases} 1, & \text{if } 1 \leq p \leq 2 \\ d^{1-2/p}, & \text{else if } p > 2 \end{cases}. \tag{46}
$$

Let us choose $\rho = W/n$. We have then proved that

$$
\begin{aligned}
A &\overset{(45)-(46)}{\leq} (I) + (II) \\
&\leq \left[\frac{1}{n}4W^2\sqrt{d}\epsilon\sqrt{\sum_{i=1}^n\left(\sqrt{d}\epsilon + 2\|\mathbf{x}_i\|_2\right)^2}\sqrt{2d\log(6W/\rho)} + 4\rho W\sqrt{d}\epsilon\left(\sqrt{d}\epsilon + \frac{2}{n}\sum_{i=1}^n\|\mathbf{x}_i\|_2\right)\right] \\
&\qquad\qquad\qquad\qquad\qquad\qquad\qquad\qquad\qquad\qquad\times \begin{cases} 1, & \text{if } 1 \leq p \leq 2 \\ d^{1-2/p}, & \text{else if } p > 2 \end{cases} \\
&= 4\frac{W^2}{n}\sqrt{d}\epsilon\left(\sqrt{d}\epsilon + \frac{2}{n}\sum_{i=1}^n\|\mathbf{x}_i\|_2 + \sqrt{\sum_{i=1}^n\left(\sqrt{d}\epsilon + 2\|\mathbf{x}_i\|_2\right)^2}\sqrt{2d\log(6n)}\right) \\
&\qquad\qquad\qquad\qquad\qquad\qquad\qquad\qquad\qquad\qquad\times \begin{cases} 1, & \text{if } 1 \leq p \leq 2 \\ d^{1-2/p}, & \text{else if } p > 2 \end{cases}.
\end{aligned}
$$

By combining (38), (40) and the above bound, we are able to finish the proof for the upper bound of the adversarial Rademacher complexity over class $\mathcal{H}\Delta\mathcal{H}$ for linear regression. Finally, this gives

$$
\begin{aligned}
&\hat{\mathfrak{R}}_S(\tilde{\ell}\circ\mathcal{H}\Delta\mathcal{H}) \leq \hat{\mathfrak{R}}_S(\ell\circ\mathcal{H}\Delta\mathcal{H}) \\
&+ 4\frac{W^2}{n}\sqrt{d}\epsilon\left(\sqrt{d}\epsilon + \frac{2}{n}\sum_{i=1}^n\|\mathbf{x}_i\|_2 + \sqrt{\sum_{i=1}^n\left(\sqrt{d}\epsilon + 2\|\mathbf{x}_i\|_2\right)^2}\sqrt{2d\log(6n)}\right) \\
&\qquad\qquad\qquad\qquad\qquad\qquad\qquad\qquad\qquad\qquad\times \begin{cases} 1, & \text{if } 1 \leq p \leq 2 \\ d^{1-2/p}, & \text{else if } p > 2 \end{cases}.
\end{aligned}
$$

$\qquad\square$

### H.3 PROOF OF THE LOWER BOUND OF THEOREM 2

In this subsection we present the proof of lower bound of the adversarial Rademacher complexity for regression under linear hypothesis.

*Proof.* Recall the definition of non-adversarial Rademacher complexity

$$\hat{\mathfrak{R}}_{\mathcal{D}}(\ell \circ \mathcal{H} \Delta \mathcal{H}) \overset{(36)}{=} \mathbb{E}\left[\sup_{\|\mathbf{v}\|_p \leq 2W} \frac{1}{n} \sum_{i=1}^{n} \sigma_i(\mathbf{v}^\top \mathbf{x}_i)^2\right]$$

$$\overset{Lemma\ 8}{\leq} \mathbb{E}\left[\sup_{\|\mathbf{v}\|_2 \leq 2W} \frac{1}{n} \sum_{i=1}^{n} \sigma_i(\mathbf{v}^\top \mathbf{x}_i)^2\right] \cdot \begin{cases} 1, & \text{if } 1 \leq p \leq 2 \\ d^{1-2/p}, & \text{else if } p > 2 \end{cases},$$

due to equivalence of norms. Now, we denote $\mathbf{v}^*$ such that $\mathbf{v}^* = \arg\max_{\|\mathbf{v}\|_2 \leq 2W} \frac{1}{n} \sum_{i=1}^{n} \sigma_i(\mathbf{v}^\top \mathbf{x}_i)^2$. One can verify that the maximum value of $\frac{1}{n} \sum_{i=1}^{n} \sigma_i(\mathbf{v}^\top \mathbf{x}_i)^2$ is:

$$\sup_{\|\mathbf{v}\|_2 \leq 2W} \frac{1}{n} \sum_{i=1}^{n} \sigma_i(\mathbf{v}^\top \mathbf{x}_i)^2 = \frac{1}{n} \sup_{\|\mathbf{v}\|_2 \leq 2W} \mathbf{v}^\top \left(\sum_{i=1}^{n} \sigma_i \mathbf{x}_i \mathbf{x}_i^\top\right) \mathbf{v} \leq \frac{4W^2}{n} \left\|\sum_{i=1}^{n} \sigma_i \mathbf{x}_i \mathbf{x}_i^\top\right\|_2$$

and if we define $\mathbf{S}(\boldsymbol{\sigma}) := \sum_{i=1}^{n} \sigma_i \mathbf{x}_i \mathbf{x}_i^\top$ the maxima is attained when $\mathbf{v}^* = 2W \mathbf{v}_{\max}(\mathbf{S}(\boldsymbol{\sigma})^2)$.

Now, we switch to adversarial Rademacher:

$$\hat{\mathfrak{R}}_{\mathcal{D}}(\tilde{\ell} \circ \mathcal{H} \Delta \mathcal{H}) = \mathbb{E}\left[\sup_{\|\mathbf{v}\|_p \leq 2W} \frac{1}{n} \sum_{i=1}^{n} \sigma_i \max_{\|\boldsymbol{\delta}\|_\infty \leq \epsilon} (\mathbf{v}^\top(\mathbf{x}_i + \boldsymbol{\delta}))^2\right]$$

$$\overset{Lemma\ 14}{\geq} \mathbb{E}\left[\sup_{\|\mathbf{v}\|_2 \leq 2W} \frac{1}{n} \sum_{i=1}^{n} \sigma_i \max_{\|\boldsymbol{\delta}\|_\infty \leq \epsilon} (\mathbf{v}^\top(\mathbf{x}_i + \boldsymbol{\delta}))^2\right] \cdot \begin{cases} d^{1-2/p}, & \text{if } 1 \leq p \leq 2 \\ 1, & \text{else if } p > 2 \end{cases}$$

According to Lemma 16, we have:

$$\hat{\mathfrak{R}}_{\mathcal{D}}(\tilde{\ell} \circ \mathcal{H} \Delta \mathcal{H}) = \mathbb{E}\left[\sup_{\|\mathbf{v}\|_2 \leq 2W} \frac{1}{n} \sum_{i=1}^{n} \sigma_i(\epsilon\|\mathbf{v}\|_1 + |\mathbf{v}^\top \mathbf{x}_i|)^2\right] \cdot \begin{cases} d^{1-2/p}, & \text{if } 1 \leq p \leq 2 \\ 1, & \text{else if } p > 2 \end{cases}.$$

**Case I:** $1 \leq p \leq 2$:

First, to avoid confusion in different Rademacher variables, let us use $\boldsymbol{\sigma}'$ and $\boldsymbol{\sigma}$ to denote the Rademacher variables in $\hat{\mathfrak{R}}_{\mathcal{D}}(\tilde{\ell} \circ \mathcal{H} \Delta \mathcal{H})$ and $\hat{\mathfrak{R}}_{\mathcal{D}}(\ell \circ \mathcal{H} \Delta \mathcal{H})$. Then, let us define $\mathbf{v}'^* := 2W \mathbf{v}_{\max}(\mathbf{S}(\boldsymbol{\sigma}')^2)$

$$\hat{\mathfrak{R}}_{\mathcal{D}}(\tilde{\ell} \circ \mathcal{H} \Delta \mathcal{H}) - \hat{\mathfrak{R}}_{\mathcal{D}}(\ell \circ \mathcal{H} \Delta \mathcal{H})$$

$$= \mathbb{E}_{\boldsymbol{\sigma}'}\left[\sup_{\|\mathbf{v}\|_p \leq 2W} \frac{1}{n} \sum_{i=1}^{n} \sigma_i'(\epsilon\|\mathbf{v}\|_1 + |\mathbf{v}^\top \mathbf{x}_i|)^2\right] - \mathbb{E}_{\boldsymbol{\sigma}}\left[\sup_{\|\mathbf{v}\|_2 \leq 2W} \frac{1}{n} \sum_{i=1}^{n} \sigma_i(\mathbf{v}^\top \mathbf{x}_i)^2\right]$$

$$\geq \mathbb{E}_{\boldsymbol{\sigma}'}\left[\frac{1}{n} \sum_{i=1}^{n} \sigma_i'(\epsilon\|\mathbf{v}'^*\|_1 + |\mathbf{v}'^{*\top} \mathbf{x}_i|)^2\right] - \mathbb{E}_{\boldsymbol{\sigma}}\left[\frac{1}{n} \sum_{i=1}^{n} \sigma_i(\mathbf{v}^{*\top} \mathbf{x}_i)^2\right]$$

$$= \frac{4W^2}{n} \mathbb{E}_{\boldsymbol{\sigma}'}\left[\sum_{i=1}^{n} \sigma_i' \left(2\epsilon\|\mathbf{v}_{\max}(\mathbf{S}(\boldsymbol{\sigma}')^2)\|_1 |\mathbf{v}_{\max}^\top(\mathbf{S}(\boldsymbol{\sigma}')^2)\mathbf{x}_i| + \epsilon^2\|\mathbf{v}_{\max}(\mathbf{S}(\boldsymbol{\sigma}')^2)\|_1^2\right)\right]$$

Let $J(\boldsymbol{\sigma}) = \sum_{i=1}^{n} \sigma_i \left(2\epsilon\|\mathbf{v}_{\max}(\mathbf{S}(\boldsymbol{\sigma})^2)\|_1 |\mathbf{v}_{\max}^\top(\mathbf{S}(\boldsymbol{\sigma})^2)\mathbf{x}_i| + \epsilon^2\|\mathbf{v}_{\max}(\mathbf{S}(\boldsymbol{\sigma})^2)\|_1^2\right)$, and we claim that $J(\boldsymbol{\sigma}) + J(-\boldsymbol{\sigma}) = 0$. Now we are going to prove this claim. First we know that $\mathbf{v}_{\max}(\mathbf{S}(\boldsymbol{\sigma})^2) = \mathbf{v}_{\max}(\mathbf{S}(-\boldsymbol{\sigma})^2)$, since $\mathbf{S}(-\boldsymbol{\sigma})^2 = (-\sum_{i=1}^{n} \sigma_i \mathbf{x}_i \mathbf{x}_i^\top)^2 = \mathbf{S}(\boldsymbol{\sigma})^2$. So

$$J(\boldsymbol{\sigma}) + J(-\boldsymbol{\sigma}) = \sum_{i=1}^{n} \sigma_i \left(2\epsilon\|\mathbf{v}_{\max}(\mathbf{S}(\boldsymbol{\sigma})^2)\|_1 |\mathbf{v}_{\max}^\top(\mathbf{S}(\boldsymbol{\sigma})^2)\mathbf{x}_i| + \epsilon^2\|\mathbf{v}_{\max}(\mathbf{S}(\boldsymbol{\sigma})^2)\|_1^2\right)$$

$$+ \sum_{i=1}^{n} -\sigma_i \left(2\epsilon\|\mathbf{v}_{\max}(\mathbf{S}(\boldsymbol{\sigma})^2)\|_1 |\mathbf{v}_{\max}^\top(\mathbf{S}(\boldsymbol{\sigma})^2)\mathbf{x}_i| + \epsilon^2\|\mathbf{v}_{\max}(\mathbf{S}(\boldsymbol{\sigma})^2)\|_1^2\right)$$

$$= 0.$$

According to Lemma 15, we can split $\{-1, +1\}^n$ into $\mathcal{A}^+$ and $\mathcal{A}^-$, such that $|\mathcal{A}^+| = |\mathcal{A}^-|$ and $\mathcal{A}^- = -\mathcal{A}^+$ where $-$ is element-wised negative sign. So we know:

$$\mathbb{E}_{\boldsymbol{\sigma}}[J(\boldsymbol{\sigma})] = \sum_{\boldsymbol{\sigma} \in \mathcal{A}^+} \frac{1}{2^n} J(\boldsymbol{\sigma}) + \sum_{\boldsymbol{\sigma} \in \mathcal{A}^-} \frac{1}{2^n} J(\boldsymbol{\sigma}) = \sum_{\boldsymbol{\sigma} \in \mathcal{A}^+} \frac{1}{2^n} J(\boldsymbol{\sigma}) + \sum_{\boldsymbol{\sigma} \in \mathcal{A}^+} \frac{1}{2^n} J(-\boldsymbol{\sigma}) = 0$$

Hence we conclude that $\hat{\mathfrak{R}}_\mathcal{D}(\tilde{\ell} \circ \mathcal{H}\Delta\mathcal{H}) \geq \hat{\mathfrak{R}}_\mathcal{D}(\ell \circ \mathcal{H}\Delta\mathcal{H})$.

**Case II:** $p > 2$:

Similarly we have:

$$\hat{\mathfrak{R}}_\mathcal{D}(\tilde{\ell} \circ \mathcal{H}\Delta\mathcal{H}) - \hat{\mathfrak{R}}_\mathcal{D}(\ell \circ \mathcal{H}\Delta\mathcal{H})$$
$$= \mathbb{E}_{\boldsymbol{\sigma}'} \left[ \sup_{\|\mathbf{v}\|_p \leq 2W} \frac{1}{n} \sum_{i=1}^n \sigma_i'(\epsilon\|\mathbf{v}\|_1 + |\mathbf{v}^\top \mathbf{x}_i|)^2 \right] - d^{1-2/p} \mathbb{E}_{\boldsymbol{\sigma}} \left[ \sup_{\|\mathbf{v}\|_2 \leq 2W} \frac{1}{n} \sum_{i=1}^n \sigma_i(\mathbf{v}^\top \mathbf{x}_i)^2 \right]$$
$$\geq \mathbb{E}_{\boldsymbol{\sigma}'} \left[ \frac{1}{n} \sum_{i=1}^n \sigma_i'(\epsilon\|\mathbf{v}'^*\|_1 + |\mathbf{v}'^{*\top} \mathbf{x}_i|)^2 \right] - d^{1-2/p} \mathbb{E}_{\boldsymbol{\sigma}} \left[ \frac{1}{n} \sum_{i=1}^n \sigma_i(\mathbf{v}^{*\top} \mathbf{x}_i)^2 \right]$$
$$= (1 - d^{1-2/p}) \frac{4W^2}{n} \mathbb{E} \left\| \sum_{i=1}^n \sigma_i \mathbf{x}_i \mathbf{x}_i^\top \right\|_2$$
$$+ \frac{4W^2}{n} \mathbb{E}_{\boldsymbol{\sigma}'} \left[ 2 \sum_{i=1}^n \sigma_i' \left( \epsilon \|\mathbf{v}_{\max}(\mathbf{S}(\boldsymbol{\sigma}')^2)\|_1 |\mathbf{v}_{\max}^\top(\mathbf{S}(\boldsymbol{\sigma}')^2) \mathbf{x}_i| + \epsilon^2 \|\mathbf{v}_{\max}(\mathbf{S}(\boldsymbol{\sigma}')^2)\|_1^2 \right) \right]$$
$$\geq (1 - d^{1-2/p}) \frac{4W^2}{n} \mathbb{E} \left\| \sum_{i=1}^n \sigma_i \mathbf{x}_i \mathbf{x}_i^\top \right\|_2,$$

where in the last step we also use the same reasoning as in **Case I**.

$\square$

# I    PROOF OF NEURAL NETWORK COMPLEXITY BOUNDS

In this section, we will present the proof of upper bound of the adversarial Rademacher complexity under two-layer neural network hypothesis (Theorem 3). We provide the proof of classification bound in Appendix I.1, and regression in Appendix I.2.

## I.1    PROOF OF BINARY CLASSIFICATION SETTING

*Proof of the classification bound of Theorem 3.* We recall that $\mathcal{B}_p(R)$ stands for the $\ell_p$ ball of radius $R$ (either in vector $\mathbb{R}^d$ or matrix $\mathbb{R}^{d \times d}$ space). To simplify notations, we denote the coordinate-wise ReLU activation function by

$$g(\mathbf{x}) := \max\{0, \mathbf{x}\}, \mathbb{R}^d \to \mathbb{R}^d.$$

where $\max$ is applied coordinate-wisely. Using the definition of the $\tilde{f} \circ \mathcal{H}\Delta\mathcal{H}$ class in (7), we upper bound the adversarial Rademacher complexity in the binary classification setting by expressing it as its non-adversarial counterpart plus an additional term. Thus we get

$$\hat{\mathfrak{R}}_\mathcal{D}(\tilde{f} \circ \mathcal{H}\Delta\mathcal{H}) = \mathbb{E}_{\boldsymbol{\sigma}} \left[ \sup_{\substack{\mathbf{a}, \mathbf{a}' \in \mathcal{B}_p(A)^2 \\ \mathbf{W}, \mathbf{W}' \in \mathcal{B}_p(W)^2}} \frac{1}{n} \sum_{i=1}^n \sigma_i \min_{\|\boldsymbol{\delta}\|_\infty \leq \epsilon} \mathbf{a}^\top g(\mathbf{W}(\mathbf{x}_i + \boldsymbol{\delta})) \mathbf{a}'^\top g(\mathbf{W}'(\mathbf{x}_i + \boldsymbol{\delta})) \right].$$

As the function $\boldsymbol{\delta} \mapsto \mathbf{a}^\top g(\mathbf{W}(\mathbf{x}_i + \boldsymbol{\delta})) \mathbf{a}'^\top g(\mathbf{W}'(\mathbf{x}_i + \boldsymbol{\delta}))$ is continuous as a composition of continuous function (linear and ReLU), then it reaches a minimum of the compact $\ell_\infty$ ball of radius $\epsilon > 0$, also denoted by $\mathcal{B}_\infty(\epsilon)$. Let $\boldsymbol{\delta}_i^*$ an argument of the minima of the latter function,

*i.e.* $\boldsymbol{\delta}_i^* \in \arg\min_{\boldsymbol{\delta}\in\mathcal{B}_\infty(\epsilon)} \mathbf{a}^\top g(\mathbf{W}(\mathbf{x}_i+\boldsymbol{\delta}))\mathbf{a'}^\top g(\mathbf{W}'(\mathbf{x}_i+\boldsymbol{\delta}))$. With this notation, we can write

$$\hat{\mathfrak{R}}_{\mathcal{D}}(\tilde{f}\circ\mathcal{H}\Delta\mathcal{H}) \le \mathfrak{R}_{\mathcal{D}}(f\circ\mathcal{H}\Delta\mathcal{H})$$

$$+\mathbb{E}_\sigma\left[\sup_{\substack{\mathbf{a},\mathbf{a}'\in\mathcal{B}_p(A)^2 \\ \mathbf{W},\mathbf{W}'\in\mathcal{B}_p(W)^2}} \frac{1}{n}\sum_{i=1}^n \sigma_i \mathbf{a}^\top g(\mathbf{W}(\mathbf{x}_i+\boldsymbol{\delta}_i^*))g(\mathbf{W}'(\mathbf{x}_i+\boldsymbol{\delta}_i^*))^\top\mathbf{a}' - \mathbf{a}g(\mathbf{W}\mathbf{x}_i)g(\mathbf{W}'\mathbf{x}_i)^\top\mathbf{a}'\right]$$

$$=\mathfrak{R}_{\mathcal{D}}(f\circ\mathcal{H}\Delta\mathcal{H})$$

$$+\mathbb{E}_\sigma\left[\sup_{\substack{\mathbf{a},\mathbf{a}'\in\mathcal{B}_p(A)^2 \\ \mathbf{W},\mathbf{W}'\in\mathcal{B}_p(W)^2}} \mathbf{a}^\top\left(\frac{1}{n}\sum_{i=1}^n \sigma_i\left(g(\mathbf{W}(\mathbf{x}_i+\boldsymbol{\delta}_i^*))g(\mathbf{W}'(\mathbf{x}_i+\boldsymbol{\delta}_i^*))^\top - g(\mathbf{W}\mathbf{x}_i)g(\mathbf{W}'\mathbf{x}_i)^\top\right)\right)\mathbf{a}'\right],$$

Let the term inside the expectation being maximized (as a continuous function over a compact set) at $\mathbf{a}^*, \mathbf{a}'^*, \mathbf{W}^*, \mathbf{W}'^*$. And let

$$\mathbf{S}_i := \sigma_i\left(g(\mathbf{W}^*(\mathbf{x}_i+\boldsymbol{\delta}_i^*))g(\mathbf{W}'^*(\mathbf{x}_i+\boldsymbol{\delta}_i^*))^\top - g(\mathbf{W}^*\mathbf{x}_i)g(\mathbf{W}'^*\mathbf{x}_i)^\top\right) \in \mathbb{R}^{d\times d} \ .$$

Then, we can rewrite the term in the expectation as

$$\mathbf{a}^{*\top}\left(\frac{1}{n}\sum_{i=1}^n \mathbf{S}_i\right)\mathbf{a}'^* \overset{\substack{\text{Cauchy-Schwarz ineq.}}}{\le} \|\mathbf{a}^*\|_2 \|\mathbf{a}'^*\|_2 \left\|\frac{1}{n}\sum_{i=1}^n \mathbf{S}_i\right\|_2$$

$$\overset{\substack{Lemma\ 8}}{\le} \|\mathbf{a}^*\|_p \|\mathbf{a}'^*\|_p \left\|\frac{1}{n}\sum_{i=1}^n \mathbf{S}_i\right\|_2 \times \begin{cases} 1, & \text{if } 1\le p\le 2 \\ d^{1-2/p}, & \text{else if } p>2 \end{cases}$$

$$\overset{\substack{\mathbf{a}^*,\mathbf{a}'^*\in\mathcal{B}_p(A)^2}}{\le} A^2 \left\|\frac{1}{n}\sum_{i=1}^n \mathbf{S}_i\right\|_2 \times \begin{cases} 1, & \text{if } 1\le p\le 2 \\ d^{1-2/p}, & \text{else if } p>2 \end{cases} \ .$$

Thus we have the following inequality

$$\hat{\mathfrak{R}}_{\mathcal{D}}(\tilde{f}\circ\mathcal{H}\Delta\mathcal{H}) \le \mathfrak{R}_{\mathcal{D}}(f\circ\mathcal{H}\Delta\mathcal{H}) + \frac{A^2}{n}\mathbb{E}_\sigma\left[\left\|\sum_{i=1}^n \mathbf{S}_i\right\|_2\right] \times \begin{cases} 1, & \text{if } 1\le p\le 2 \\ d^{1-2/p}, & \text{else if } p>2 \end{cases} \ . \quad (47)$$

Let us now estimate the spectral norm of the average of random matrices $\sum_{i=1}^n \mathbf{S}_i$ using the matrix Bernstein inequality of Theorem 4. We have that

$$\|\mathbf{S}_i\|_2 = \left\|g(\mathbf{W}^*(\mathbf{x}_i+\boldsymbol{\delta}_i^*))g(\mathbf{W}'^*(\mathbf{x}_i+\boldsymbol{\delta}_i^*))^\top - g(\mathbf{W}^*\mathbf{x}_i)g(\mathbf{W}'^*\mathbf{x}_i)^\top\right\|_2$$

$$= \|g(\mathbf{W}^*(\mathbf{x}_i+\boldsymbol{\delta}_i^*))(g(\mathbf{W}'^*(\mathbf{x}_i+\boldsymbol{\delta}_i^*)) - g(\mathbf{W}'^*\mathbf{x}_i))^\top$$

$$+ (g(\mathbf{W}^*(\mathbf{x}_i+\boldsymbol{\delta}_i^*)) - g(\mathbf{W}^*\mathbf{x}_i))g(\mathbf{W}'^*\mathbf{x}_i)^\top\|_2$$

$$\le \left\|g(\mathbf{W}^*(\mathbf{x}_i+\boldsymbol{\delta}_i^*))(g(\mathbf{W}'^*(\mathbf{x}_i+\boldsymbol{\delta}_i^*)) - g(\mathbf{W}'^*\mathbf{x}_i))^\top\right\|_2$$

$$+ \left\|(g(\mathbf{W}^*(\mathbf{x}_i+\boldsymbol{\delta}_i^*)) - g(\mathbf{W}^*\mathbf{x}_i))g(\mathbf{W}'^*\mathbf{x}_i)^\top\right\|_2$$

$$\overset{\substack{\text{Cauchy-Schwarz}}}{\le} \|\mathbf{W}^*(\mathbf{x}_i+\boldsymbol{\delta}_i^*)\|_2 \|\mathbf{W}'^*\boldsymbol{\delta}_i^*\|_2 + \|\mathbf{W}^*\boldsymbol{\delta}_i^*\|_2 \|\mathbf{W}'^*\mathbf{x}_i\|_2$$

$$\overset{\substack{Lemma\ 8}}{\le} W^2\sqrt{d}\epsilon(\sqrt{d}\epsilon + 2\|\mathbf{x}_i\|_2) \times \begin{cases} 1, & \text{if } 1\le p\le 2 \\ d^{1-2/p}, & \text{else if } p>2 \end{cases} \quad (48)$$

$$\le W^2\sqrt{d}\epsilon(\sqrt{d}\epsilon + 2\|\mathbf{X}\|_{2,\infty}) \times \begin{cases} 1, & \text{if } 1\le p\le 2 \\ d^{1-2/p}, & \text{else if } p>2 \end{cases} \ , \quad (49)$$

where we used the 1-Lipschitzness of the ReLU function. Now let us examine the upper bound of the variance of the sum. We notice that $\mathbf{S}_i^\top \mathbf{S}_i$ is a deterministic matrix as $\sigma_i^2 = 1$. Thus, we have

$$
\mathrm{Var}\left(\sum_{i=1}^n \mathbf{S}_i\right) := \left\|\sum_{i=1}^n \mathbb{E}_\sigma[\mathbf{S}_i^\top \mathbf{S}_i]\right\|_2
$$

$$
\leq \sum_{i=1}^n \|\mathbf{S}_i\|_2^2
$$

$$
\overset{(48)}{\leq} \left(W^2\sqrt{d}\epsilon\right)^2 \left(\sum_{i=1}^n (\sqrt{d}\epsilon + 2\|\mathbf{x}_i\|_2)^2\right) \times \begin{cases} 1, & \text{if } 1 \leq p \leq 2 \\ (d^{1-2/p})^2, & \text{else if } p > 2 \end{cases}.
$$

Using (49), we can apply matrix Bernstein inequality of Theorem 4 we get

$$
\mathbb{E}_\sigma\left[\left\|\sum_{i=1}^n \mathbf{S}_i\right\|_2\right] \leq \left(W^2\epsilon\sqrt{d}\sqrt{2\left(\sum_{i=1}^n (\sqrt{d}\epsilon + 2\|\mathbf{x}_i\|_2)^2\right)\log(2d)}\right.
$$

$$
\left. + \frac{1}{3}W^2\epsilon\sqrt{d}(\sqrt{d}\epsilon + 2\|\mathbf{X}\|_{2,\infty})\log(2d)\right) \times \begin{cases} 1, & \text{if } 1 \leq p \leq 2 \\ d^{1-2/p}, & \text{else if } p > 2 \end{cases}
$$

$$
= W^2\epsilon\sqrt{d\log(2d)}\left(\sqrt{2\sum_{i=1}^n (\sqrt{d}\epsilon + 2\|\mathbf{x}_i\|_2)^2} + \frac{1}{3}\sqrt{\log(2d)}(\sqrt{d}\epsilon + 2\|\mathbf{X}\|_{2,\infty})\right)
$$

$$
\times \begin{cases} 1, & \text{if } 1 \leq p \leq 2 \\ d^{1-2/p}, & \text{else if } p > 2 \end{cases}.
$$

The above combined with (47) concludes the proof

$$
\hat{\mathfrak{R}}_{\mathcal{D}}(\tilde{f} \circ \mathcal{H}\Delta\mathcal{H}) \leq \mathfrak{R}_{\mathcal{D}}(f \circ \mathcal{H}\Delta\mathcal{H})
$$

$$
+ \frac{A^2}{n}W^2\epsilon\sqrt{d\log(2d)}\left(\sqrt{2\sum_{i=1}^n (\sqrt{d}\epsilon + 2\|\mathbf{x}_i\|_2)^2} + \frac{1}{3}\sqrt{\log(2d)}(\sqrt{d}\epsilon + 2\|\mathbf{X}\|_{2,\infty})\right)
$$

$$
\times \begin{cases} 1, & \text{if } 1 \leq p \leq 2 \\ d^{2-4/p}, & \text{else if } p > 2 \end{cases}.
$$

$\square$

## I.2 PROOF OF REGRESSION SETTING

*Proof of the regression bound of Theorem 3.* First, by the definition of $\mathfrak{R}_{\mathcal{D}}(\tilde{\ell} \circ \mathcal{H}\Delta\mathcal{H})$ we have that

$$
\mathfrak{R}_{\mathcal{D}}(\tilde{\ell} \circ \mathcal{H}\Delta\mathcal{H}) = \mathbb{E}_\sigma\left[\sup_{\substack{\mathbf{a},\mathbf{a}'\in\mathcal{B}_p(A)^2 \\ \mathbf{W},\mathbf{W}'\in\mathcal{B}_p(W)^2}} \frac{1}{n}\sum_{i=1}^n \sigma_i \max_{\|\boldsymbol{\delta}\|_\infty \leq \epsilon} \left(\mathbf{a}^\top g(\mathbf{W}(\mathbf{x}_i+\boldsymbol{\delta})) - \mathbf{a}'g(\mathbf{W}'(\mathbf{x}_i+\boldsymbol{\delta}))\right)^2\right]
$$

$$
\leq \mathfrak{R}_{\mathcal{D}}(\ell \circ \mathcal{H}\Delta\mathcal{H})
$$

$$
+ \mathbb{E}_\sigma\left[\sup_{\substack{\mathbf{a},\mathbf{a}'\in\mathcal{B}_p(A)^2 \\ \mathbf{W},\mathbf{W}'\in\mathcal{B}_p(W)^2}} \frac{1}{n}\sum_{i=1}^n \sigma_i \left(\mathbf{a}^\top g(\mathbf{W}(\mathbf{x}_i+\boldsymbol{\delta}_i^*)) - \mathbf{a}'^\top g(\mathbf{W}'(\mathbf{x}_i+\boldsymbol{\delta}_i^*))\right)^2\right.
$$

$$
\left. - \left(\mathbf{a}^\top g(\mathbf{W}\mathbf{x}_i) - \mathbf{a}'^\top g(\mathbf{W}'\mathbf{x}_i)\right)^2\right],
$$

where, like in Appendix I.1, we denote by $\boldsymbol{\delta}_i^*$ an argument of the maxima of the inner function, *i.e.* $\boldsymbol{\delta}_i^* \in \arg\max_{\boldsymbol{\delta}\in\mathcal{B}_\infty(\epsilon)} \left(\mathbf{a}^\top g(\mathbf{W}(\mathbf{x}_i+\boldsymbol{\delta})) - \mathbf{a}'g(\mathbf{W}'(\mathbf{x}_i+\boldsymbol{\delta}))\right)^2$. Then, by introducing following matrices

- $\mathbf{P}_i := g(\mathbf{W}(\mathbf{x}_i + \boldsymbol{\delta}_i^*))g(\mathbf{W}(\mathbf{x}_i + \boldsymbol{\delta}_i^*))^\top - g(\mathbf{W}\mathbf{x}_i)g(\mathbf{W}\mathbf{x}_i)^\top \in \mathbb{R}^{d\times d}$

- $\mathbf{Q}_i := g(\mathbf{W}'(\mathbf{x}_i + \boldsymbol{\delta}_i^*))g(\mathbf{W}'(\mathbf{x}_i + \boldsymbol{\delta}_i^*))^\top - g(\mathbf{W}'\mathbf{x}_i)g(\mathbf{W}'\mathbf{x}_i)^\top \in \mathbb{R}^{d\times d}$

- $\mathbf{R}_i := g(\mathbf{W}(\mathbf{x}_i + \boldsymbol{\delta}_i^*))g(\mathbf{W}'(\mathbf{x}_i + \boldsymbol{\delta}_i^*))^\top - g(\mathbf{W}\mathbf{x}_i)g(\mathbf{W}'\mathbf{x}_i)^\top \in \mathbb{R}^{d\times d}$

we can rewrite the upper bound of the adversarial Rademacher complexity as

$$\mathfrak{R}_\mathcal{D}(\tilde{\ell}\circ\mathcal{H}\Delta\mathcal{H}) \le \mathfrak{R}_\mathcal{D}(\ell\circ\mathcal{H}\Delta\mathcal{H}) + \mathbb{E}_\sigma\left[\sup_{\substack{\mathbf{a},\mathbf{a}'\in\mathcal{B}_p(A)^2 \\ \mathbf{W},\mathbf{W}'\in\mathcal{B}_p(W)^2}} \frac{1}{n}\sum_{i=1}^n \sigma_i(\mathbf{a}^\top\mathbf{P}_i\mathbf{a} + \mathbf{a}'^\top\mathbf{Q}_i\mathbf{a}' + \mathbf{a}^\top\mathbf{R}_i\mathbf{a}')\right]$$

$$\le \mathfrak{R}_\mathcal{D}(\ell\circ\mathcal{H}\Delta\mathcal{H}) + \mathbb{E}_\sigma\left[\sup_{\substack{\mathbf{a}\in\mathcal{B}_p(A) \\ \mathbf{W},\mathbf{W}'\in\mathcal{B}_p(W)^2}} \frac{1}{n}\sum_{i=1}^n \sigma_i\mathbf{a}^\top\mathbf{P}_i\mathbf{a}\right]$$

$$+ \mathbb{E}_\sigma\left[\sup_{\substack{\mathbf{a}'\in\mathcal{B}_p(A) \\ \mathbf{W},\mathbf{W}'\in\mathcal{B}_p(W)^2}} \frac{1}{n}\sum_{i=1}^n \sigma_i\mathbf{a}'^\top\mathbf{Q}_i\mathbf{a}'\right] + \mathbb{E}_\sigma\left[\sup_{\substack{\mathbf{a},\mathbf{a}'\in\mathcal{B}_p(A)^2 \\ \mathbf{W},\mathbf{W}'\in\mathcal{B}_p(W)^2}} \frac{1}{n}\sum_{i=1}^n \sigma_i\mathbf{a}^\top\mathbf{R}_i\mathbf{a}'\right]$$

$$\le \mathfrak{R}_\mathcal{D}(\ell\circ\mathcal{H}\Delta\mathcal{H}) + \frac{1}{n}\mathbb{E}_\sigma\left[\sup_{\substack{\mathbf{a}\in\mathcal{B}_p(A) \\ \mathbf{W},\mathbf{W}'\in\mathcal{B}_p(W)^2}} \mathbf{a}^\top\left(\sum_{i=1}^n \sigma_i\mathbf{P}_i\right)\mathbf{a}\right]$$

$$+ \frac{1}{n}\mathbb{E}_\sigma\left[\sup_{\substack{\mathbf{a}'\in\mathcal{B}_p(A) \\ \mathbf{W},\mathbf{W}'\in\mathcal{B}_p(W)^2}} \mathbf{a}'^\top\left(\sum_{i=1}^n \sigma_i\mathbf{Q}_i\right)\mathbf{a}'\right] + \frac{1}{n}\mathbb{E}_\sigma\left[\sup_{\substack{\mathbf{a},\mathbf{a}'\in\mathcal{B}_p(A)^2 \\ \mathbf{W},\mathbf{W}'\in\mathcal{B}_p(W)^2}} \mathbf{a}^\top\left(\sum_{i=1}^n \sigma_i\mathbf{R}_i\right)\mathbf{a}'\right].$$

Following the same steps that lead to (47), we get that

$$\mathfrak{R}_\mathcal{D}(\tilde{\ell}\circ\mathcal{H}\Delta\mathcal{H}) \le \mathfrak{R}_\mathcal{D}(\ell\circ\mathcal{H}\Delta\mathcal{H})$$
$$+ \frac{A^2}{n}\underbrace{\left(\mathbb{E}_\sigma\left\|\sum_{i=1}^n \sigma_i\mathbf{P}_i^*\right\| + \mathbb{E}_\sigma\left\|\sum_{i=1}^n \sigma_i\mathbf{Q}_i^*\right\| + \mathbb{E}_\sigma\left\|\sum_{i=1}^n \sigma_i\mathbf{R}_i^*\right\|\right)}_{(I)} \times \begin{cases} 1, & \text{if } 1\le p\le 2 \\ d^{1-2/p}, & \text{else if } p>2 \end{cases},$$

where the matrices are defined by

- $\mathbf{P}_i^* := g(\mathbf{W}_\mathbf{P}^*(\mathbf{x}_i + \boldsymbol{\delta}_i^*))g(\mathbf{W}_\mathbf{P}^*(\mathbf{x}_i + \boldsymbol{\delta}_i^*))^\top - g(\mathbf{W}_\mathbf{P}^*\mathbf{x}_i)g(\mathbf{W}_\mathbf{P}^*\mathbf{x}_i)^\top \in \mathbb{R}^{d\times d}$

- $\mathbf{Q}_i^* := g(\mathbf{W}_\mathbf{Q}'^*(\mathbf{x}_i + \boldsymbol{\delta}_i^*))g(\mathbf{W}_\mathbf{Q}'^*(\mathbf{x}_i + \boldsymbol{\delta}_i^*))^\top - g(\mathbf{W}_\mathbf{Q}'^*\mathbf{x}_i)g(\mathbf{W}_\mathbf{Q}'^*\mathbf{x}_i)^\top \in \mathbb{R}^{d\times d}$

- $\mathbf{R}_i^* := g(\mathbf{W}_\mathbf{R}^*(\mathbf{x}_i + \boldsymbol{\delta}_i^*))g(\mathbf{W}_\mathbf{R}'^*(\mathbf{x}_i + \boldsymbol{\delta}_i^*))^\top - g(\mathbf{W}_\mathbf{R}^*\mathbf{x}_i)g(\mathbf{W}_\mathbf{R}'^*\mathbf{x}_i)^\top \in \mathbb{R}^{d\times d}$

for some optimal matrices $\mathbf{W}_\mathbf{P}^*, \mathbf{W}_\mathbf{P}'^*, \mathbf{W}_\mathbf{Q}^*, \mathbf{W}_\mathbf{Q}'^*, \mathbf{W}_\mathbf{R}^*, \mathbf{W}_\mathbf{R}'^*$ with $\ell_p$-norm smaller than $W$. Now examine the spectral norm of $\sigma_i\mathbf{P}_i^*, \sigma_i\mathbf{Q}_i^*$ and $\sigma_i\mathbf{R}_i^*$ by following the same steps leading to (48):

$$\|\sigma_i\mathbf{P}_i^*\|_2 = \left\|g(\mathbf{W}_\mathbf{P}^*(\mathbf{x}_i + \boldsymbol{\delta}_i^*))g(\mathbf{W}_\mathbf{P}^*(\mathbf{x}_i + \boldsymbol{\delta}_i^*))^\top - g(\mathbf{W}_\mathbf{P}^*\mathbf{x}_i)g(\mathbf{W}_\mathbf{P}^*\mathbf{x}_i)^\top\right\|_2$$

$$\le W^2\epsilon\sqrt{d}(\sqrt{d}\epsilon + 2\|\mathbf{x}_i\|_2) \times \begin{cases} 1, & \text{if } 1\le p\le 2 \\ d^{1-2/p}, & \text{else if } p>2 \end{cases}, \qquad (50)$$

$$\|\sigma_i\mathbf{Q}_i^*\|_2 = \left\|g(\mathbf{W}_\mathbf{Q}'^*(\mathbf{x}_i + \boldsymbol{\delta}_i^*))g(\mathbf{W}_\mathbf{Q}'^*(\mathbf{x}_i + \boldsymbol{\delta}_i^*))^\top - g(\mathbf{W}_\mathbf{Q}'^*\mathbf{x}_i)g(\mathbf{W}_\mathbf{Q}'^*\mathbf{x}_i)^\top\right\|_2$$

$$\le W^2\epsilon\sqrt{d}(\sqrt{d}\epsilon + 2\|\mathbf{x}_i\|_2) \times \begin{cases} 1, & \text{if } 1\le p\le 2 \\ d^{1-2/p}, & \text{else if } p>2 \end{cases},$$

$$\|\sigma_i\mathbf{R}_i^*\|_2 = \left\|g(\mathbf{W}_\mathbf{R}^*(\mathbf{x}_i + \boldsymbol{\delta}_i^*))g(\mathbf{W}_\mathbf{R}'^*(\mathbf{x}_i + \boldsymbol{\delta}_i^*))^\top - g(\mathbf{W}_\mathbf{R}^*\mathbf{x}_i)g(\mathbf{W}_\mathbf{R}'^*\mathbf{x}_i)^\top\right\|_2$$

$$\le W^2\epsilon\sqrt{d}(\sqrt{d}\epsilon + 2\|\mathbf{x}_i\|_2) \times \begin{cases} 1, & \text{if } 1\le p\le 2 \\ d^{1-2/p}, & \text{else if } p>2 \end{cases}.$$

Now we examine the variance of $\sum_{i=1}^n \sigma_i \mathbf{P}_i^*$, $\sum_{i=1}^n \sigma_i \mathbf{Q}_i^*$ and $\sum_{i=1}^n \sigma_i \mathbf{R}_i^*$:

$$
\begin{aligned}
\operatorname{Var}\left(\sum_{i=1}^n \sigma_i \mathbf{P}_i^*\right) &= \left\|\sum_{i=1}^n \mathbb{E}_\sigma[\mathbf{P}_i^{*\top}\mathbf{P}_i^*]\right\|_2 \\
&\leq \sum_{i=1}^n \left\|g(\mathbf{W}_\mathbf{P}^*(\mathbf{x}_i + \boldsymbol{\delta}_i^*))g(\mathbf{W}_\mathbf{P}^*(\mathbf{x}_i + \boldsymbol{\delta}_i^*))^\top - g(\mathbf{W}_\mathbf{P}^*\mathbf{x}_i)g(\mathbf{W}_\mathbf{P}^*\mathbf{x}_i)^\top\right\|_2^2 \\
&\overset{(50)}{\leq} \sum_{i=1}^n \left(W^2\epsilon\sqrt{d}(\sqrt{d}\epsilon + 2\|\mathbf{x}_i\|_2)\right)^2 \times \begin{cases} 1, & \text{if } 1 \leq p \leq 2 \\ d^{2-4/p}, & \text{else if } p > 2 \end{cases} \\
&= \left(W^2\epsilon\sqrt{d}\right)^2 \sum_{i=1}^n \left(\sqrt{d}\epsilon + 2\|\mathbf{x}_i\|_2\right)^2 \times \begin{cases} 1, & \text{if } 1 \leq p \leq 2 \\ d^{2-4/p}, & \text{else if } p > 2 \end{cases}.
\end{aligned}
$$

Similarly, we can show that $\operatorname{Var}\left(\sum_{i=1}^n \sigma_i \mathbf{Q}_i^*\right)$ and $\operatorname{Var}\left(\sum_{i=1}^n \sigma_i \mathbf{R}_i^*\right)$ can be upper bounded the same way. Now, by applying three times the matrix Bernstein inequality (Theorem 4) we have:

$$
(I) \leq W^2\epsilon\sqrt{d\log(2d)}\left(3\sqrt{2\sum_{i=1}^n(\sqrt{d}\epsilon + 2\|\mathbf{x}_i\|_2)^2} + \sqrt{\log(2d)}(\sqrt{d}\epsilon + 2\|\mathbf{X}\|_{2,\infty})\right)
$$

$$
\times \begin{cases} 1, & \text{if } 1 \leq p \leq 2 \\ d^{1-2/p}, & \text{else if } p > 2 \end{cases}.
$$

which concludes the proof. $\qquad \square$

## J  PROOF OF LEMMA 5

In this section we present the proof of Lemma 5.

*Proof.* The proof idea is to show that, by perturbing the standard risk on $\mathcal{T}$ within the adversary set, the perturbed risk can approximate the standard risk on $\mathcal{T}'$ with some error. First, let us define a perturbed risk for any perturbation

$$
\mathcal{R}_\mathcal{T}(h_\mathbf{w}, y, \boldsymbol{\delta}) := \mathbb{E}_{\mathbf{x}\sim\mathcal{T}}\left[\ell(h_\mathbf{w}(\mathbf{x} + \boldsymbol{\delta}), y(\mathbf{x})\right] \tag{51}
$$

$$
= \sum_{\mathbf{x}\in\mathcal{X}} \mathbf{p}(\mathbf{x})\frac{1}{2}|\operatorname{sign}(\mathbf{w}(\mathbf{x} + \boldsymbol{\delta}(\mathbf{x})) - y(\mathbf{x})|
$$

$$
= \mathbf{p}^\top \tilde{\boldsymbol{\ell}}\left(\{\boldsymbol{\delta}_i\}_{i=1}^{|\mathcal{X}|}\right). \tag{52}
$$

We then recall the definition of the standard risk on $\mathcal{T}'$

$$
\mathcal{R}_{\mathcal{T}'}(h_\mathbf{w}, y) = \sum_{\mathbf{x}\in\mathcal{X}} \mathbf{p}'(\mathbf{x})\frac{1}{2}|\operatorname{sign}(\mathbf{w}^\top\mathbf{x}) - y(\mathbf{x})|
$$

$$
= \mathbf{p}'^\top\boldsymbol{\ell}. \tag{53}
$$

We recall the definition of adversarially robust risk over domain $\mathcal{T}$ for the labeling function $y(\cdot)$

$$
\mathcal{R}_\mathcal{T}^{adv}(h_\mathbf{w}, y) = \mathbb{E}_{\mathbf{x}\sim\mathcal{T}}\left[\max_{\|\boldsymbol{\delta}\|_\infty \leq \epsilon} \ell(h_\mathbf{w}(\mathbf{w} + \boldsymbol{\delta}), y(\mathbf{x}))\right]. \tag{54}
$$

So, for any $\boldsymbol{\delta}$ such that $\|\boldsymbol{\delta}\|_\infty \leq \epsilon$, we get that

$$
\mathcal{R}_{\mathcal{T}'}(h_\mathbf{w}, y) - \mathcal{R}_\mathcal{T}^{adv}(h_\mathbf{w}, y) \overset{(51)-(54)}{\leq} \mathcal{R}_{\mathcal{T}'}(h_\mathbf{w}, y) - \mathcal{R}_\mathcal{T}(h_\mathbf{w}, y, \boldsymbol{\delta}) \overset{(52)-(53)}{=} \mathbf{p}'^\top\boldsymbol{\ell} - \mathbf{p}^\top\tilde{\boldsymbol{\ell}}(\{\boldsymbol{\delta}_i\}_{i=1}^{|\mathcal{X}|}).
$$

Since the above inequality holds for any $\boldsymbol{\delta}$ such that $\|\boldsymbol{\delta}\|_\infty \leq \epsilon$, we must have

$$
\mathcal{R}_{\mathcal{T}'}(h_\mathbf{w}, y) - \mathcal{R}_\mathcal{T}^{adv}(h_\mathbf{w}, y) \leq \min_{\|\boldsymbol{\delta}_i\|_\infty \leq \epsilon} |\mathbf{p}'^\top\boldsymbol{\ell} - \mathbf{p}^\top\tilde{\boldsymbol{\ell}}(\{\boldsymbol{\delta}_i\}_{i=1}^{|\mathcal{X}|})|.
$$

Now, let's examine the coordinates in $\Lambda$. For $i$-th coordinate, if it is in $\Lambda$, we know that

$$-\epsilon\|\mathbf{w}\|_1 \leq \mathbf{w}^\top \mathbf{x}_i \leq \epsilon\|\mathbf{w}\|_1 \ ,$$

which implies that, there is a $\boldsymbol{\delta}$ that can change the sign of $\mathrm{sign}(\mathbf{w}(\mathbf{x} + \boldsymbol{\delta}))$, and hence change the value of $\tilde{\ell}_i$. That is, if $\mathbf{w}^\top \mathbf{x} y(\mathbf{x}) \geq 0$, there is a $\boldsymbol{\delta}^* = \arg\min_{\|\boldsymbol{\delta}\|_\infty \leq \epsilon} \mathbf{w}^\top \boldsymbol{\delta} y(\mathbf{x})$, such that

$$\mathbf{w}^\top (\mathbf{x} + \boldsymbol{\delta}^*) y(\mathbf{x}) = \mathbf{w}^\top \mathbf{x} y(\mathbf{x}) - \epsilon\|\mathbf{w}\|_1 \leq 0 \ .$$

Similarly, if $\mathbf{w}^\top \mathbf{x} y(\mathbf{x}) \leq 0$, there is a $\boldsymbol{\delta}^* = \arg\max_{\|\boldsymbol{\delta}\|_\infty \leq \epsilon} \mathbf{w}^\top \boldsymbol{\delta} y(\mathbf{x})$, such that:

$$\mathbf{w}^\top \mathbf{x} y(\mathbf{x}) + \mathbf{w}^\top \boldsymbol{\delta}^* y(\mathbf{x}) = \mathbf{w}^\top \mathbf{x} y(\mathbf{x}) + \epsilon\|\mathbf{w}\|_1 \geq 0 \ .$$

Finally, we get

$$\min_{\|\boldsymbol{\delta}_i\|_\infty \leq \epsilon} |\mathbf{p}'^\top \boldsymbol{\ell} - \mathbf{p}^\top \tilde{\boldsymbol{\ell}}(\{\boldsymbol{\delta}_i\}_{i=1}^{|\mathcal{X}|})|$$

$$= \begin{cases} \min_{\tilde{\boldsymbol{\ell}} \in \{0,1\}^N} & \left|\mathbf{p}^\top \tilde{\boldsymbol{\ell}} - \mathbf{p}'^\top \boldsymbol{\ell}\right| \\ \mathrm{s.t.} & \tilde{\ell}_i = \ell_i, \ \forall \, i \in [N] \setminus \Lambda \end{cases}$$

$$= V^*(\mathbf{p}', \mathbf{p}, \boldsymbol{\ell}, \Lambda) \ .$$

**Corollary 3.** *Continuing with the settings and assumptions in Lemma 5, the following statement holds with probability at least $1 - c$:*

$$\mathcal{R}_{\mathcal{T}'}(h_{\mathbf{w}}, y) \leq \mathcal{R}_{\mathcal{S}}^{adv-label}(h_{\mathbf{w}}, y) + \mathcal{R}_{\mathcal{S}}^{adv-label}(h_{\mathbf{w}_{\mathcal{S}}^*}, y) + V^*(\mathbf{p}', \mathbf{p}, \boldsymbol{\ell}, \Lambda_\epsilon)$$

$$+ disc_{\mathcal{H}\Delta\mathcal{H}}^{adv}(\hat{\mathcal{S}}, \hat{\mathcal{T}}) + \mathcal{R}_{\mathcal{T}}^{adv}(h_{\mathbf{w}_{\mathcal{T}}^*}, h_{\mathbf{w}_{\mathcal{S}}^*}) + \mathcal{R}_{\mathcal{T}}^{adv}(h_{\mathbf{w}_{\mathcal{T}}^*}, y)$$

$$+ \hat{\mathfrak{R}}_{\mathcal{S}}(\tilde{\ell} \circ \mathcal{H}\Delta\mathcal{H}) + \hat{\mathfrak{R}}_{\mathcal{T}}(\tilde{\ell} \circ \mathcal{H}\Delta\mathcal{H}) + 3M\left(\sqrt{\frac{\log(1/c)}{n_{\mathcal{S}}}} + \sqrt{\frac{\log(1/c)}{n_{\mathcal{T}}}}\right). \quad (55)$$

Combining Lemmas 2 and 5 imply Corollary 3. Equation (55) shows that, small robust risk on source domain will imply small standard risk on any target domain $\mathcal{T}'$, if the residual error $V^*(\mathbf{p}', \mathbf{p}, \boldsymbol{\ell}, \Lambda_\epsilon)$ is also small. The residual error mainly depends on the adversarial budget. If the adversarial budget $\epsilon$ is larger, $\Lambda_\epsilon$ will be larger, which means the quantity $V^*(\mathbf{p}', \mathbf{p}, \boldsymbol{\ell}, \Lambda_\epsilon)$ will be smaller. However, there is a trade-off since $\epsilon$ will also affect other terms in the right hand side of the bound, such as adversarial Rademacher and the adversarial disagreement between $h_{\mathbf{w}_{\mathcal{T}}^*}$ and $h_{\mathbf{w}_{\mathcal{S}}^*}$. $\square$

## K    DETAILS ON EXPERIMENTS

In Table 2, we present the details of the convolutional network. For the convolutional layer (Conv2D or Conv1D), the first argument is the number channel. For a fully connected layer (FC), we list the number of hidden units as the first argument.

Table 2: Convolutional network architecture.

| Layer | Details |
|---|---|
| **feature extractor** | |
| conv1 | Conv2D(64, kernel size=5, stride=1, padding=2) |
| bn1 | BN2D, RELU, MaxPool2D(kernel size=2, stride=2) |
| conv2 | Conv2D(64, kernel size=5, stride=1, padding=2) |
| bn2 | BN2D, ReLU, MaxPool2D(kernel size=2, stride=2) |
| conv3 | Conv2D(128, kernel size=5, stride=1, padding=2) |
| bn3 | BN2D, ReLU |
| **classifier** | |
| fc1 | FC(2048) |
| bn4 | BN1D, ReLU |
| fc2 | FC(512) |
| bn5 | BN1D, ReLU |
| fc3 | FC(10) |

