# OpenReview forum: "Rademacher Complexity Over $\mathcal{H} \Delta \mathcal{H}$ Class for Adversarially Robust Domain Adaptation"
_ICLR.cc/2023/Conference — Submitted to ICLR 2023_

### Official Review · Reviewer_K4Lj · 2022-10-24

**Confidence:** 5
**Correctness:** 3
**Technical Novelty And Significance:** 2
**Empirical Novelty And Significance:** Not applicable
**Recommendation:** 3

**Clarity, Quality, Novelty And Reproducibility:**

Clarity: The motivation and writing are clear.

Quality and novelty : The conclusion of this paper is somewhat intuitive. In fact, there are many related works propose the similar relationship between Rademacher complexity in standard scenario and in adversarial scenario. The authors just replace the hypothesis class here. I do think the work of this paper is meaningful, but not novel enough.

**Strength And Weaknesses:**

## Pros:
1. The writing is clear and easy to follow. The paper is well-organized.
2. The experiments partly validate the theoretical results.
## Cons:
I have two main concerns about the paper:
1. About the definition of adversarially robust risk. In this paper, $$\mathcal{R} _\mathcal{D}^{adv}(h _w,h _{w'})=\mathbb{E} _{x\sim \mathcal{D}}[max _\delta \ell(h _w(x+\delta),h _{w'}(x+\delta))].$$ However, the classical definition [A][B] of adversarial error is $$\mathcal{R}
_\mathcal{D}^{adv}(h _w,h _{w'})=\mathbb{E} _{x\sim \mathcal{D}}[max _\delta \ell(h _w(x+\delta),h _{w'}(x))].$$
The definition of adversarial error is different with the classical one. Could the authors give some explainations about this? Furthermore, it seems that the authors use the classical definition to train their network in the experiments (see eq. (13)). Does it contradict with the theory?

2. About the reason why adversarial training gets better domain generalization than standard training. The authors explain this by an error bound which shows that the target clean error can be bounded by the source adversarial error. In my opinion, the bound can indeed explain that adversarial training in source domain can robustify the model on target domain. But, **there is no direct evidence** that adversarial training entails better standard accuracy on target domain, **compared to the normally trained model.**

Some typos in the paper:
1. In Lemma 2 and Corollary 1, the hypothesis is denoted by $w$ and $h_w$, are these two notations denote the same hypothesis?

[A]: Theoretically Principled Trade-off between Robustness and Accuracy. ICML 2019.

[B]: Towards Deep Learning Models Resistant to Adversarial Attacks. ICLR 2018.


**Summary Of The Paper:**

In this paper, the authors try to unravel why domain generalization is much more difficult in the adversarial scenario. Since the generalization bound is with respect to the adversarial Rademacher complexity term, they propose to analyze this key measure and show that adversarial Rademacher complexity is always greater than the non-adversarial one. They also try to explain the impact of adversarial training over generalization by a bound with respect to clean target error and adversarial source error. The experiments partly validate the theoretical results.

**Summary Of The Review:**

I believe the paper is well written and I appreciated the theoretical bounds about generalization error. However there are some parts of the paper that are confusing. Therefore, I believe the paper could be significantly improved by clarifying the points above.

---

> ### Author Response · Authors · 2022-11-13
> **Response to Reviewer K4Lj (1/2)**
>
> Thank you for your valuable comments, and we are sorry about any confusion due to our notational abuse. We will try to address your concerns here:
>
> # Definition of robust risk
> Thank you for catching this important point and we are sorry about the abuse of the notation. As stated in our response to Reviewer RkC2, we clarify our naming of these risks to be more precise. We refered to [GKKW21], and followed their naming convention, we will rename the first risk constant-in-the-ball adversarial risk and write it as $\mathcal{R}^{label-adv}\_\mathcal{D}(h\_\mathbf{w},y\_\mathcal{D}) := \mathbb{E}\_{\mathbf{x}\sim \mathcal{D}} \left[\max_{\|\boldsymbol{\delta}\|\_\infty\leq \epsilon}\ell (h\_\mathbf{w}(\mathbf{x}+\boldsymbol{\delta}), y\_\mathcal{D}(\mathbf{x}))\right]$.
> The second (exact-in-the-ball) adversarially robust risk will still be referred to as $\mathcal{R}^{adv}\_{\mathcal{D}}(h\_\mathbf{w},h\_{\mathbf{w}'}) := \mathbb{E}\_{\mathbf{x}\sim \mathcal{D}}  [\max_{\|\boldsymbol{\delta}\|\_\infty\leq \epsilon}\ell (h\_\mathbf{w}(\mathbf{x}+\boldsymbol{\delta}), h\_{\mathbf{w}'}(\mathbf{x}+\boldsymbol{\delta})) ]$.
> We will adapt this change in the revised version of article. However, we believe this confusion about the naming of these two object $\textbf{does not have any impact}$ on our theory. For example, in proving Lemma 2, if we adapt corrected notations we have:
> $$
> \begin{align}
>     \mathcal{R}^{label-adv}\_\mathcal{T}(h\_\mathbf{w},y\_\mathcal{T}) &= \mathbb{E}\_{\mathbf{x}\sim \mathcal{T}} \left[\max\_{\\|\boldsymbol{\delta}\\|\_\infty\leq \epsilon} \ell (h\_\mathbf{w}(\mathbf{x}+\boldsymbol{\delta}), y\_\mathcal{T} (\mathbf{x}))\right] \\\\
>     &\leq  \mathbb{E}\_{\mathbf{x}\sim \mathcal{T}} \left[\max_{\\|\boldsymbol{\delta}\\|\_\infty\leq \epsilon} \ell (h\_\mathbf{w}(\mathbf{x}+\boldsymbol{\delta}), h\_\mathbf{w\_{\mathcal{T}}^*}(\mathbf{x}+\boldsymbol{\delta})) + \ell (h\_\mathbf{w\_{\mathcal{T}}^*}(\mathbf{x}+\boldsymbol{\delta}), y\_\mathcal{T} (\mathbf{x}))\right] \\\\
>     &\leq \mathbb{E}\_{\mathbf{x}\sim \mathcal{T}} \left[\max\_{\\|\boldsymbol{\delta}\\|\_\infty\leq \epsilon} \ell (h\_\mathbf{w}(\mathbf{x}+\boldsymbol{\delta}),  h\_\mathbf{w\_{\mathcal{T}}^*}(\mathbf{x}+\boldsymbol{\delta}))\right] +
>      \mathbb{E}\_{\mathbf{x}\sim \mathcal{T}} \left[\max\_{\\|\boldsymbol{\delta}\\|\_\infty\leq \epsilon} \ell (h\_\mathbf{w\_{\mathcal{T}}^*}(\mathbf{x}+\boldsymbol{\delta}), y\_\mathcal{T} (\mathbf{x}))\right] \\\\
>     &= \mathcal{R}^{adv}\_\mathcal{T} ( h\_\mathbf{w} , h\_\mathbf{w\_{\mathcal{T}}^*} ) + \mathcal{R}^{label-adv}\_\mathcal{T} ( h\_\mathbf{w\_{\mathcal{T}}^*} , y\_\mathcal{T} ) \\\\
>    & \leq \mathcal{R}^{adv}\_\mathcal{T} ( h\_\mathbf{w} , h\_\mathbf{w\_{\mathcal{S}}^*}) + \mathcal{R}^{adv}\_\mathcal{T} ( h\_\mathbf{w\_{\mathcal{S}}^*} , h\_\mathbf{w\_{\mathcal{T}}^*} ) + \mathcal{R}^{label-adv}\_\mathcal{T} ( h\_\mathbf{w\_{\mathcal{T}}^*}, y\_\mathcal{T} )\\\\
>   &  \leq \mathcal{R}^{adv}\_\mathcal{S} ( h\_\mathbf{w} , h\_\mathbf{w\_{\mathcal{S}}^*}) + disc^{adv}\_{\mathcal{H}\Delta\mathcal{H}}(\mathcal{S},\mathcal{T}) + \mathcal{R}^{adv}\_\mathcal{T} (h\_\mathbf{w\_{\mathcal{S}}^*}, h\_\mathbf{w\_{\mathcal{T}}^*} ) + \mathcal{R}^{label-adv}\_\mathcal{T} (h\_\mathbf{w\_{\mathcal{T}}^*}, y\_\mathcal{T} )
> \end{align}
> $$
> where the risk decomposition still holds. We will definitely update the notations in the revised article, and thank you again for catching this mistake!
>
> # Technical Novelty
> Taking classification setting for example, Yin et al consider the Rademacher complexity over the loss class between model predictions and labels, i.e., $\frak{R} = \mathbb{E}[\sup_{\mathbf{w}}\frac{1}{n}\sum_{i=1}^n \sigma_i \min_{\|\boldsymbol{\delta}\|\leq \infty} \mathbf{w}^\top (\mathbf{x}+\boldsymbol{\delta})]$, where the inner minimization problem is $\textbf{linear}$ in $\mathbf{w}$ and $\boldsymbol{\delta}$. We consider the loss between predictions among two models, hence the Rademacher complexity is $\frak{R} = \mathbb{E}[\sup_{\mathbf{w},\mathbf{w}'}\frac{1}{n}\sum_{i=1}^n \sigma_i \min_{\|\boldsymbol{\delta}\|\leq \infty} \mathbf{w}^\top (\mathbf{x}+\boldsymbol{\delta})\mathbf{w}'^\top (\mathbf{x}+\boldsymbol{\delta})]$, where the inner problem is quadratic in terms of $\mathbf{w}$ and $\boldsymbol{\delta}$. Hence, unlike previous works on single domain Rademacher complexity, where the inner problem has simple closed form solution, we have to use $\varepsilon$-nets and covering number idea to estimate upper bound. We would also like to emphasize that another key contribution of this submission is the lower bound proof, where we derive the (complicated) closed form solution to inner quadratic programming, and leverage the symmetric property of Rademacher random variables to avoid heavy computation

---

> > ### Author Response · Authors · 2022-11-13
> > **Response to Reviewer K4Lj (2/2)**
> >
> >
> > # Adversarial training and transfer learning
> > The phenomena that adversarially trained model can sometimes perform better in standard domain adaptation is widely observed (including our experiments). In Lemma 5, what we showed is that small robust risk can guarantee small worst case generalization on different domain, with some residual error controlled by adversarial budget. This reveals how robustness captures domain shift, which is a gift from robust training and such result does not hold for standard ERM training. However, to show that robust model is strictly better than ERM model is beyond the scope of this paper, which we leave as an interesting future work.

---

> ### Author Response · Authors · 2022-11-18
> **Rebuttal Revision Submitted**
>
> Dear Reviewer K4Lj
>
> Thank you for your accurate and constructive comments, and we have included the solutions to your concerns in the revised version (significant changes are highlighted in blue). In particular, we corrected the definitions for two kinds of robust risks, and fixed the Lemma 2, the key generalization Lemma, which we believe will address your concerns.
>
> Also, after our correcting Lemma 2, there should be no gap between bound and experiments. In experiments, we minimize $\mathcal{R}_\\mathcal{S}^{adv-label}(h,y_\mathcal{S})$, which is the term that appears in RHS of Lemma 2.
>
> If you have more questions, please feel free to make a comment, and we are more than happy to hear from you soon.
>
> Thank you,
> The authors of submission 1294

---

> ### Author Response · Authors · 2022-11-27
> **A Kindly Reminder**
>
> Dear Reviewer K4Lj,
>
> Thank you again for your valuable comments, and we sincerely anticipate your valuable feedback. We believe our response will resolve your main concern about robust risk definitions, and we also adapt your suggestions on improving the paper quality. The authors really hope you can go through them and let us know if they addressed your concerns.
>
> If you have any further questions, please feel free to ask, and we are more than happy to explain/discuss with you.
>
> Thank you,
> The authors of paper 1294

---

> ### Author Response · Authors · 2022-12-03
> **A friendly reminder that the discussion stage 2 will be closed in 10 days**
>
> Dear Reviewer K4Lj,
>
> Thank you again for your accurate comments. Since the end of discussion stage 2 is approaching, we are eager to hear from you about whether our rebuttal addresses your concerns. If you have any other concerns, please let us know so that we can provide further explanation before the discussion period is ended.
>
> Thanks,
> The authors of paper 1294

---

> ### Author Response · Authors · 2022-12-08
> **A friendly reminder that the discussion stage will be closed in 4 days**
>
> Dear Reviewer K4Lj,
>
> Thank you again for your professional comments. Since the discussion stage will end in 4 days, we kindly ask you to respond to our rebuttal so that we can have time to address your further concern/questions.
>
> Thanks, The authors of submission 1294

---

> ### Author Response · Authors · 2022-12-10
> **A friendly reminder that the discussion stage will be closed in 2 days**
>
> Dear Reviewer K4Lj,
>
> Thanks again for your valuable review. We found that our rebuttal did resolve Reviewer RkC2's concern for robust risk definition, so we hope our rebuttal also works for you on the same question. If your concern remains, please do not hesitate to let us know, so that we can have enough time to provide further explanations.
>
> In addition, even though our problem setup and analysis techniques are totally different from the previous works, which we explained in our rebuttal, we are still more than happy to provide more detailed explanations on our novelty if you still have concern on that.
>
> There is no rush but the discussion stage will be closed in 2 days, so it would be great if you can let us know your feedback in advance so that we can compile response in time.
>
> Thank you,
> The authors of paper 1294

---

### Official Review · Reviewer_Du9x · 2022-10-25

**Confidence:** 4
**Correctness:** 4
**Technical Novelty And Significance:** 3
**Empirical Novelty And Significance:** Not applicable
**Recommendation:** 8

**Clarity, Quality, Novelty And Reproducibility:**

The paper is relatively clear and of sufficient quality. The exact story of Yin. et. al. (2018) dampens the novelty of this work however I concede that the proofs certainly require extra consideration.


**Strength And Weaknesses:**

Strength
- Extension of the same flavor of results from standard generalization -> robust generalization -> robust domain adaptation.
- Dimension-dependent lower bounds for uniform convergence in robust domain adaptation task.
- Link between standard target risk and adversarial source risk.



**Summary Of The Paper:**

This paper attempts to provide generalization bounds for adversarially robust domain adaptation.

The seminal work of Ben-David (2010) established the H-Delta-H divergence as a complexity measure for this task in the binary classification setting. Mansour (2010) established the discrepancy based distance as a complexity measure for domain adaption for multi-class classification with general loss functions. The discrepancy-based distance is further estimated from finite samples via empirical Rademacher complexity of the disagreement hypothesis class H-Delta-H.

This paper traces the following abstract template of results similar to Yin et. al. (2018) and Awasthi. et. al. (2019) but for the task of robust domain adaptation.
- Define an adversarial Rademacher complexity via the adversarial loss function. (Here in addition the hypothesis class is H-Delta-H and the adversarial discrepancy is defined).
- Show that adversarial Rademacher complexity is bounded by standard-Rademacher complexity + an expression that depends on the norm of the predictors and size of corruption.
- Show that adversarial Rademacher complexity can be lower bounded by the standard complexity with a dimensional cost.
- Instantiate the adversarial Rademacher complexity upper bound for linear predictors and 1-hidden layer neural networks.

In addition to the above, the authors show that adversarial target risk can be linked to the adversarial source risk via the adversarial Rademacher complexity and adversarial discrepancy distance. This is in the same spirit as the result from Mansour (2009).

Certainly establishing the above points require sufficient effort and the authors appear to have done so rigorously. Finally, the authors establish a link between standard target risk and adversarial source risk - a result that can potentially explain why adversarially trained classifiers have better standard domain adaptation.


**Summary Of The Review:**

I recommend accept as I believe this work clears the bar of sound technical contribution.

---

> ### Author Response · Authors · 2022-11-13
> **Response to Reviewer Du9x**
>
> Thank you for your thorough reading of our article, and your compliment on our work. We are glad that the structure of our paper appears clear to you. We will still try to clarify a bit that even though the story is similar to Yin et al, but the proof technique significantly differs with theirs, as we mentioned in general comments to all reviewers. We will definitely clarify these difference in the revised version!

---

> ### Author Response · Authors · 2022-11-18
> **Rebuttal Revision Submitted**
>
> Dear Reviewer Du9x
>
> Thanks again for your compliment and accurate comments. We have submitted revised version, and if you have any questions/comments, please feel free to post.
>
> Look forward to hearing from you soon.
>
> Thank you,
> The author of submission 1294.

---

> ### Author Response · Authors · 2022-11-27
> **A Kindly Reminder**
>
> Dear Reviewer Du9x,
>
> Thank you again for your constructive and complimentary comments, and we sincerely look forward to your valuable feedback.
> If you have any further questions, please feel free to ask, and we are more than happy to explain/discuss with you.
>
> Thank you,
> The authors of paper 1294

---

> > ### Comment · Reviewer_Du9x · 2022-12-08
> > **Response to Authors**
> >
> > Hi,
> > I'm fairly convinced on the merits and novelty of the proof techniques following the rebuttal from authors. I recommend accept.

---

> > > ### Author Response · Authors · 2022-12-08
> > > **Thank you for your feedback**
> > >
> > > Dear Reviewer Du9x,
> > >
> > > Thank you for your post-rebuttal feedback, and we are glad that our rebuttal looks clear to you. Thanks for your compliments on our work and improving the score as well!
> > >
> > > Thank you,
> > > The authors of paper 1294

---

### Official Review · Reviewer_RkC2 · 2022-10-25

**Confidence:** 4
**Correctness:** 3
**Technical Novelty And Significance:** 3
**Empirical Novelty And Significance:** 3
**Recommendation:** 6

**Clarity, Quality, Novelty And Reproducibility:**

The paper is clear, and the quality of the contribution is adequate. For reasons outline above, I am not sure about certain aspects of the novelty/originality of the submission (particularly Q2 and the methodology to get bounds in Sections 3 and 4). Other parts of the paper (e.g., bringing robust learning and domain adaptation together, the particular work on the class $\mathcal{H}\Delta\mathcal{H}$ through Rademacher complexity) are novel and interesting.


**Strength And Weaknesses:**

Note: Work cited in parentheses refers to the authors’ bibliography. Work cited in the form [ABC12] is for this review and the bibliography appears at the end of the review.

## Strong points

The work is timely and interesting, and brings new ideas to the field of robustness and domain adaptation. The paper is clear and well-written. Lots of results are presented.

## Weak points/omissions/clarifications:

I think there are issues in the definitions of the problem (in the definition of robustness), and the related work needs clarifications.

**Robust Risk Definitions**

Adversarial risks defined on p.2 and p.3:
The adversarial risk defined on p.2 is not the same as the adversarial risk defined on p.3. Indeed, in the former, the label of the perturbed point is compared to the label $y_{\mathcal{D}}(x)$ of the *unperturbed* point, while in the latter, it is compared to the another hypothesis’  labelling (potentially the ground truth) *also* on the perturbed point. The risk on p.2 has been defined in the literature as corrupted instantce/constant-in-the-ball robustness, and the second as error region/exact-in-the-ball.

It would be necessary to name the robust risk(s) used in this paper, as when the two definitions don’t coincide, the guarantees obtained have a different meaning.  On one hand, the error region/exact-in-the-ball risk talks about the probability measure of the expansion of the error region; we ask that the target and hypothesis *agree* in the perturbation region. On the other hand, for the more commonly-used robust risk in the literature, which is the corrupted instance/constant-in-the-ball one, a correctly classified point should not change labels if it is perturbed.

In Lemma 2 for example, a key assumption is that the loss function is symmetric and obeys the triangle inequality. This is the case for the exact-in-the-ball/error region one, but not for the constant-in-the-ball, because the first labelling function is w.r.t. a perturbed point, while the second is w.r.t. an unperturbed point. I think this poses some issues with the triangle inequality too.

(Q1:) In this paper, it seems that most results use the exact-in-the-ball while Lemma 5 and Corollary 1, and the adversarial training procedure in section 6 appear to use the constant-in-the-ball. I have skimmed through the proof of Lemma 5, and it does not seem to be using prior results in the paper — can you confirm that the proof is “stand alone”, and for this particular notion of robustness? Otherwise, there could potentially be a mistake in there.

(Q2:) Moreover, with this definition of robustness, I think that the adversarial RC used in this result is the same one as in (Yin et al 2018). If so, could you comment on the difference between your results and theirs (Thm 2 in particular), as well as other follow-up work using the same notion of robustness?

Work that has compared different notions of robustness, to cite/read for reference: [DMM18], [GKKW21] and [PJ21]. All papers also look at fundamental limitations of robust learning compared to standard learning.


**Related Work**

(Diochnos et al. 2019). The exponential dependence on the budget not totally right: in the Theorem 3.4 statement, we can see that the budget (= number of bits) is defined as $b = \rho \cdot n$, so the exponential dependence on $\rho$ is actually the normalized budget. E.g., the bounds are superpolynomial only when the number of bits the adversary can flip at test time is $\omega(\sqrt{n\log(n)})$. The work of [GKKW22] gives an actual exponential dependence on the number of bits ($=b$) the adversary is allowed to perturb (though for a more restricted setting), so a superpolynomial lower bound happens when the budget is $\omega(\log(n))$ instead. In both works the dependence on the budget is for the error region/exact-in-the-ball robustness and is for specific problems: assumptions on concept classes and distribution families are necessary to obtain such guarantees — closeness and Normal Lévy families for (Diochnos et al. 2019) and monotone conjunctions (and superclasses like linear classifiers and decision lists) under the uniform distribution for [GKKW22].

(PAC Learning frameworks for robustness). The works of (Cullina et al. 2018) and (Montasser et al. 2019) use the corrupted instance/constant-in-the-ball robustness, while the work of (Diochnos et al. 2019) uses the error region/exact-in-the-ball one. This means that the guarantees obtained differ in meaning and are general incomparable (e.g. the perturbation budget doesn’t explicitly appear in the work of (Montasser et al. 2019)). Also follow-up works [MHS21] look at implementing this algorithm with the use of a Perfect Adversary Oracle and [MHS22] gives a characterization of robust learnability through a minimax learner. The notion of robustness used by each work should be stated explicitly.

(The $\mathcal{H}\Delta\mathcal{H}$ class). (Cullina et al. 2018) also had a look at the robust loss of a hypothesis class, though for the constant-in-the-ball/corrupted instance notion of robustness and through the adversarial VC dimension, defined in their work. [APU20] also looks at the margin class (robust loss function) and relate its VC dimension to the sample complexity of robust learning for this notion of robustness. Recent work [GKKW22b] looks at the adversarial perturbation region of $\mathcal{C}\Delta\mathcal{H}$ (i.e. the robust loss functions w.r.t. to exact-in-the-ball/error region robustness), where $\mathcal{C}$ is the set of possible labelling functions, though through the lens of VC dimension as well, similarly to (Cullina et al. 2018) and [APU20]. Since the VC dimension and Rademacher complexity are related, there may be a link here.

## Other Questions

- How do the goals of robust learning and (standard and adversarial) domain adaptation differ? It would be a good idea to devote a section in the appendix to go into more details on this.
- Could you comment on the methodology used to get RC upper bounds in Sections 3 and 4, and how they differ in approach of the works mentioned in Appendix A?
- On p.8, you say that “If the adversarial budget $\epsilon$ is larger, $\Lambda$ will be larger, which means the quantity $V^*(p’,p,\ell,\Lambda)$ will be smaller." Can you explicitly state this dependence?

**Typos/Comments**

- p.2, second paragraph: a few typos:

"That is, the gap between robust risks on old domain and new domain can be dramatically huge..." -> "That is, the gap between robust risks on the old and new domains can be dramatically huge...";

"This observation naturally leads to the question of, *why the robust risk is harder to adapt to different domains?*, which we aim to examine in this paper." -> "This observation naturally leads to the question of why the robust risk is harder to adapt to different domains, which we aim to examine in this paper." or "This observation naturally leads to the question *Why is the robust risk harder to adapt to different domains?*, which we aim to examine in this paper.";

"...we properly extend this complexity measure to adversarial learning setting, and propose the adversarial Radmeacher complexity over H∆H class." -> "...we properly extend this complexity measure to *the* adversarial learning setting, and propose the adversarial Radmeacher complexity over *the* H∆H class."
- p.3, paragraph above Defn 2: “source domain $\mathcal{S}$ and target domain $\mathcal{T}$” — is there a “distribution” missing after this? Also, $\hat{\mathcal{T}}$ and $\hat{\mathcal{S}}$ should be switched.
- p.3, paragraph after Defn 2: “The another advantage” -> “Another advantage”
- p.3 paragraph above Defn 3: $d_{H\Delta H}$ -> $disc_{H\Delta H}$?
- p.3, last sentence: “to adversarial setting” -> “to the adversarial setting”
- p.4, paragraph after Defn 4: I think this is misleading: Lemma 19 is only derived for linear classifiers, while the sentence implies that the result is more general.
- p.4, Lemma 2 statement: in the first term on the RHS of the inequality (adversarial RC w.r.t. $\mathcal{S}$), should $\mathbf{w}$ be $h_{\mathbf{w}}$ instead?

**References**

[APU20] Ashtiani, H., Pathak, V., and Urner, R. (2020)  Black-box certification and learning under adversarial perturbations. In *International Conference on Machine Learning*.

[DMM18] Diochnos, D., Mahloujifar, S., and Mahmoody, M. (2018). Adversarial risk and robustness: General definitions and implications for the uniform distribution. In Advances in Neural Information Processing Systems.

[GKKW21] Gourdeau, P., Kanade, V., Kwiatkowska, M., and Worrell, J. (2021). On the hardness of robust classification. In *Journal of Machine Learning Research*, 22.

[GKKW22] Gourdeau, P., Kanade, V., Kwiatkowska, M., and Worrell, J. (2022). Sample complexity bounds for robustly learning decision lists against evasion attacks. In *International Joint Conference in Artificial Intelligence*.

[GKKW22b] Gourdeau, P., Kanade, V., Kwiatkowska, M., and Worrell, J. (2022). When are local queries useful for robust learning?. In *Advances in Neural Information Processing Systems*.

[MHS21] Montasser, O., Hanneke, S. and Srebro, N. (2021). Adversarially Robust Learning with Unknown Perturbation Sets. In *Conference on Learning Theory*.

[MHS22] Montasser, O., Hanneke, S. and Srebro, N. (2022). Adversarially Robust Learning: A Generic Minimax Optimal Learner and Characterization. In *Advances in Neural Information Processing Systems*.

[PJ21] Pydi, M. S. and Jog, V. (2021). The many faces of adversarial risk. In *Advances in Neural Information Processing Systems*.

**Summary Of The Paper:**

This paper studies robust learning and domain adaptation from a learning theory perspective. It does so notably by looking at the adversarial Rademacher complexity (RC) of the robust loss function between two classes. Various results are presented: adversarially robust domain adaptation (Lemma 2), an upper bound on the RC of binary classification for linear classifiers (Lemma 3), upper and lower bounds on the adversarial RC of binary classification for linear classifiers (Theorem 1), as well as similar results for regression tasks (linear classifiers) and neural networks with ReLU activations. They finally related the robust and standard risks for linear classifiers. The authors finish with an empirical study.

**Summary Of The Review:**

I think this is interesting work, with novel contributions. I believe there remains work to be done in how the paper situates itself in the literature, and that certain aspects of the contribution should be explained in more details.

I gave a 5/10, because of the questions above. I am open to changing my score after the response from the authors, and I am looking forward to our discussion.

---

> ### Author Response · Authors · 2022-11-13
> **Response to Review RkC2 (1/2)**
>
> We appreciate your detailed, accurate and constructive comments and we are sorry for any inconvenience caused by the grammatical and writing problems. We will address such problems and improve the writing quality for the final version. Your suggestions of related works studying other notions of robustness will be extremely helpful for comparison with our work in our introduction.
> Moreover, we will gladly address your questions and incorporate them in the revised version as detailed below:
>
> # Definition of robust risks:
> Thank you for spotting this vocabulary inconsistency. Indeed, we need to clarify our naming of these risks to be more precise. We referred to [GKKW21], and following their naming convention, we will rename the first risk constant-in-the-ball adversarial risk and write it as $\mathcal{R}^{label-adv}\_\mathcal{D}(h\_\mathbf{w},y\_\mathcal{D}) := \mathbb{E}\_{\mathbf{x}\sim \mathcal{D}} \left[\max_{\|\boldsymbol{\delta}\|\_\infty\leq \epsilon}\ell (h\_\mathbf{w}(\mathbf{x}+\boldsymbol{\delta}), y\_\mathcal{D}(\mathbf{x}))\right]$ and $\mathcal{R}^{adv}\_{\mathcal{D}}(h\_\mathbf{w},h\_{\mathbf{w}'}) := \mathbb{E}\_{\mathbf{x}\sim \mathcal{D}}  [\max_{\|\boldsymbol{\delta}\|\_\infty\leq \epsilon}\ell (h\_\mathbf{w}(\mathbf{x}+\boldsymbol{\delta}), h\_{\mathbf{w}'}(\mathbf{x}+\boldsymbol{\delta})) ]$.
> We will adapt this change in the revised version of article.
>
> However, we believe this confusion about the naming of these two objects does not have any impact on our theory.
> Lemma 2 still holds as triangle inequality is applied at loss level, not prediction nor input point level.
> Indeed, if we assume that the loss obeys triangle inequality, i.e., $\ell(a,b) \leq \ell(a,c)+\ell(c,b)$, for all $a,b,c$, then when starting on the left-hand side with the constant-in-the-ball adversarial risk, we get:
> $$
> \begin{align}
>     \mathcal{R}^{label-adv}\_\mathcal{T}(h\_\mathbf{w},y\_\mathcal{T}) &= \mathbb{E}\_{\mathbf{x}\sim \mathcal{T}} \left[\max\_{\\|\boldsymbol{\delta}\\|\_\infty\leq \epsilon} \ell (h\_\mathbf{w}(\mathbf{x}+\boldsymbol{\delta}), y\_\mathcal{T} (\mathbf{x}))\right] \\\\
>     &\leq  \mathbb{E}\_{\mathbf{x}\sim \mathcal{T}} \left[\max_{\\|\boldsymbol{\delta}\\|\_\infty\leq \epsilon} \ell (h\_\mathbf{w}(\mathbf{x}+\boldsymbol{\delta}), h\_\mathbf{w\_{\mathcal{T}}^*}(\mathbf{x}+\boldsymbol{\delta})) + \ell (h\_\mathbf{w\_{\mathcal{T}}^*}(\mathbf{x}+\boldsymbol{\delta}), y\_\mathcal{T} (\mathbf{x}))\right] \\\\
>     &\leq \mathbb{E}\_{\mathbf{x}\sim \mathcal{T}} \left[\max\_{\\|\boldsymbol{\delta}\\|\_\infty\leq \epsilon} \ell (h\_\mathbf{w}(\mathbf{x}+\boldsymbol{\delta}),  h\_\mathbf{w\_{\mathcal{T}}^*}(\mathbf{x}+\boldsymbol{\delta}))\right] +
>      \mathbb{E}\_{\mathbf{x}\sim \mathcal{T}} \left[\max\_{\\|\boldsymbol{\delta}\\|\_\infty\leq \epsilon} \ell (h\_\mathbf{w\_{\mathcal{T}}^*}(\mathbf{x}+\boldsymbol{\delta}), y\_\mathcal{T} (\mathbf{x}))\right] \\\\
>     &= \mathcal{R}^{adv}\_\mathcal{T} ( h\_\mathbf{w} , h\_\mathbf{w\_{\mathcal{T}}^*} ) + \mathcal{R}^{label-adv}\_\mathcal{T} ( h\_\mathbf{w\_{\mathcal{T}}^*} , y\_\mathcal{T} ) \\\\
>    & \leq \mathcal{R}^{adv}\_\mathcal{T} ( h\_\mathbf{w} , h\_\mathbf{w\_{\mathcal{S}}^*}) + \mathcal{R}^{adv}\_\mathcal{T} ( h\_\mathbf{w\_{\mathcal{S}}^*} , h\_\mathbf{w\_{\mathcal{T}}^*} ) + \mathcal{R}^{label-adv}\_\mathcal{T} ( h\_\mathbf{w\_{\mathcal{T}}^*}, y\_\mathcal{T} )\\\\
>   &  \leq \mathcal{R}^{adv}\_\mathcal{S} ( h\_\mathbf{w} , h\_\mathbf{w\_{\mathcal{S}}^*}) + disc^{adv}\_{\mathcal{H}\Delta\mathcal{H}}(\mathcal{S},\mathcal{T}) + \mathcal{R}^{adv}\_\mathcal{T} (h\_\mathbf{w\_{\mathcal{S}}^*}, h\_\mathbf{w\_{\mathcal{T}}^*} ) + \mathcal{R}^{label-adv}\_\mathcal{T} (h\_\mathbf{w\_{\mathcal{T}}^*}, y\_\mathcal{T} )
> \end{align}
> $$
> where the risk decomposition still holds. We will definitely update the notations in the revised article and thank you again for pointing this out!
>
> # Regarding Lemma 5
> We agree with you that the technique to prove Lemma 5 differs to previous sections. After our clarification on the notions of risks, in Lemma 5, we want to show that small robust risk $\mathcal{R}\_{\mathcal{T}}^{label-adv}(h\_{\mathbf{w}},y\_\mathcal{T})$ will imply small standard risk on any other domain $\mathcal{T}'$, with the residual error controlled by adversarial budget.
>
> # Difference of the goals of robust learning and (standard and adversarial) domain adaptation
> In the goal of robust learning, we care about the robust risk and empirical robust risk, on the same domain; While in standard domain adaptation, we consider the (standard) risk on the target domain and the risk on the source domain. In adversarially robust domain adaptation, we care about the relation between adversarially robust risks on target and source domain, respectively. We will add a explanatory paragraph in the Appendix of the revised version.

---

> > ### Author Response · Authors · 2022-11-13
> > **Response to Reviewer RkC2 (2/2)**
> >
> > # Technical difference compared to Yin et al
> > Taking classification setting for example, Yin et al consider the Rademacher complexity over the loss class between model predictions and labels, i.e., $\frak{R} = \mathbb{E}[\sup_{\mathbf{w}}\frac{1}{n}\sum_{i=1}^n \sigma_i \min_{\|\boldsymbol{\delta}\|\leq \infty} \mathbf{w}^\top (\mathbf{x}+\boldsymbol{\delta})]$, where the inner minimization problem is $\textbf{linear}$ in $\mathbf{w}$ and $\boldsymbol{\delta}$. We consider the loss between predictions among two models, hence the Rademacher complexity is $\frak{R} = \mathbb{E}[\sup_{\mathbf{w},\mathbf{w}'}\frac{1}{n}\sum_{i=1}^n \sigma_i \min_{\|\boldsymbol{\delta}\|\leq \infty} \mathbf{w}^\top (\mathbf{x}+\boldsymbol{\delta})\mathbf{w}'^\top (\mathbf{x}+\boldsymbol{\delta})]$, where the inner problem is quadratic in terms of $\mathbf{w}$ and $\boldsymbol{\delta}$. Hence, unlike previous works on single domain Rademacher complexity, where the inner problem has simple closed form solution, we have to use $\varepsilon$-nets and covering number idea to estimate upper bound. We would also like to emphasize that another key contribution of this submission is the lower bound proof, where we derive the (complicated) closed form solution to inner quadratic programming, and leverage the symmetric property of Rademacher random variables to avoid heavy computation. Here we also briefly talk about the comparisons of technique aspects of the related works. Yin et al succeed to separate randomness coming from Rademacher random variables and a deterministic terms. Then, they apply classical Khintchine’s inequality to bound the randomness.
> > Unlike them, we cannot find closed form solution for the inside minimization problem. So, we tackle all terms together by applying one of Tropp's matrix concentration bound.
> > Concerning the neural network case, (Khim \& Loh, 2018) use a tree transform to get upper bound of the RC, whereas (Yin et al, 2019) make appear a SDP relaxation. On our side, we do not aim at solving such a surrogate problem. Instead we separate our upper bound into three average of random matrices and again, apply the matrix Bernstein inequality.
> >
> > # Regarding dependency of $V$ on $\epsilon$
> > Recall the definition of $\Lambda$: $\Lambda = \\{i: |\mathbf{w}^\top \mathbf{x}\_i| \leq \epsilon\ \|\mathbf{w}\\|\_1, \mathbf{x}\_i \in \mathcal{X}, \forall \mathbf{w}, \\|\mathbf{w}\\|\_p \leq W\\}$. Fix $\\{\mathbf{x}\_i\\}$ and $W$, if $\epsilon$ is larger, then it means that we allow more $i$ such that the inequality $|\mathbf{w}^\top \mathbf{x}\_i| \leq \epsilon \\|\mathbf{w}\\|\_1$ can hold. Actually, the cardinality of this set is $|\Lambda| = \sum_{i=1}^{|\mathcal{X}|} \mathbf{1}\\{\sup_{\mathbf{w}:\|\mathbf{w}\|_p \leq W}\frac{|\mathbf{w}^\top \mathbf{x}_i|}{\|\mathbf{w}\|_1} \leq \epsilon \\}$. If the $\Lambda$ includes more indices, then we are able to flip more signs of $\boldsymbol{\ell}'$ so that the optimal value of Subset Sum Problem $V^*$ will be smaller. Unfortuantely, since the problem is combintoric and NP-complete, we are not able to give analytic form of $V^*$ in terms of $\epsilon$.
> >
> > We also appreciate your pointing out the typos and we will address them in the revised version!

---

> ### Author Response · Authors · 2022-11-18
> **Rebuttal Revision Submitted**
>
> Dear Reviewer RkC2
>
> Again thank you for your very professional and constructive comments, and we have incorporated your suggestions or solutions to your concerns in the revised version (significant changes are highlighted in blue). In particular, we correct the definitions for two kinds of robust risks, and fixed the Lemma 2, the key generalization Lemma, which we believe will address your concerns.
>
> If you have more questions, please feel free to make a comment, and we are more than happy to hear from you soon.
>
> Thank you,
> The authors of submission 1294

---

> ### Author Response · Authors · 2022-11-27
> **A Kindly Reminder**
>
> Dear Reviewer RkC2,
>
> Thank you again for your constructive comments, and we sincerely anticipate your valuable feedback. We believe our response will resolve your main concern about robust risk definitions, and we also adapt your suggestions on improving the paper quality. The authors really hope you can go through them and let us know if they addressed your concerns.
>
> If you have any further questions, please feel free to ask, and we are more than happy to explain/discuss with you.
>
> Thank you,
> The authors of paper 1294

---

> ### Author Response · Authors · 2022-12-03
> **A friendly reminder that the discussion stage 2 will be closed in 10 days**
>
> Dear Reviewer RkC2,
>
> Thank you again for your professional comments. We would like to hear from you about whether our rebuttal addresses your concerns. If not, please let us know so that we can provide further explanation timely. We are also more than happy to discuss with you on any technical points of this paper!
>
> Thanks,
> The authors of submission 1294

---

> ### Author Response · Authors · 2022-12-08
> **A friendly reminder that the discussion stage will be closed in 4 days**
>
> Dear Reviewer RkC2,
>
> Thank you again for your professional comments. Since the discussion stage will end in 4 days, we kindly ask you to respond to our rebuttal so that we can have time to address your further concern/questions.
>
> Thanks, The authors of submission 1294

---

> > ### Comment · Reviewer_RkC2 · 2022-12-08
> > **Response**
> >
> > Dear authors,
> >
> > Many apologies for the delay and thank you for providing an updated copy of your paper and clarifying comments.
> >
> > I have the following remaining concern:
> >
> > - In Lemma 2, you say that the loss function is symmetric. Suppose we consider a point $x$, a hypothesis $h_w$ such that $h_w(x')=0$ for all $x'=x+\delta$ for $\Vert\delta\Vert_\infty\leq\epsilon$ (i.e. $h_w$ is constant in the perturbation region) and a labeling function $y(x)=0$ for all the points except a unique $x^*$ such that $\Vert x-x^* \Vert\leq \delta$ where $y(x^*)=1$, then the robust loss $\ell_\delta(\cdot,\cdot,\cdot)$ inducing the robust risk $\mathcal{R}^{adv-label}$ for this specific point $x$ satisfies $$\ell_\delta(x,h_w,y)=\mathbf{1}[\max_\delta \ell(h_w(x+\delta),y(x))]=\mathbf{1}[\max_\delta \ell(h_w(x+\delta),0)]=0$$ as $h_w$ is constant in the perturbation region but $$\ell_\delta(x,y,h_w)=\mathbf{1}[\max_\delta \ell(y(x+\delta),h_w(x))]=\mathbf{1}[\max_\delta \ell(y(x+\delta),0)]=1$$ from the existence of $x^*$. The robust loss (which is defined w.r.t. a specific input point), is not symmetric. If one puts all of the probability mass on $x$, it follows that the robust risk $\mathcal{R}^{adv-label}$ is not symmetric in general. But this is the loss used in Lemma 2.
> >
> > Typos/comments:
> > - It would be good to add additional terminology for the robust risks, which was mentioned in the first review: [DMM18] calls the exact-in-the-ball risk "error region risk" and the constant-int-the-ball "corrupted instance risk" from the work of [FMS15]
> > -  over $\mathcal{H}\Delta \mathcal{H}$ class -> over the $\mathcal{H}\Delta \mathcal{H}$ class (twice in the abstract)
> > - shed lights -> shed light (abstract)
> >
> > [FMS15] Feige, Mansour, Schapire. Learning and inference in the presence of corrupted inputs. COLT 2015

---

> > > ### Author Response · Authors · 2022-12-08
> > > **Thank you for your feedback**
> > >
> > > Dear Reviewer RkC2,
> > >
> > > Thank you for your feedback and we appreciate your careful reading on our updated version.
> > >
> > > Sorry for the confusion on the loss definition because of our unclear definition of $\tilde{\ell}$. We will try to address your concern as follows.
> > >
> > > First, In the definition of $\mathcal{R}^{adv-label}$, we assume the perturbation $\delta$ is only applied on the model $h$, not labeling function $y$, so in our setting we avoid the issue you mentioned. Indeed, we only require the symmetric property of $\tilde{\ell}(h,h'):= \max_{\delta} \ell (h(x+\delta), h'(x+\delta))$  just for making the adversarial discrepancy symmetry. We will definitely clarify this in the future version, and many thanks for your catching this point!
> > >
> > > We would also like to thank you for the additional typos/references you provided, and we will correct them in the future version.
> > >
> > > Thank you,
> > >
> > > The Authors of paper 1294

---

> > > > ### Comment · Reviewer_RkC2 · 2022-12-10
> > > > **Clarification**
> > > >
> > > > Thank you for your response.
> > > >
> > > > Just to be sure: where do you use the symmetry property in the proof of Lemma 2?
> > > >
> > > > I'm also confused in the proof of Lemma 2 in Appx E, because of the risk decomposition in the first line. On the LHS, you have $\mathcal{R}^{adv}$ but $\mathcal{R}^{adv-label}$ on the RHS (and then the opposite change in the 4th inequality). I tried finding a justification for this but couldn't. Also the first sentence of the proof doesn't make sense compared to the equations (it talks about $\mathcal{R}^{adv}(\mathbf{w},\mathbf{v}^*_T)$).
> > > >
> > > > Best,
> > > > Reviewer RkC2

---

> > > > > ### Author Response · Authors · 2022-12-10
> > > > > **Thanks for your questions**
> > > > >
> > > > > Thanks for your questions. There actually is somewhere we used symmetric property, which we will clarify while answering your questions about the proof.
> > > > >
> > > > > For you concern about proof of Lemma 2, we apologize the typo of the first sentence. It should be $\mathcal{R}^{label-adv}\_\mathcal{T}(h\_\mathbf{w},y\_\mathcal{T}) $. We will fix that and thank you for catching that!
> > > > >  The first line holds because the loss obeys triangle inequality:
> > > > >
> > > > > $$
> > > > > \begin{align}
> > > > >     \mathcal{R}^{adv-label}\_\mathcal{T}(h\_\mathbf{w},y\_\mathcal{T}) &= \mathbb{E}\_{\mathbf{x}\sim \mathcal{T}} \left[\max\_{\\|\boldsymbol{\delta}\\|\_\infty\leq \epsilon} \ell (h\_\mathbf{w}(\mathbf{x}+\boldsymbol{\delta}), y\_\mathcal{T} (\mathbf{x}))\right] \\\\
> > > > >     &\leq  \mathbb{E}\_{\mathbf{x}\sim \mathcal{T}} \left[\max_{\\|\boldsymbol{\delta}\\|\_\infty\leq \epsilon} \ell (h\_\mathbf{w}(\mathbf{x}+\boldsymbol{\delta}), h\_\mathbf{w\_{\mathcal{T}}^*}(\mathbf{x}+\boldsymbol{\delta})) + \ell (h\_\mathbf{w\_{\mathcal{T}}^*}(\mathbf{x}+\boldsymbol{\delta}), y\_\mathcal{T} (\mathbf{x}))\right] \\\\
> > > > >     &\leq  \mathbb{E}\_{\mathbf{x}\sim \mathcal{T}} \left[\max\_{\\|\boldsymbol{\delta}\\|\_\infty\leq \epsilon} \ell (h\_\mathbf{w}(\mathbf{x}+\boldsymbol{\delta}),  h\_\mathbf{w\_{\mathcal{T}}^*}(\mathbf{x}+\boldsymbol{\delta}))\right] +
> > > > >      \mathbb{E}\_{\mathbf{x}\sim \mathcal{T}} \left[\max\_{\\|\boldsymbol{\delta}\\|\_\infty\leq \epsilon} \ell (h\_\mathbf{w\_{\mathcal{T}}^*}(\mathbf{x}+\boldsymbol{\delta}), y\_\mathcal{T} (\mathbf{x}))\right] \\\\
> > > > >     &= \mathcal{R}^{adv}\_\mathcal{T} ( h\_\mathbf{w} , h\_\mathbf{w\_{\mathcal{T}}^*} ) + \mathcal{R}^{label-adv}\_\mathcal{T} ( h\_\mathbf{w\_{\mathcal{T}}^*} , y\_\mathcal{T} ) \\\\
> > > > > \end{align}
> > > > > $$
> > > > >
> > > > > So that we decomposed $\mathcal{R}^{adv-label}\_\mathcal{T}(h\_\mathbf{w},y\_\mathcal{T}) $ into $\mathcal{R}^{adv}\_\mathcal{T} ( h\_\mathbf{w} , h\_\mathbf{w\_{\mathcal{T}}^*} ) + \mathcal{R}^{adv-label}\_\mathcal{T} ( h\_\mathbf{w\_{\mathcal{T}}^*} , y\_\mathcal{T} )$. The same thing happens in the 4th inequality:
> > > > > $$
> > > > > \begin{align}
> > > > >     \mathcal{R}^{adv}\_\mathcal{S}(h\_\mathbf{w},h\_\mathbf{w\_{\mathcal{S}}^*} ) &= \mathbb{E}\_{\mathbf{x}\sim \mathcal{S}} \left[\max\_{\\|\boldsymbol{\delta}\\|\_\infty\leq \epsilon} \ell (h\_\mathbf{w}(\mathbf{x}+\boldsymbol{\delta}), h\_{\mathbf{w}^*_\mathcal{S}}(\mathbf{x}+\boldsymbol{\delta}))\right] \\\\
> > > > >     &\leq   \mathbb{E}\_{\mathbf{x}\sim \mathcal{S}} \left[\max\_{\\|\boldsymbol{\delta}\\|\_\infty\leq \epsilon} \ell (h\_\mathbf{w}(\mathbf{x}+\boldsymbol{\delta}),  y\_{\mathcal{S} }(\mathbf{x}) ))\right] +  \mathbb{E}\_{\mathbf{x}\sim \mathcal{S}} \left[\max\_{\\|\boldsymbol{\delta}\\|\_\infty\leq \epsilon} \ell (y\_{\mathcal{S}}(\mathbf{x}), h\_{\mathbf{w}^*_\mathcal{S}}(\mathbf{x}+\boldsymbol{\delta}))\right] \\\\
> > > > > & =  \mathcal{R}^{adv-label}\_\mathcal{S}(h\_\mathbf{w} ) + \mathcal{R}^{adv-label}\_\mathcal{S}(h\_\mathbf{w\_{\mathcal{S}}^*} )
> > > > > \end{align}
> > > > > $$
> > > > > where the last step holds due to triangle inequality which allows us to switch $y\_{\mathcal{S}}$ and $h\_{\mathbf{w}^*_\mathcal{S}}(\mathbf{x}+\boldsymbol{\delta})$.
> > > > >
> > > > > Thank you,
> > > > >
> > > > > The authors of paper 1294

---

> > > > > > ### Comment · Reviewer_RkC2 · 2022-12-11
> > > > > > **Comment**
> > > > > >
> > > > > > Thank you for all the responses above.
> > > > > >
> > > > > > I think you should add this in the proof in the appendix as it is not immediately obvious.
> > > > > >
> > > > > > I have updated my score from 5 to 6.
> > > > > >
> > > > > > Best,
> > > > > > RkC2

---

> > > > > > > ### Author Response · Authors · 2022-12-12
> > > > > > > **Thank you for improving the score**
> > > > > > >
> > > > > > > Dear Reviewer RkC2,
> > > > > > >
> > > > > > > Thank you so much for your feedback and improving the score! We also appreciate your constructive comments which do make this paper better!
> > > > > > >
> > > > > > > Thanks,
> > > > > > >
> > > > > > > The author of paper 1294

---

### Author Response · Authors · 2022-11-13
**General  comments: Definition of Robust Risks**

Dear Reviewers and AC,

Thank you for your valuable works during the reviewing stage. We appreciate the compliments from Reviewer RkC2 and Du9x regarding the novelty of our work, and we try to address some common questions here.

# 1. Definition of Robust Risks (Reviewer RkC2 and K4Lj)
We admit that we abuse the notations, and sorry for any inconvenience on reading. Indeed, we should define the following two risks, one is between prediction and label, the other is between predictions of two models:
$\mathcal{R}^{label-adv}\_\mathcal{D}(h\_\mathbf{w},y\_\mathcal{D}) := \mathbb{E}\_{\mathbf{x}\sim \mathcal{D}} \left[\max_{\|\boldsymbol{\delta}\|\_\infty\leq \epsilon}\ell (h\_\mathbf{w}(\mathbf{x}+\boldsymbol{\delta}), y\_\mathcal{D}(\mathbf{x}))\right]$ and $\mathcal{R}^{adv}\_{\mathcal{D}}(h\_\mathbf{w},h\_{\mathbf{w}'}) := \mathbb{E}\_{\mathbf{x}\sim \mathcal{D}}  [\max_{\|\boldsymbol{\delta}\|\_\infty\leq \epsilon}\ell (h\_\mathbf{w}(\mathbf{x}+\boldsymbol{\delta}), h\_{\mathbf{w}'}(\mathbf{x}+\boldsymbol{\delta})) ]$.  In words, whenever we compare model prediction and labels, we do not disturb input data in labeling function, and that is constant-in-the-ball risk mentioned by Reviewer RkC2. While when we are comparing two models predictions, we disturb the input data of two models, and it is so-called exact-in-the-ball risk. However, this notational abuse will not impact our proof. For example, in proving Lemma 2, if we adapt corrected notations, we have:
$$
\begin{align}
    \mathcal{R}^{label-adv}\_\mathcal{T}(h\_\mathbf{w},y\_\mathcal{T}) &= \mathbb{E}\_{\mathbf{x}\sim \mathcal{T}} \left[\max\_{\\|\boldsymbol{\delta}\\|\_\infty\leq \epsilon} \ell (h\_\mathbf{w}(\mathbf{x}+\boldsymbol{\delta}), y\_\mathcal{T} (\mathbf{x}))\right] \\\\
    &\leq  \mathbb{E}\_{\mathbf{x}\sim \mathcal{T}} \left[\max_{\\|\boldsymbol{\delta}\\|\_\infty\leq \epsilon} \ell (h\_\mathbf{w}(\mathbf{x}+\boldsymbol{\delta}), h\_\mathbf{w\_{\mathcal{T}}^*}(\mathbf{x}+\boldsymbol{\delta})) + \ell (h\_\mathbf{w\_{\mathcal{T}}^*}(\mathbf{x}+\boldsymbol{\delta}), y\_\mathcal{T} (\mathbf{x}))\right] \\\\
    &\leq  \mathbb{E}\_{\mathbf{x}\sim \mathcal{T}} \left[\max\_{\\|\boldsymbol{\delta}\\|\_\infty\leq \epsilon} \ell (h\_\mathbf{w}(\mathbf{x}+\boldsymbol{\delta}),  h\_\mathbf{w\_{\mathcal{T}}^*}(\mathbf{x}+\boldsymbol{\delta}))\right] +
     \mathbb{E}\_{\mathbf{x}\sim \mathcal{T}} \left[\max\_{\\|\boldsymbol{\delta}\\|\_\infty\leq \epsilon} \ell (h\_\mathbf{w\_{\mathcal{T}}^*}(\mathbf{x}+\boldsymbol{\delta}), y\_\mathcal{T} (\mathbf{x}))\right] \\\\
    &= \mathcal{R}^{adv}\_\mathcal{T} ( h\_\mathbf{w} , h\_\mathbf{w\_{\mathcal{T}}^*} ) + \mathcal{R}^{label-adv}\_\mathcal{T} ( h\_\mathbf{w\_{\mathcal{T}}^*} , y\_\mathcal{T} ) \\\\
   & \leq \mathcal{R}^{adv}\_\mathcal{T} ( h\_\mathbf{w} , h\_\mathbf{w\_{\mathcal{S}}^*}) + \mathcal{R}^{adv}\_\mathcal{T} ( h\_\mathbf{w\_{\mathcal{S}}^*} , h\_\mathbf{w\_{\mathcal{T}}^*} ) + \mathcal{R}^{label-adv}\_\mathcal{T} ( h\_\mathbf{w\_{\mathcal{T}}^*}, y\_\mathcal{T} )\\\\
  &  \leq \mathcal{R}^{adv}\_\mathcal{S} ( h\_\mathbf{w} , h\_\mathbf{w\_{\mathcal{S}}^*}) + disc^{adv}\_{\mathcal{H}\Delta\mathcal{H}}(\mathcal{S},\mathcal{T}) + \mathcal{R}^{adv}\_\mathcal{T} (h\_\mathbf{w\_{\mathcal{S}}^*}, h\_\mathbf{w\_{\mathcal{T}}^*} ) + \mathcal{R}^{label-adv}\_\mathcal{T} (h\_\mathbf{w\_{\mathcal{T}}^*}, y\_\mathcal{T} )
\end{align}
$$
where the risk decomposition still holds. We will definitely update the notations in the revised article, and thank Reviewer RkC2 and K4Lj again for pointing this out!

---

> ### Author Response · Authors · 2022-11-13
> **General Comments: Technical Novelty**
>
> # 2. Technical Novelty (Reviewer RkC2 and K4Lj)
> We would argue that our proof techniques are fundamentally different to any of previous works on single domain adversarial Rademacher complexity, due to the definition of $\mathcal{H}\Delta\mathcal{H}$ Rademacher complexity.
> Taking classification setting for example, [Yin et al 19] consider the Rademacher complexity over the loss class between model predictions and labels, i.e., $\frak{R} = \mathbb{E}[\sup_{\mathbf{w}}\frac{1}{n}\sum_{i=1}^n \sigma_i \min_{\|\boldsymbol{\delta}\|\leq \infty} \mathbf{w}^\top (\mathbf{x}+\boldsymbol{\delta})]$, where the inner minimization problem is $\textbf{linear}$ in $\mathbf{w}$ and $\boldsymbol{\delta}$. We consider the loss between predictions among two models, hence the Rademacher complexity is $\frak{R} = \mathbb{E}[\sup_{\mathbf{w},\mathbf{w}'}\frac{1}{n}\sum_{i=1}^n \sigma_i \min_{\|\boldsymbol{\delta}\|\leq \infty} \mathbf{w}^\top (\mathbf{x}+\boldsymbol{\delta})\mathbf{w}'^\top (\mathbf{x}+\boldsymbol{\delta})]$, where the inner problem is quadratic in terms of $\mathbf{w}$ and $\boldsymbol{\delta}$. Hence, unlike previous works on single domain Rademacher complexity, where the inner problem has simple closed form solution, we have to use $\varepsilon$-nets and covering number idea to estimate upper bound. Another key technical contribution is the proof of lower bound results. For lower bound proofs, controlling the magnitude of Rademacher complexity with the inner problem being quadratic objective is significantly harder than linear objective.
> We derive the (complicated) closed form solution to inner quadratic programming, and leverage the symmetric property of Rademacher random variables to avoid heavy computation.

---

### Decision · Program_Chairs · 2023-01-20

**Decision:**

Reject

**Justification For Why Not Higher Score:**

N/A

**Justification For Why Not Lower Score:**

N/A

**Metareview: Summary, Strengths And Weaknesses:**

This paper aims to explain why domain generalization is much more difficult in the adversarial scenario, and provide a bound with respect to clean target error and adversarial source error. The experiments shows the theoretical results. The reviewers find many major issues about this paper. For example, the definition of adversarial error is different with the classical one, and many related work are not well clarified.